# Fusion in the periodic Temperley-Lieb algebra and connectivity operators of loop models

Yacine Ikhlef[a], Alexi Morin-Duchesne[b,c]

[a]*Sorbonne Université, CNRS, Laboratoire de Physique Théorique et Hautes Énergies, LPTHE, F-75005 Paris, France*

[b]*Max-Planck-Institut für Mathematik, 53111 Bonn, Germany*

[c]*Leibniz Universität Hannover, 30167 Hannover, Germany*

ikhlef @ lpthe.jussieu.fr      alexi.morin.duchesne @ gmail.com

## Abstract

In two-dimensional loop models, the scaling properties of critical random curves are encoded in the correlators of connectivity operators. In the dense $O(n)$ loop model, any such operator is naturally associated to a standard module of the periodic Temperley-Lieb algebra. We introduce a new family of representations of this algebra, with connectivity states that have two marked points, and argue that they define the fusion of two standard modules. We obtain their decomposition on the standard modules for generic values of the parameters, which in turn yields the structure of the operator product expansion of connectivity operators.

# 1  Introduction

In the study of critical two-dimensional statistical models, the conceptual framework and computational tools of Conformal Field Theory (CFT) have proven very efficient [1]. Most notably, the conformal minimal models provide a classification of scale invariant phase transitions and the conformal bootstrap fixes the multi-point correlation functions. A related area of interest is concerned with random curves within these critical models, to describe for instance interfaces of spin clusters, dense or dilute polymers, and the contours of percolation clusters. In the continuum scaling limit, these objects scale to random fractal curves, whose full description involves the study of their connectivity correlation functions [2]. For this, one needs to consider a model which comprises not only the operators describing the local degrees of freedom, but also those encoding the connectivity properties of the curves of interest. On the lattice, models of random polygons like the $O(n)$ loop model [3] are exactly adapted to this situation. The corresponding CFTs are typically non-rational [4] and logarithmic [5], namely they involve an infinite number of primary operators, but also logarithmic fields [6, 7] that are governed by a non-diagonalisable evolution operator in Euclidean space.

Critical random curves have been studied since the early times of CFT [8], and more recently they have motivated new advances in mathematical physics. In particular, some of the latter works deal with (i) logarithmic CFTs associated to loop models [9–11], (ii) structure constants of the algebra of connectivity operators [12–15], their relations with the imaginary Liouville CFT [16–18], and their numerical conformal bootstrap [19–23], and (iii) the representation theory of the underlying diagram algebra on the lattice [24–30] – namely, the Temperley-Lieb algebra. In particular, a good deal of consistent results were obtained in the case of connectivity operators sitting at the boundary of the system. From the CFT point of view, these boundary operators are degenerate under the Virasoro algebra [31]. Their fusion rules are known and their correlation functions are accessible analytically (see for example [32, 33]), either using the conformal bootstrap or from the exact knowledge of the singular vectors.

The fusion of the boundary operators has a counterpart in the $O(n)$ loop model at finite size, in terms of the fusion of representations of the diagrammatic algebras. For the ordinary Temperley-Lieb algebra $\mathsf{TL}_N(\beta)$, where $N$ is the system size and $\beta$ is the loop weight, the fusion procedure defines a representation of $\mathsf{TL}_{N_\mathsf{a}+N_\mathsf{b}}(\beta)$, out of a pair of representations of $\mathsf{TL}_{N_\mathsf{a}}(\beta)$ and $\mathsf{TL}_{N_\mathsf{b}}(\beta)$. This can be done in various ways. A first lattice construction of fusion [9, 34] is based on the representations of the Temperley-Lieb algebra that arise in the study of the loop model on a strip with certain integrable boundary conditions attached at each end of the strip. A second construction [35, 36] defines the new representations inductively using the inclusion $\mathsf{TL}_{N_\mathsf{a}}(\beta) \otimes \mathsf{TL}_{N_\mathsf{b}}(\beta) \subset \mathsf{TL}_{N_\mathsf{a}+N_\mathsf{b}}(\beta)$. This amounts to studying the lattice model on a domain shaped like a pair of pants: the degrees of freedom of two subsystems of sizes $N_\mathsf{a}$ and $N_\mathsf{b}$ evolve separately until they are joined into a larger system of size

$N = N_{\sf a} + N_{\sf b}$ where they then interact. The two constructions turn out to be equivalent and produce fusion rules that are consistent with those of the chiral boundary fields.

In the present work, we are interested in the operator algebra of *bulk* connectivity operators for critical loop models – a problem for which the above lattice approach needs to be adapted substantially. A fundamental step in this program is the construction of the lattice analogs of the operator product expansion of the non-chiral connectivity operators. In radial quantisation, the evolution operators, namely the Hamiltonians and transfer matrices, are seen as dilation operators acting radially. They are therefore endowed with periodic boundary conditions. As a result, the proper algebraic structure is the enlarged periodic Temperley-Lieb algebra $\mathcal{E}\mathsf{PTL}_N(\beta)$. A connectivity operator $\mathcal{O}_{k,x}(r)$ in the loop model is described by its number of defects $2k \in \mathbb{N}$ and its twist parameter $x \in \mathbb{C}^\times$. It is naturally associated, through the state-operator correspondence, with a standard module $\mathsf{W}_{k,x}(N)$ over $\mathcal{E}\mathsf{PTL}_N(\beta)$, spanned by link states with $2k$ defects attached to a marked point. In the scaling limit, these operators have conformal dimensions that depend continuously on $x$. They are in general not degenerate under the Virasoro algebra, and hence the usual method, based on the translation of null-vector equations into differential equations for the correlation functions, does not apply. For the same reasons, their fusion rules under the operator product expansion cannot be obtained by standard methods.

To define fusion at the lattice level, one should find a way to glue two periodic systems of sizes $N_{\sf a}$ and $N_{\sf b}$ into a larger periodic system of size $N = N_{\sf a} + N_{\sf b}$, a construction which is clearly not as straightforward as in the non-periodic case. There are in fact multiple ways to perform this gluing and thus to construct representations of $\mathcal{E}\mathsf{PTL}_N(\beta)$ from pairs of representations on smaller lattice sizes. Two such proposals were recently put forward [37–39] – see also [40]. In each case, the authors build representations from pairs of representations on smaller lattices, argue that these can be interpreted as the fusion of these representations, and obtain the module decomposition as direct sums of indecomposable representations. From the resulting module decompositions, it is readily observed that the two proposals [37, 38] and [39] are inequivalent. Moreover, it is presently unclear whether these two prescriptions for fusion are physically useful to compute the operator product expansion of the bulk connectivity operators.

In this paper, we present a new candidate for the fusion of representations of $\mathcal{E}\mathsf{PTL}_N(\beta)$, which we believe is a good lattice analog of the operator product expansion of the bulk connectivity operators. Consider two operators $\mathcal{O}_{k,x}(r_{\sf a})$ and $\mathcal{O}_{\ell,y}(r_{\sf b})$ in a correlation function of the loop model, that may potentially involve more such operators. To fully define the correlation function, one needs to keep track not only of the windings of the loop segments around the point $r_{\sf a}$, or around the point $r_{\sf b}$, but also of the loop segments that wind around both $r_{\sf a}$ and $r_{\sf b}$. For this reason, we introduce modules $\mathsf{X}_{k,\ell,x,y,z}(N)$ that depend not only on $x$ and $y$, but also on a third parameter $z$. These modules will be spanned by link states drawn on a disc with two marked points $\sf a$ and $\sf b$. In the absence of defects, the three free variables $x, y, z$ parameterise the weights of the different kinds of loops: $\alpha_{\sf a} = x + x^{-1}$, $\alpha_{\sf b} = y + y^{-1}$ and $\alpha_{\sf ab} = z + z^{-1}$. For non-zero defect numbers, the parameters $x, y, z$ instead couple to the winding of the defects around the two marked points. We will study the decomposition of $\mathsf{X}_{k,\ell,x,y,z}(N)$ over the standard modules and argue that it produces the fusion rule $\mathcal{O}_{k,x}(r_{\sf a}) \times \mathcal{O}_{\ell,y}(r_{\sf b})$.

**Summary of the results.** We focus on values $\beta = -q - q^{-1}$ of the loop weight where $q$ is not a root of unity. A first main result is the construction of a family of modules $\mathsf{X}_{k,\ell,x,y,z}(N)$ over $\mathcal{E}\mathsf{PTL}_N(\beta)$, for half-integers $k, \ell$, and $x, y, z \in \mathbb{C}^\times$. These modules correspond in the above sense to the fusion $\mathsf{W}_{k,x}(N_{\sf a}) \times \mathsf{W}_{\ell,y}(N_{\sf b})$ of standard modules, in a twist channel characterised by the parameter $z$. A second main result is the decomposition of $\mathsf{X}_{k,\ell,x,y,z}(N)$ over the irreducible standard modules, for

generic values of the parameters $q$ and $z$:

$$\mathsf{X}_{k,\ell,x,y,z}(N) \simeq \mathsf{W}_{k-\ell,z}(N) \oplus \bigoplus_{m=k-\ell+1}^{N/2} \bigoplus_{n=0}^{2m-1} \mathsf{W}_{m,z^{(k-\ell)/m}\exp(i\pi n/m)}(N)\,, \qquad k \geqslant \ell. \qquad (1.1)$$

The decomposition for $k < \ell$ is obtained from $\mathsf{X}_{k,\ell,x,y,z}(N) \simeq \mathsf{X}_{\ell,k,y,x,z^{-1}}(N)$. A third main result regards the decomposition for $q$ generic but $z$ set to $z = \pm q^r$, with $r$ a positive integer. We show examples where it includes a reducible yet indecomposable module with three composition factors.

**Outline of the paper.** In Section 2, we review the main properties of the algebra $\mathcal{E}\mathsf{PTL}_N(\beta)$, in particular the important results of Graham and Lehrer [41] on the standard modules $\mathsf{W}_{k,x}(N)$. We also propose an alternative construction of the Graham-Lehrer homomorphisms between standard modules for generic values of $q$. In Section 3, we recall the definition of fusion of standard modules for the ordinary Temperley-Lieb algebra, and we then present the definition of the new representations $\mathsf{X}_{k,\ell,x,y,z}(N)$ of $\mathcal{E}\mathsf{PTL}_N(\beta)$. In Section 4, we study the structure of $\mathsf{X}_{k,\ell,x,y,z}(N)$, and obtain the decomposition (1.1) for generic values of $q$ and $z$. We also discuss in an example the partially non-generic case $z = \pm q^r$ with $r$ an integer, namely in the module $\mathsf{X}_{0,0,x,y,z}(N)$, and find that it exhibits a reducible yet indecomposable module. In Section 5, we discuss the relation between the representations $\mathsf{X}_{k,\ell,x,y,z}(N)$ and the bulk connectivity operators, and in particular the consequences of the decomposition (1.1) for the computation of the correlation functions, in the scaling limit. Final remarks are given in Section 6. The properties of the Jones-Wenzl projectors are reviewed in Appendix A, and certain more technical computations of Section 4 are relegated to Appendix B.

## 2 The periodic Temperley-Lieb algebra and its standard modules

### 2.1 Definition of the algebra

The periodic Temperley-Lieb algebra, also called the affine Temperley-Lieb algebra, was first introduced by D. Levy in the context of the spin-$\frac{1}{2}$ XXZ chain and the Potts model [42]. Its representation theory was subsequently investigated by multiple groups [43, 41, 44, 45]. Here we work with the enlarged incarnation [46] of this algebra, $\mathcal{E}\mathsf{PTL}_N(\beta)$, which includes the rotation generators $\Omega$ and $\Omega^{-1}$. For the loop weight $\beta$, we use the convention

$$\beta = -q - q^{-1}\,, \qquad q \in \mathbb{C}^\times\,. \qquad (2.1)$$

We study the generic case, namely values of $q$ that are not roots of unity.

Let $N$ be an integer larger than 2. One set of generators for $\mathcal{E}\mathsf{PTL}_N(\beta)$ is the set $\{e_1, e_2, \ldots, e_N, \Omega, \Omega^{-1}\}$. These are subject to the relations

$$e_j^2 = \beta\, e_j\,, \qquad e_j\, e_{j\pm 1}\, e_j = e_j\,, \qquad e_i\, e_j = e_j\, e_i \qquad \text{for } |i-j| > 1\,, \qquad (2.2a)$$

$$\Omega\, e_j\, \Omega^{-1} = e_{j-1}\,, \qquad \Omega\, \Omega^{-1} = \Omega^{-1}\, \Omega = \mathbf{1}\,, \qquad e_{N-1}e_{N-2}\cdots e_2 e_1 = \Omega^2 e_1\,, \qquad (2.2b)$$

where $\mathbf{1}$ is the identity operator, and the indices $i, j$ are taken modulo $N$.

The elements of $\mathcal{E}\mathsf{PTL}_N(\beta)$ are represented by connectivity diagrams on a periodic system of $N$ sites. Two equivalent presentations are possible. In the first, the diagrams live inside a horizontal rectangle with periodic boundary conditions in the horizontal direction. There are $N$ nodes on the top segment and likewise $N$ nodes on the bottom segment, with the labels $1, \ldots, N$. The $2N$ nodes

are connected pairwise by non-intersecting loop segments. Two diagrams are identified if they can be mapped to one another by a continuous deformation of the loop segments preserving their endpoints, namely they are homotopic. In the second presentation, the loop segments live in an annulus and connect the nodes, which are drawn on the inner and outer circular edges of the annulus. For the generators and the identity, the diagrams are

$$\Omega = \quad \equiv \quad , \qquad \Omega^{-1} = \quad \equiv \quad , \qquad (2.3a)$$

$$e_j = \quad \equiv \quad \text{for} \quad 1 \leqslant j \leqslant N-1, \qquad (2.3b)$$

$$e_N = \quad \equiv \quad , \qquad \mathbf{1} = \quad \equiv \quad . \qquad (2.3c)$$

We draw a dashed segment between the nodes 1 and $N$ that indicates where the cut is performed to produce the rectangular diagram from the one on the annulus. Connectivity diagrams like those for $e_N$, $\Omega$ and $\Omega^{-1}$ have loop segments that cross this segment.

In the annulus presentation, the product $a_1 a_2$ of two elements of $\mathcal{E}\mathsf{PTL}_N(\beta)$ is obtained by drawing $a_2$ inside $a_1$. The new connectivity diagram is obtained by reading the connection of the nodes from the inner and outer perimeters. The diagram may also contain *contractible* loops, namely loops which do not wrap around the annulus. Each such loop is removed and replaced by a multiplicative factor $\beta$. Using this product, with words in the generators one can obtain any pairing of the $2N$ nodes by non-intersecting loop segments. Loop segments that tie the inner and outer perimeter can wind arbitrarily many times around the annulus. The algebra is thus infinite dimensional. Moreover, for $N$ even there can be an arbitrary number of non-contractible loops, namely closed loops that encircle the inner perimeter. Of course, the same definition of the action of the algebra applies to the rectangular diagrams, namely $a_1 a_2$ is obtained by drawing $a_2$ above $a_1$ and reading the new connectivity diagram from the lower and upper segments.

The (ordinary) Temperley-Lieb algebra $\mathsf{TL}_N(\beta)$ is the subalgebra of $\mathcal{E}\mathsf{PTL}_N(\beta)$ generated by $e_1, \ldots, e_{N-1}$. In the annular presentation, the restricted set of connectivity diagrams of $\mathsf{TL}_N(\beta)$ are those without loop segments crossing the dashed segment. Below, we use the diagrams on the annulus when discussing $\mathcal{E}\mathsf{PTL}_N(\beta)$ and resort to the rectangular presentation for results involving $\mathsf{TL}_N(\beta)$.

## 2.2  Useful elements of the algebra

In this subsection, we recall the definition of some elements of $\mathcal{E}\mathsf{PTL}_N(\beta)$ that play an important role in the next sections: the Jones-Wenzl projectors and the braid transfer matrices.

**Jones-Wenzl projectors.** The Jones-Wenzl projectors $P_1, \ldots, P_N$ [25, 27, 47] are elements of the ordinary Temperley-Lieb algebra $\mathsf{TL}_N(\beta) \subset \mathcal{E}\mathsf{PTL}_N(\beta)$. They are defined recursively as

$$P_1 = \mathbf{1}, \qquad P_{n+1} = P_n + \frac{[n]_q}{[n+1]_q} P_n e_n P_n, \tag{2.4}$$

where we use the notation

$$[k]_q = \frac{q^k - q^{-k}}{q - q^{-1}}, \qquad k \in \mathbb{C}, \tag{2.5}$$

whereby $\beta = -[2]_q$. We depict the Jones-Wenzl projectors as

$$P_n = \boxed{\phantom{xx} n \phantom{xx}} . \tag{2.6}$$

Some of the known properties of these projectors are given in Appendix A.

**Braid transfer matrices.** The braid tile is defined as

$$\tag{2.7}$$

A useful identity satisfied by the tiles is the *push-through* property

$$\tag{2.8}$$

The two braid transfer matrices $\boldsymbol{F}$ and $\bar{\boldsymbol{F}}$ in $\mathcal{E}\mathsf{PTL}_N(\beta)$ are defined in terms of the braid tile as

$$\tag{2.9}$$

and are therefore to be understood in what follows as a sum of $2^N$ connectivity diagrams of $\mathcal{E}\mathsf{PTL}_N(\beta)$. These operators have been studied previously in various contexts [48–51]. They are in the center of $\mathcal{E}\mathsf{PTL}_N(\beta)$, namely

$$\boldsymbol{F}\Omega = \Omega\boldsymbol{F}, \qquad \bar{\boldsymbol{F}}\Omega = \Omega\bar{\boldsymbol{F}}, \qquad \boldsymbol{F}e_j = e_j\boldsymbol{F}, \qquad \bar{\boldsymbol{F}}e_j = e_j\bar{\boldsymbol{F}}, \qquad j = 1, 2, \ldots, N. \tag{2.10}$$

## 2.3 Standard and vacuum modules

We denote by $\mathsf{W}_{k,z}(N)$ the standard modules of $\mathcal{E}\mathsf{PTL}_N(\beta)$, with $0 \leqslant k \leqslant \frac{N}{2}$, $2k \in \mathbb{Z}_{\geqslant 0}$, $2k \equiv N \bmod 2$ and $z \in \mathbb{C}^\times$. The link states of $\mathsf{W}_{k,z}(N)$ are diagrams drawn inside a disc wherein $N$ nodes from the perimeter are connected by non-intersecting loop segments. Moreover a marked point is drawn inside this disc. For $k = 0$, all the nodes are connected pairwise by arcs that do not pass through the marked point. For $k > 0$, $2k$ nodes from the perimeter are attached to the marked point by *defects*, and the

rest of the nodes are connected pairwise by arcs. Two link states are considered identical if they are homotopic. For example, here are the link states for $\mathsf{W}_{0,z}(4)$, $\mathsf{W}_{1,z}(4)$ and $\mathsf{W}_{2,z}(4)$:

$$\mathsf{W}_{0,z}(4): \tag{2.11a}$$

$$\mathsf{W}_{1,z}(4): \qquad\qquad\qquad\qquad\qquad\qquad\qquad \mathsf{W}_{2,z}(4): \tag{2.11b}$$

For convenience, we choose a presentation where the position of the marked point changes from diagram to diagram, instead of one where the arcs are deformed around the marked point. The two are of course equivalent. One can also draw these link states along a horizontal line. For instance, in this setup the last states of $\mathsf{W}_{0,z}(4)$ and $\mathsf{W}_{1,z}(4)$ drawn in (2.11) are depicted as ⊃⊂ and ⌣⌣. However, it will turn out to be more natural for the construction in Section 3.2 to draw the link states on discs.

The standard action is defined as follows. To compute $c \cdot w$, with $c \in \mathcal{E}\mathsf{PTL}_N(\beta)$ and $w \in \mathsf{W}_{k,z}(N)$, we draw $w$ inside $c$, join their loop segments, and read the new link state on the outer perimeter. The action of $c$ is interpreted as an evolution down a long cylinder of the state $w$, which is seen as the top cap of the cylinder. For $k = 0$, if a closed loop is created, it is removed and replaced by a weight $\alpha = z + z^{-1}$ if it is non-contractible (namely it wraps around the marked point), and $\beta$ if it is contractible. For $k > 0$, if two defects are connected, the result is set to zero. Otherwise each closed loop is replaced by a factor of $\beta$. (Loops cannot encircle the marked point in this case.) Moreover, if a defect crosses the dashed line, it is unwound at the cost of a twist factor. This factor is $z$ if the defect crosses the dashed line with the marked point to its right as it evolves down the cylinder, and $z^{-1}$ if it crosses this line with the marked point to its left. (The role of $z$ is thus quite different for $k = 0$ and for $k > 0$.) Here are examples of the standard action for $N = 4$:

$$e_1 \cdot \qquad = \beta \qquad , \qquad e_3 \cdot \qquad = \alpha \qquad , \qquad e_4 \cdot \qquad = \qquad , \tag{2.12a}$$

$$\Omega \cdot \qquad = z \qquad , \qquad e_4 \cdot \qquad = z^{-1} \qquad , \qquad e_2 \cdot \qquad = 0 . \tag{2.12b}$$

The dimension of $\mathsf{W}_{k,z}(N)$ is given by the binomial coefficient

$$\dim \mathsf{W}_{k,z}(N) = \binom{N}{\frac{N-2k}{2}} . \tag{2.13}$$

The standard modules define finite-dimensional representations over an infinite-dimensional algebra, so they admit extra relations. First, for the special case $k = 0$ with $N$ even, the assignment of weights $\alpha$ to the non-contractible loops is achieved from the relations

$$\forall w \in \mathsf{W}_{0,z}(N), \qquad E\,\Omega^{\pm 1}E \cdot w = (\underbrace{z + z^{-1}}_{=\,\alpha})\, E \cdot w , \tag{2.14}$$

where $E = e_2 e_4 \cdots e_N$. Second, the element $\Omega^N$, which is central in $\mathcal{E}\mathsf{PTL}_N(\beta)$, acts as a multiple of the identity on the standard modules:

$$\forall w \in \mathsf{W}_{k,z}(N), \qquad \Omega^N \cdot w = z^{2k}\, w . \tag{2.15}$$

Hence, the possible eigenvalues $\omega_n(z)$ of $\Omega$ in $\mathsf{W}_{k,z}(N)$ are $N$-th roots of $z^{2k}$:

$$\omega_n(z) = z^{2k/N} \mathrm{e}^{2\pi \mathrm{i} n/N}, \qquad n = 0, \ldots, N-1. \tag{2.16}$$

The projector on the eigenspace of $\Omega$ of eigenvalue $\omega_n(z)$ is

$$\Pi_n = \frac{1}{N} \sum_{j=0}^{N-1} [\omega_n(z)]^{-j} \, \Omega^j. \tag{2.17}$$

We also note that, on the standard module $\mathsf{W}_{k,z}(N)$, the central elements $\boldsymbol{F}$ and $\bar{\boldsymbol{F}}$ (see Section 2.2) act as multiples of the identity, namely

$$\forall w \in \mathsf{W}_{k,z}(N), \qquad \boldsymbol{F} \cdot w = \left( z q^k + \frac{1}{z q^k} \right) w, \qquad \bar{\boldsymbol{F}} \cdot w = \left( \frac{z}{q^k} + \frac{q^k}{z} \right) w. \tag{2.18}$$

We also define the *vacuum module* $\mathsf{V}(N)$ over $\mathcal{E}\mathsf{PTL}_N(\beta)$. This module is constructed on the vector space of link states drawn on a disc with $N$ nodes and no marked point. To illustrate, the bases of $\mathsf{V}(4)$ and $\mathsf{V}(6)$ are

$$\tag{2.19a}$$

$$\tag{2.19b}$$

The module $\mathsf{V}(N)$ has the same dimension as the standard module $\mathsf{V}_0(N)$ over $\mathsf{TL}_N(\beta)$ with no defects, namely

$$\dim \mathsf{V}(N) = \binom{N}{\frac{N}{2}} - \binom{N}{\frac{N-2}{2}}. \tag{2.20}$$

The action of $\mathcal{E}\mathsf{PTL}_N(\beta)$ on $\mathsf{V}(N)$ is similar to the action on standard modules, with the difference that there are no non-contractible loops, and therefore all closed loops are assigned the weight $\beta$.

## 2.4 The Graham-Lehrer theorem

In this subsection, we recall the main result of Graham and Lehrer [41] and its implications for the module decomposition of the standard modules.[1] In their Theorem 3.4, they construct non-zero homomorphisms between two standard modules $\mathsf{W}_{\ell,y}(N)$ and $\mathsf{W}_{k,z}(N)$ for special values of the twist parameters $y$ and $z$:

$$\mathsf{W}_{\ell,\varepsilon q^k}(N) \to \mathsf{W}_{k,\varepsilon q^\ell}(N), \qquad \ell > k, \qquad \varepsilon \in \{-1, +1\}, \qquad q \in \mathbb{C}^\times. \tag{2.21}$$

Since the algebra $\mathcal{E}\mathsf{PTL}_N(\beta)$ and its standard modules are invariant under the transformation $q \leftrightarrow q^{-1}$, there are also non-zero homomorphisms

$$\mathsf{W}_{\ell,\varepsilon q^{-k}}(N) \to \mathsf{W}_{k,\varepsilon q^{-\ell}}(N), \qquad \ell > k, \qquad \varepsilon \in \{-1, +1\}, \qquad q \in \mathbb{C}^\times. \tag{2.22}$$

---

[1]We adopt a different notation compared to Graham and Lehrer, with the equivalence given by $(\mathsf{W}_{k,z})_{\mathrm{GL}} \equiv \mathsf{W}_{k/2,z}$.

These are the only non-trivial homomorphisms that exist between standard modules. In Section 2.5, we give a construction of such homomorphisms for $q$ generic, that is explicit in terms of link states.

We note that the condition on $z$ and $y$ for a non-zero homomorphism $\mathsf{W}_{\ell,y}(N) \to \mathsf{W}_{k,z}(N)$ to exist can be reformulated in terms of the braid transfer matrices, namely, it will exist if and only if $\ell > k$, and $\boldsymbol{F}$ and $\bar{\boldsymbol{F}}$ have identical eigenvalues on the two modules [51]. We say that the parameter $z$ of $\mathsf{W}_{k,z}(N)$ is generic if it is not of the form $z = \varepsilon q^{\pm\ell}$ with $\varepsilon \in \{-1, +1\}$ and $\ell \in \{k+1, k+2, \dots, N/2\}$. Moreover, we say that $q$ is *generic* if it is not a solution of $q^{2m} = 1$ with $m \in \mathbb{Z}$. In the following, we mainly focus on the generic values of $q$.

The results of Graham and Lehrer yield the structure of the standard modules. For generic values of $q$ and $z$, the standard module $\mathsf{W}_{k,z}(N)$ is irreducible. For $q$ generic but $z$ non-generic, the non-zero homomorphism discussed above implies that $\mathsf{W}_{k,z=\varepsilon q^{\pm\ell}}(N)$ has a submodule isomorphic to $\mathsf{W}_{\ell,\varepsilon q^{\pm k}}(N)$. This module is itself irreducible, so we write $\mathsf{W}_{\ell,\varepsilon q^{\pm k}}(N) \simeq \mathsf{I}_{\ell,\varepsilon q^{\pm k}}(N)$. The quotient $\mathsf{W}_{k,\varepsilon q^{\pm\ell}}(N)/\mathsf{I}_{\ell,\varepsilon q^{\pm k}}(N)$ is also irreducible, and we denote it by $\mathsf{I}_{k,\varepsilon q^{\pm\ell}}(N)$. Furthermore, since $\mathsf{W}_{k,z=\varepsilon q^{\pm\ell}}(N)$ admits only one proper submodule $\mathsf{I}_{\ell,\varepsilon q^{\pm k}}(N)$, it does *not* decompose as a direct sum $\mathsf{I}_{k,\varepsilon q^{\pm\ell}}(N) \oplus \mathsf{I}_{\ell,\varepsilon q^{\pm k}}(N)$.

The above results on the structure of standard modules are expressed in terms of Loewy diagrams as

$$\mathsf{W}_{k,z}(N) \simeq \mathsf{I}_{k,z}(N) \qquad \text{for } q, z \text{ generic}, \tag{2.23}$$

$$\mathsf{W}_{k,\varepsilon q^{\pm\ell}}(N) \simeq \begin{array}{c} \mathsf{I}_{k,\varepsilon q^{\pm\ell}}(N) \\ \searrow \\ \mathsf{I}_{\ell,\varepsilon q^{\pm k}}(N) \end{array} \qquad \ell > k, \qquad \varepsilon \in \{-1, +1\}, \qquad q \text{ generic}. \tag{2.24}$$

For a gentle introduction to Loewy diagrams, we direct the reader to the appendix of [7]. In short, we recall that the Loewy diagram associated to a module $M$ describes a filtration by submodules

$$0 = M_0 \subset M_1 \subset \cdots \subset M_n = M, \tag{2.25}$$

where each subquotient $Q_j = M_j/M_{j-1}$ is the socle (namely, the maximal completely reducible submodule) of $M/M_{j-1}$. The irreducible components of a subquotient $Q_j$ are called the *composition factors* of $M$. Each node of the Loewy diagram is associated to a composition factor. An arrow $I \to I'$ in the Loewy diagram, where $I$ and $I'$ are composition factors belonging to the subquotients $Q_j$ and $Q_{j-1}$ respectively, indicates that the algebra acting on $I$ can produce a non-zero element of $I'$, but not vice-versa.

In contrast to standard modules, the vacuum module $\mathsf{V}(N)$ is always irreducible. We note that a module isomorphic to $\mathsf{V}(N)$ appears as a composition factor in the standard module $\mathsf{W}_{0,-q}(N)$. Indeed, the latter is a reducible module, with a submodule isomorphic to $\mathsf{W}_{1,-1}(N)$. The irreducible quotient module is isomorphic to the vacuum module:

$$\mathsf{I}_{0,-q}(N) = \mathsf{W}_{0,-q}(N)/\mathsf{W}_{1,-1}(N) \simeq \mathsf{V}(N). \tag{2.26}$$

The modules discussed above have one or two composition factors. In general, Loewy diagrams can have more than two layers, with composition factors that belong neither to the socle nor the head. This can occur for instance for standard modules if both $z$ and $q$ are non-generic. In this case, all non-zero homomorphisms are still of the form (2.21) and (2.22). However, there can be more than one such homomorphisms into a given module $\mathsf{W}_{k,z}(N)$, characterised by different values of $\ell$. The resulting module decomposition is more complex and involves more than two composition factors. It is however beyond the scope of this paper.

## 2.5  An alternative construction of the homomorphisms for $q$ generic

In this subsection, we give an alternative construction of the homomorphisms $\mathsf{W}_{\ell,y}(N) \to \mathsf{W}_{k,z}(N)$ for generic values of $q$ (but non-generic values of $y$ and $z$), that avoids the sum over forest polynomials used by Graham and Lehrer. We shall use the following strategy, in which the defect numbers $k$ and $\ell$ are fixed, and the system size $N$ varies. We first treat the "fundamental case" of a system size $N = 2\ell$, namely the smallest system size where both $\mathsf{W}_{k,z}(N)$ and $\mathsf{W}_{\ell,y}(N)$ are defined (recall that $\ell > k$): the module $\mathsf{W}_{\ell,y}(2\ell)$ is actually one-dimensional, and hence the construction of the homomorphism amounts to defining a vector $w_{k,z}(\ell) \in \mathsf{W}_{k,z}(2\ell)$ and proving that the action of the algebra on $w_{k,z}(\ell)$ is identical to its action on the single basis state of $\mathsf{W}_{\ell,y}(2\ell)$. Then, as a second step, we consider larger values of the system size $N$, for which we define the homomorphism using what we call an *insertion map*, based on the properties of the vector $w_{k,z}(\ell)$ constructed previously.

**The case $N = 2\ell$.**   Let us define the unique link state $v_k(\ell) \in \mathsf{W}_{k,z}(2\ell)$ that has all of its $\ell - k$ arcs crossing the dashed segment between the marked point and the perimeter of the circle. We also define $w_{k,z}(\ell) = P_{2\ell} \cdot v_k(\ell)$. These two states are depicted as

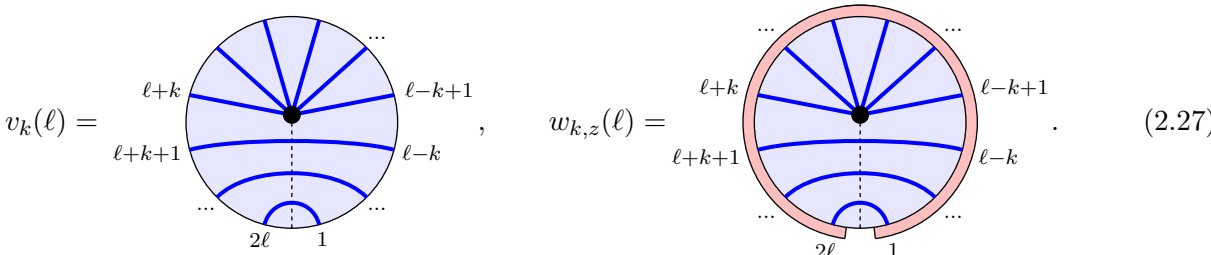

$$v_k(\ell) = \qquad , \qquad w_{k,z}(\ell) = \qquad . \qquad (2.27)$$

In the second diagram, the projector $P_{2\ell}$ is drawn as a pink ribbon encircling $v_k(\ell)$. Clearly, these two states are well-defined for both generic and non-generic values of $z$. However, for non-generic values of $z$, the state $w_{k,z}(\ell) \in \mathsf{W}_{k,z}(2\ell)$ has certain special properties, namely it satisfies the three relations

$$w_{k,z}(\ell) \neq 0, \qquad \Omega^{\pm 1} \cdot w_{k,z}(\ell) = y^{\pm 1} w_{k,z}(\ell), \qquad e_j \cdot w_{k,z}(\ell) = 0, \qquad j = 1, \ldots, 2\ell, \qquad (2.28)$$

for $(y, z) = (\varepsilon q^{\pm k}, \varepsilon q^{\pm \ell})$.   The proofs of these relations are given in Appendix B, and rely on the properties of the Jones-Wenzl projectors discussed in Appendix A.

The module $\mathsf{W}_{\ell,y}(2\ell)$ is one-dimensional – it is spanned by the single link state $v_\ell(\ell)$. Altogether, the properties (2.28) ensure that the linear map $\mathsf{W}_{\ell,y}(2\ell) \to \mathsf{W}_{k,z}(2\ell)$ defined by $v_\ell(\ell) \mapsto w_{k,z}(\ell)$ is a homomorphism of modules for $(y, z) = (\varepsilon q^{\pm k}, \varepsilon q^{\pm \ell})$, with $\varepsilon \in \{-1, +1\}$.

**The case $N > 2\ell$.**   In this case, we define a linear map $\Phi : \mathsf{W}_{\ell,y}(N) \to \mathsf{W}_{k,z}(N)$ on the link states of $\mathsf{W}_{\ell,y}(N)$ using what we call *the insertion algorithm*. Given a link state $v \in \mathsf{W}_{\ell,y}(N)$, we construct $\Phi(v) \in \mathsf{W}_{k,z}(N)$ by selecting the $2\ell$ nodes of $v$ tied to defects, erasing these defects, and inserting instead the linear combination corresponding to the state $w_{k,z}(\ell)$ constructed in (2.27). We illustrate this with an example for the case $\mathsf{W}_{2,q}(12) \to \mathsf{W}_{1,q^2}(12)$:

$$v = \qquad \mapsto \qquad \Phi(v) = \qquad . \qquad (2.29)$$

As a second example, for the homomorphism $W_{1,1}(4) \to W_{0,q}(4)$, the map $\Phi$ applied to the states of $W_{1,1}(4)$ in the basis (2.11b) yields the four states



$$\tag{2.30}$$

From the properties (2.28), it is clear that the action of $\mathcal{E}\mathsf{PTL}_4(\beta)$ on these four states is invariant. In other words, these four states span a submodule of $W_{0,q}(4)$. Indeed, this action yields a vanishing result if two nodes attached to the projector are connected together. Otherwise, it modifies the connection between the nodes and the projector, in precisely the same way as it does for the defects in $W_{1,1}(4)$. In the general case, for $(y,z) = (\varepsilon q^{\pm k}, \varepsilon q^{\pm \ell})$, the map $\Phi$ is non-zero, and satisfies the homomorphism relation

$$\forall a \in \mathcal{E}\mathsf{PTL}_N(\beta), \qquad \forall v \in W_{\ell,y}(N), \qquad a \cdot \Phi(v) = \Phi(a \cdot v), \tag{2.31}$$

and hence the states $\Phi(v)$ with $v \in W_{\ell,y}(N)$ form a non-zero submodule of $W_{k,z}(N)$.

We note that this insertion algorithm was used previously to compute the determinant of the Gram matrix of the standard modules [52]. This idea will also turn out to be useful in Section 4, although in a slightly different form, to obtain the module decompositions of the representations $X_{k,\ell,x,y,z}(N)$.

# 3 Fusion of standard modules

## 3.1 Fusion for the ordinary Temperley-Lieb algebra

In this section, we review the construction of fusion for standard modules over the ordinary Temperley-Lieb algebra $\mathsf{TL}_N(\beta)$. This construction can be defined either algebraically or diagrammatically [9, 35, 34, 36]. We denote by $V_k(N)$ the standard module over $\mathsf{TL}_N(\beta)$ with $2k$ defects, and draw the link states as planar diagrams above a horizontal line. For instance, the two link states that span $V_{1/2}(3)$ are ⌞◠ and ◠⌟.

The algebraic construction of fusion is based on the fact that $\mathsf{TL}_{N_\mathsf{a}}(\beta) \otimes \mathsf{TL}_{N_\mathsf{b}}(\beta)$ is a subalgebra of $\mathsf{TL}_{N_\mathsf{a}+N_\mathsf{b}}(\beta)$, and uses the idea of induction. The fusion $V_k(N_\mathsf{a}) \times V_\ell(N_\mathsf{b})$ of two standard modules is defined as the representation of $\mathsf{TL}_{N_\mathsf{a}+N_\mathsf{b}}(\beta)$

$$V_{k,\ell}(N_\mathsf{a} + N_\mathsf{b}) = \mathsf{TL}_{N_\mathsf{a}+N_\mathsf{b}}(\beta) \otimes_{\mathsf{TL}_{N_\mathsf{a}}(\beta) \otimes \mathsf{TL}_{N_\mathsf{b}}(\beta)} \left( V_k(N_\mathsf{a}) \otimes V_\ell(N_\mathsf{b}) \right). \tag{3.1}$$

This can be understood as follows. First, a basis for the vector space will include all the states of the form $v \otimes w$ with $v \in V_k(N_\mathsf{a})$ and $w \in V_\ell(N_\mathsf{b})$. Acting on $v \otimes w$ with $a \in \mathsf{TL}_{N_\mathsf{a}+N_\mathsf{b}}(\beta)$, we obtain $(a_1 \cdot v) \otimes (a_2 \cdot w)$ if $a = a_1 \otimes a_2 \in \mathsf{TL}_{N_\mathsf{a}}(\beta) \otimes \mathsf{TL}_{N_\mathsf{b}}(\beta)$. If $a \notin \mathsf{TL}_{N_\mathsf{a}}(\beta) \otimes \mathsf{TL}_{N_\mathsf{b}}(\beta)$, then $a(v \otimes w)$ cannot be simplified in terms of the standard action on $V_k(N_\mathsf{a})$ and $V_\ell(N_\mathsf{b})$. It instead produces a new state of the vector space. One thus acts successively with elements of the algebra until a complete basis is obtained. This is the meaning of $\otimes_{\mathsf{TL}_{N_\mathsf{a}} \otimes \mathsf{TL}_{N_\mathsf{b}}}$ in (3.1): it acts as a filter and only lets through elements of $\mathsf{TL}_{N_\mathsf{a}+N_\mathsf{b}}(\beta)$ that lie in the subalgebra $\mathsf{TL}_{N_\mathsf{a}}(\beta) \otimes \mathsf{TL}_{N_\mathsf{b}}(\beta)$.

To illustrate, we consider the example $V_{1/2}(3) \otimes V_1(2)$. A basis of $V_{1/2,1}(5)$ is made of the elements

$$\begin{aligned}
&◠⌟ \otimes ⌣⌣, \quad ⌞◠ \otimes ⌣⌣, \quad e_3(⌞◠ \otimes ⌣⌣), \quad e_4 e_3(⌞◠ \otimes ⌣⌣), \quad e_3(◠⌟ \otimes ⌣⌣), \\
&e_2 e_3(◠⌟ \otimes ⌣⌣), \quad e_4 e_3(◠⌟ \otimes ⌣⌣), \quad e_2 e_4 e_3(◠⌟ \otimes ⌣⌣), \quad e_3 e_2 e_4 e_3(◠⌟ \otimes ⌣⌣).
\end{aligned} \tag{3.2}$$

In this algebraic setup, it is a priori not obvious that this constitutes a full basis, nor that the resulting representation depends only on the sum $N_{\mathsf{a}} + N_{\mathsf{b}}$. For instance, one could think that the state $e_2 e_3 (\,\text{⌞⌣}\, \otimes \,\text{⌊⌋}\,)$ is missing from the basis. However the simple calculation

$$e_2 e_3 (\,\text{⌞⌣}\, \otimes \,\text{⌊⌋}\,) = \frac{1}{\beta} e_2 e_3 (e_2 \cdot \,\text{⌞⌣}\, \otimes \,\text{⌊⌋}\,) = \frac{1}{\beta} e_2 e_3 e_2 (\,\text{⌞⌣}\, \otimes \,\text{⌊⌋}\,)$$

$$= \frac{1}{\beta} e_2 (\,\text{⌞⌣}\, \otimes \,\text{⌊⌋}\,) = \,\text{⌞⌣}\, \otimes \,\text{⌊⌋}\, \tag{3.3}$$

shows that this state already appears in (3.2).

A second equivalent construction of the fusion $\mathsf{V}_k(N_{\mathsf{a}}) \times \mathsf{V}_\ell(N_{\mathsf{b}})$ is diagrammatic. The module $\mathsf{V}_{k,\ell}(N_{\mathsf{a}} + N_{\mathsf{b}})$ is defined on the vector space spanned by link states on $N_{\mathsf{a}} + N_{\mathsf{b}}$ nodes, with defect numbers that vary between $2|k - \ell|$ and $2(k + \ell)$. These defects are partitioned into two subsets: the first contains the leftmost $(r + k - \ell)$ defects and the second the rightmost $(r - k + \ell)$ defects, with $r$ taking the values $|k - \ell|, |k - \ell| + 1, \ldots, k + \ell$ for the different link states. In the diagrams, we use different colors to distinguish defects from the two subsets. For instance, the link states that span $\mathsf{V}_{1/2,1}(5)$ are

$$\text{⌣⌋⌋⌋}, \quad \text{⌊⌣⌋⌋}, \quad \text{⌊⌊⌣⌋}, \quad \text{⌊⌊⌊⌣}, \quad \text{⌣⌣⌋}, \quad \text{⌢⌋}, \quad \text{⌣⌋⌣}, \quad \text{⌊⌣⌣}, \quad \text{⌊⌢}. \tag{3.4}$$

The action of $\mathsf{TL}_N(\beta)$ on $\mathsf{V}_{k,\ell}(N_{\mathsf{a}} + N_{\mathsf{b}})$ is defined similarly to the action on standard modules. The only difference regards the weights assigned to the connection of defects. If $a \in \mathsf{TL}_N(\beta)$ acting on a link state in $\mathsf{V}_{k,\ell}(N)$ connects two defects from the same subset, then the result is set to zero. If $a$ connects two defects from different subsets, the result is not set to zero. These defects are instead removed with a weight 1.

The identification between the states in (3.2) and (3.4) is clear, as we drew them in the same order in the two bases. In general, one can show that the action of $\mathsf{TL}_{N_{\mathsf{a}} + N_{\mathsf{b}}}(\beta)$ on $\mathsf{V}_{k,\ell}(N_{\mathsf{a}} + N_{\mathsf{b}})$ is identical in the algebraic and diagrammatic constructions, so that the two definitions are indeed equivalent. Furthermore, in the diagrammatic setup, the definition of the full basis is clear, and by definition the representations depend only on $N_{\mathsf{a}}$ and $N_{\mathsf{b}}$ through their sum $N_{\mathsf{a}} + N_{\mathsf{b}}$. In the above example, a convenient change of basis shows that, for generic values of $q$, the module $\mathsf{V}_{1/2,1}(5)$ is isomorphic to the direct sum $\mathsf{V}_{1/2}(5) \oplus \mathsf{V}_{3/2}(5)$. These two summands account for the two possible values of the total number of defects of the states in (3.4). More generally, the fusion of $\mathsf{V}_k(N_{\mathsf{a}})$ and $\mathsf{V}_\ell(N_{\mathsf{b}})$ decomposes as

$$\mathsf{V}_k(N_{\mathsf{a}}) \times \mathsf{V}_\ell(N_{\mathsf{b}}) = \mathsf{V}_{k,\ell}(N) \simeq \bigoplus_{m=|k-\ell|}^{k+\ell} \mathsf{V}_m(N), \qquad \text{for } q \text{ generic}, \tag{3.5}$$

where $N = N_{\mathsf{a}} + N_{\mathsf{b}}$. This has the same structure as the tensor product of $s\ell(2)$ irreducible representations.

## 3.2 Representations over $\mathcal{E}\mathsf{PTL}_N(\beta)$ with two marked points

In this section, we introduce new families of representations $\mathsf{X}_{k,\ell,x,y,z}(N)$ of $\mathcal{E}\mathsf{PTL}_N(\beta)$, for $x, y, z \in \mathbb{C}^\times$, which we interpret as the fusion $\mathsf{W}_{k,x}(N) \times_z \mathsf{W}_{\ell,y}(N)$ of standard modules. Our construction is inspired from the diagrammatic definition of fusion for $\mathsf{TL}_N(\beta)$ described in Section 3.1. As detailed below, the parameter $z$ is a free variable in the construction that describes the interaction of the two standard modules in their fusion. In the absence of defects, it parameterises the weight of certain classes of loops, whereas in the presence of defects, it couples to the winding of the defects.

The module $\mathsf{X}_{k,\ell,x,y,z}(N)$ is spanned by link states living on a disc with two marked points $\mathsf{a}$ and $\mathsf{b}$. We color these points respectively in green and purple in the diagrams. The circular perimeter has $N$ nodes attached to loop segments. They can either be connected pairwise by arcs, be connected by a defect to point $\mathsf{a}$, or be connected by a defect to point $\mathsf{b}$. We also define a point $\mathsf{c}$ on the perimeter between the nodes $N$ and 1, and draw two dashed segments $\mathsf{ac}$ and $\mathsf{bc}$. Certain arcs may separate the points $\mathsf{a}$ and $\mathsf{b}$: we call these *through-arcs*. To illustrate, here are three such link states for $N = 12$:



$$(3.6)$$

We split our description of the modules $\mathsf{X}_{k,\ell,x,y,z}(N)$ in three cases: (i) $k = \ell = 0$, (ii) $k > 0, \ell = 0$ and $k = 0, \ell > 0$, and (iii) $k, \ell > 0$.

**The case $k = \ell = 0$.** For $\mathsf{X}_{0,0,x,y,z}(N)$, the link states that span the vector space have no defects attached to either $\mathsf{a}$ or $\mathsf{b}$. For instance, the full set of link states of $\mathsf{X}_{0,0,x,y,z}(4)$ is

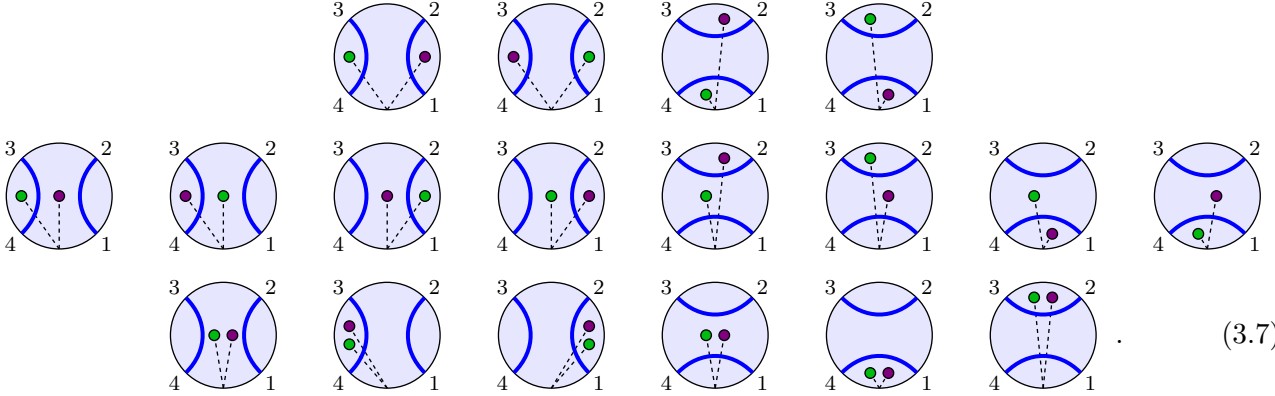

$$(3.7)$$

Likewise, the state $v_1$ in (3.6) is a link state of $\mathsf{X}_{0,0,x,y,z}(12)$. Here, we choose the convention where the loop segments are fixed whereas the position of the two marked points vary according to the link state. The other convention is also possible, but makes the diagrams more tangled up.

The action $a \cdot v$ of $a \in \mathcal{E}\mathsf{PTL}_N(\beta)$ on a link state $v \in \mathsf{X}_{0,0,x,y,z}(N)$ is obtained by drawing $v$ inside $a$ and reading the new link state from the outer perimeter. In this construction, the point $\mathsf{c}$ between the nodes $N$ and 1 is moved to the outer perimeter, so that the dashed segments once again connect the marked points to the link state's perimeter. A closed loop formed in the process is erased and is assigned (i) a weight $\alpha_{\mathsf{a}} = x + x^{-1}$ if it wraps around the point $\mathsf{a}$ only, (ii) a weight $\alpha_{\mathsf{b}} = y + y^{-1}$ if it wraps around the point $\mathsf{b}$ only, (iii) a weight $\alpha_{\mathsf{ab}} = z + z^{-1}$ if it wraps around both $\mathsf{a}$ and $\mathsf{b}$, and (iv) a weight $\beta$ it encircles neither $\mathsf{a}$ nor $\mathsf{b}$. Here are examples to illustrate:

$$e_3 \cdot \left( \vcenter{} \right) = \alpha_{\mathsf{a}} \left( \vcenter{} \right), \qquad e_2 \cdot \left( \vcenter{} \right) = \alpha_{\mathsf{ab}} \left( \vcenter{} \right), \qquad (3.8\mathrm{a})$$

$$e_2 e_4 \cdot \left( \vcenter{} \right) = \alpha_{\mathsf{b}} \left( \vcenter{} \right), \qquad e_1 \cdot \left( \vcenter{} \right) = \beta \left( \vcenter{} \right). \qquad (3.8\mathrm{b})$$

**The cases $k > 0, \ell = 0$ and $k = 0, \ell > 0$.** We describe the first case only, as the second is obtained from the first by interchanging the marked points $\mathsf{a}$ and $\mathsf{b}$, and changing $k \to \ell$, $x \leftrightarrow y$ and $z \to z^{-1}$. The link states that span the vector space of $\mathsf{X}_{k,0,x,y,z}(N)$ have $2k$ defects attached to the point $\mathsf{a}$, no defects attached to the point $\mathsf{b}$, and $\frac{N}{2} - k$ arcs. Some of these arcs can be through-arcs. Two link states $v_1$ and $v_2$ with no through-arcs are identified up to a twist factor if one can transform $v_1$ into $v_2$ by only pushing the marked point $\mathsf{b}$ across some defects. This is clarified below when we define the action of the algebra. Therefore, the only states with no through-arcs that we select for the basis of $\mathsf{X}_{k,0,x,y,z}(N)$ are those wherein the defects do not cross either of the two dashed segments. This is achieved by correctly choosing the location of the marked point $\mathsf{b}$ for each link state without through-arcs. For instance, the full set of states of $\mathsf{W}_{1,0,x,y,z}(4)$ is

$$(3.9)$$

Likewise, the state $v_2$ in (3.6) is a link state of $\mathsf{X}_{1,0,x,y,z}(12)$.

The action of $a \in \mathcal{E}\mathsf{PTL}_N(\beta)$ on a link state $v$ is obtained as before by drawing $v$ inside $a$ and reading the new link state from the outer perimeter. In this construction, the point $\mathsf{c}$ is again moved outwards, to the new link state's perimeter. A closed loop is assigned a weight $\alpha_\mathsf{b} = y + y^{-1}$ if it encircles point $\mathsf{b}$ and $\beta$ if it does not. If two defects are connected together, the result is set to zero. As a result, if the action of $a \in \mathcal{E}\mathsf{PTL}_N(\beta)$ creates a closed loop that encircles the point $\mathsf{a}$, the overall weight will vanishes because $a$ also connects defects of $\mathsf{a}$ together. Moreover, if a defect that evolves down the cylinder crosses the segment $\mathsf{ac}$, the segment $\mathsf{bc}$, or both, it produces a twist factor as follows. If it crosses the edge $\mathsf{ac}$, it is assigned a weight $x$ if the point $\mathsf{a}$ lies to its right and $x^{-1}$ if $\mathsf{a}$ lies to its left. Likewise, if a defect crosses the dashed segment $\mathsf{bc}$, it is assigned a weight $x^{-1}z$ if $\mathsf{b}$ lies to its right and $xz^{-1}$ if $\mathsf{b}$ lies to its left. If a defect crosses more than one dashed segment, then the resulting twist factor is the product of the corresponding weights. If it crosses none of the two segments, it is given the weight 1. Here are examples to illustrate:

$$(3.10\text{a})$$

$$(3.10\text{b})$$

$$(3.10\text{c})$$

We note that the second example in (3.10b) can be computed in two steps as

$$(3.11)$$

The two rightmost link states appearing in this example are identical up to the position of the point $\mathsf{b}$ with respect to the defects attached to point $\mathsf{a}$. As explained above in describing the link state basis, these two link states are identified up to a twist factor, here equal to $xz^{-1}$. In our convention, only the rightmost link state is selected to be a part of the basis (3.9). Finally, we summarize the rules for the unwinding of defects with the following examples:

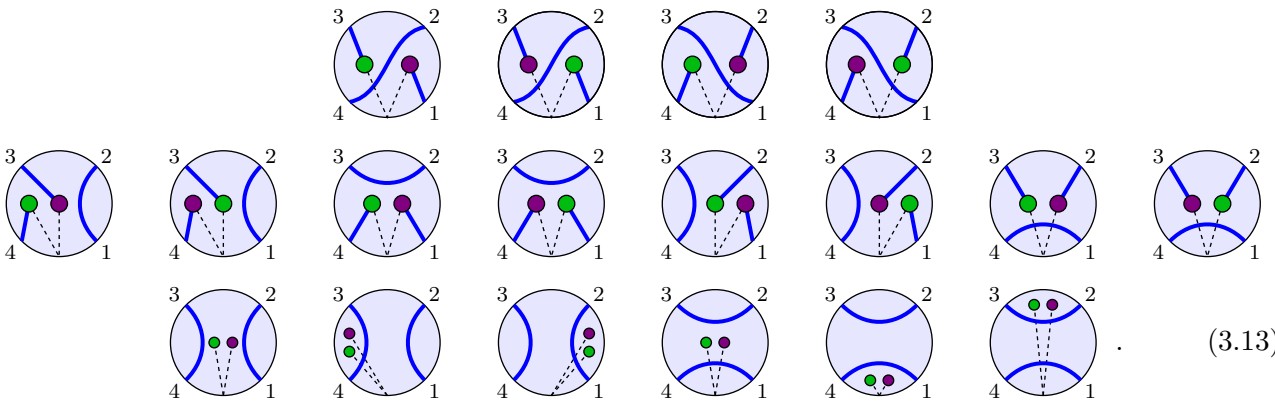

(3.12)

**The case $k, \ell > 0$.** The link states that span the vector space of $\mathsf{X}_{k,\ell,x,y,z}(N)$ have a total number $2r$ of defects that varies, with $r$ taking the values $|k-\ell|, |k-\ell|+1, \ldots, k+\ell$. Of these, $(r+k-\ell)$ are attached to point $\mathsf{a}$ and $(r+\ell-k)$ are attached to point $\mathsf{b}$. The remaining $(N-2r)$ nodes are connected pairwise by arcs. These can be through-arcs if $r = k+\ell$, but not if $r < k+\ell$.[2] As in the previous case, the only states with no through-arcs that we select for the basis of $\mathsf{X}_{k,\ell,x,y,z}(N)$ are those wherein no defects cross the two dashed segments. This is done by choosing appropriately the location of the two marked points. For instance, the full set of states of $\mathsf{X}_{\frac{1}{2},\frac{1}{2},x,y,z}(4)$ is

(3.13)

Likewise, the state $v_3$ in (3.6) is a link state in $\mathsf{X}_{\frac{3}{2},\frac{1}{2},x,y,z}(12)$.

The action $a \cdot v$ of the algebra is defined as usual with $v$ drawn inside $a$, and the point $\mathsf{c}$ moved to the outer perimeter. A closed loop is assigned a weight $\beta$ if it encircles neither of the marked points. If $k \neq \ell$, loops may not encircle any of the two marked points. If $k = \ell$ however, loops may encircle both $\mathsf{a}$ and $\mathsf{b}$ and are then assigned weight $\alpha_{\mathsf{ab}} = z + z^{-1}$.

If a defect crosses a dashed segment, it produces a twist factor as follows.[3] A defect attached to point $\mathsf{a}$ crossing the segment $\mathsf{ac}$ is assigned a weight $x$ if $\mathsf{a}$ is to its right and $x^{-1}$ if $\mathsf{a}$ to its left. A

---

[2]This choice of vector space ensures that the resulting module is cyclic, namely that all the states in $\mathsf{X}_{k,\ell,x,y,z}(N)$ can be produced by the action of the algebra on any link state with $2(k+\ell)$ defects and $\frac{N}{2} - k - \ell$ through-arcs, see Proposition 4.1.

[3]For $k = \ell$, one can define more generally a module $\mathsf{X}_{k,k,x,y,z,w}(N)$ with an extra parameter $w$, where the loops

defect attached to point $\mathsf{a}$ crossing the segment $\mathsf{bc}$ is assigned a weight $x^{-1}z$ if $\mathsf{b}$ lies to its right and $xz^{-1}$ if $\mathsf{b}$ lies to its left. A defect attached to point $\mathsf{b}$ crossing segment $\mathsf{bc}$ is assigned a weight $y$ if $\mathsf{b}$ lies to its right and $y^{-1}$ if $\mathsf{b}$ lies to its left. A defect attached to point $\mathsf{b}$ crossing the dashed segment $\mathsf{ac}$ is assigned a weight $(yz)^{-1}$ if $\mathsf{a}$ lies to its right and a weight $yz$ if $\mathsf{a}$ lies to its left. Finally, if a defect crosses more than one segment, then the resulting twist factor is the product of the corresponding weights. In other words, the allocation of twist factors for the defects attached to point $\mathsf{a}$ are identical to those illustrated in (3.12). The same rules for defects attached to the point $\mathsf{b}$ are obtained from (3.12) by interchanging the color of two marked points, and changing $x \to y$ and $z \to z^{-1}$.

If the action of the algebra connects together two defects attached to the same marked point, the result is set to zero. The action of the algebra may however reduce the number of defects if it connects defects tied to $\mathsf{a}$ with defects tied to $\mathsf{b}$. The resulting weight depends on two factors: (i) the number of times the defect crosses each of the two dashed segments, and (ii) the relative positions of the marked points $\mathsf{a}$ and $\mathsf{b}$. In general, the weight of a defect is given by $\mu^\delta x^{n_\mathsf{a}} y^{n_\mathsf{b}}$. The numbers $n_\mathsf{a}$ and $n_\mathsf{b}$ count the number of times the defect winds around the points $\mathsf{a}$ and $\mathsf{b}$, respectively. Following an observer traveling on the defect from $\mathsf{a}$ to $\mathsf{b}$, $n_\mathsf{a}$ is positive if the observer crosses the dashed line $\mathsf{ac}$ with $\mathsf{a}$ to its right, and negative if $\mathsf{a}$ lies to its left. Likewise, following the observer traveling from $\mathsf{b}$ to $\mathsf{a}$, $n_\mathsf{b}$ is positive if the observer crosses the dashed line $\mathsf{bc}$ with $\mathsf{b}$ to its right. Moreover, we have $\delta = 0$ if the segment $\mathsf{ac}$ is to the left of $\mathsf{bc}$ and $\delta = 1$ if it lies to its right. In other words, there is an extra multiplicative weight $\mu$ if $\mathsf{ac}$ lies to the left of $\mathsf{bc}$. The value of $\mu$ is set to

$$\mu = \frac{yz}{x}. \tag{3.14}$$

This choice ensures that the definition of this action of $\mathcal{E}\mathsf{PTL}_N(\beta)$ on $\mathsf{X}_{k,\ell,x,y,z}(N)$ yields a well-defined representation. For instance, it ensures that the following two computations yield identical results:

$$e_1 e_3 \cdot \left(\begin{smallmatrix}3 & & 2\\ & & \\ 4 & & 1\end{smallmatrix}\right) = y^{-1} e_1 \cdot \left(\begin{smallmatrix}3 & & 2\\ & & \\ 4 & & 1\end{smallmatrix}\right) = \mu y^{-1} \left(\begin{smallmatrix}3 & & 2\\ & & \\ 4 & & 1\end{smallmatrix}\right), \tag{3.15a}$$

$$e_3 e_1 \cdot \left(\begin{smallmatrix}3 & & 2\\ & & \\ 4 & & 1\end{smallmatrix}\right) = x^{-1} z\, e_3 \cdot \left(\begin{smallmatrix}3 & & 2\\ & & \\ 4 & & 1\end{smallmatrix}\right) = x^{-1} z \left(\begin{smallmatrix}3 & & 2\\ & & \\ 4 & & 1\end{smallmatrix}\right). \tag{3.15b}$$

Let us give more examples that illustrate the allocation of weights to the connection of two defects:

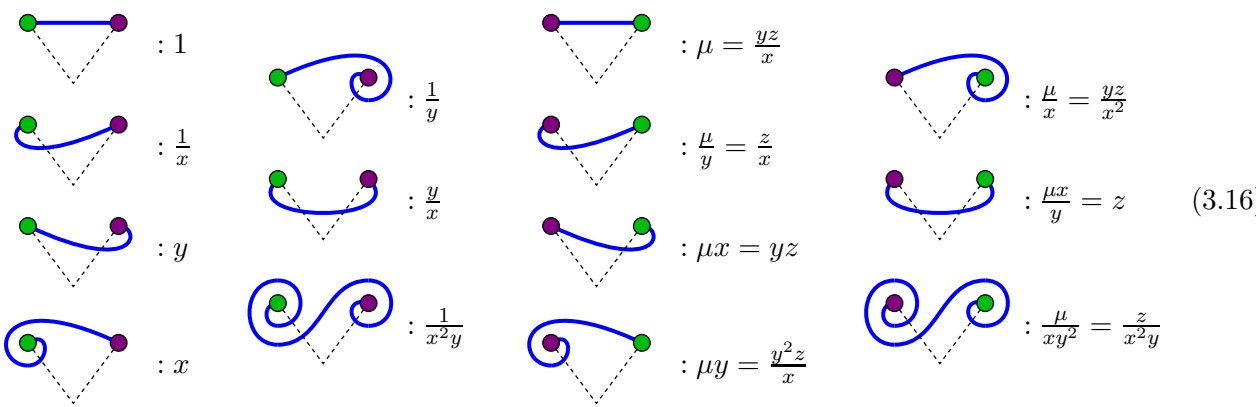

$$\tag{3.16}$$

encircling $\mathsf{a}$ and $\mathsf{b}$ are allocated the weight $\alpha_\mathsf{ab} = z + z^{-1}$, whereas the twist of defects are assigned twist factors $x^{\pm 1}$, $(w/x)^{\pm 1}$, $y^{\pm 1}$ and $(yw)^{\mp 1}$. The resulting representation can be shown to be isomorphic to $\mathsf{X}_{k,k,x,y,z}(N)$ for generic values of $w$. For simplicity, we choose not to discuss this generalisation further here.

We illustrate the action of $\mathcal{E}\mathsf{PTL}_N(\beta)$ on $\mathsf{X}_{k,\ell,x,y,z}(N)$ with two more examples for $N = 4$:

$$e_4 \cdot \underset{\substack{4 \qquad 1}}{\overset{\substack{3 \qquad 2}}{\bigcirc}} = \frac{\mu x}{y} \underset{\substack{4 \qquad 1 \\ =z}}{\overset{\substack{3 \qquad 2}}{\bigcirc}}, \qquad e_3 \cdot \underset{\substack{4 \qquad 1}}{\overset{\substack{3 \qquad 2}}{\bigcirc}} = z^{-1} \underset{\substack{4 \qquad 1}}{\overset{\substack{3 \qquad 2}}{\bigcirc}}. \qquad (3.17)$$

**Status as representations.** For each of these three families of modules, the action of $\mathcal{E}\mathsf{PTL}_N(\beta)$ defined above on the link states is extended linearly to all linear combinations of link states. Moreover, we claim that the diagrammatic rules described above define well-defined representations. This is in fact not obvious. We do not give here a complete proof of this claim, and only describe a sketch of a proof. The goal is to verify that $a_1 \cdot (a_2 \cdot v) = (a_1 a_2) \cdot v$ for all $a_1, a_2 \in \mathcal{E}\mathsf{PTL}_N(\beta)$ and $v \in \mathsf{X}_{k,\ell,x,y,z}(N)$. Equivalently, one could check that the action of $\mathcal{E}\mathsf{PTL}_N(\beta)$ on $\mathsf{X}_{k,\ell,x,y,z}(N)$ is consistent with the defining relations (2.2), but the first formulation turns out to be easier. Indeed, the diagrammatic definition for the product $a_1 a_2$ of two connectivities of $\mathcal{E}\mathsf{PTL}_N(\beta)$ consist in two rules: (i) each contractible loop is removed and replaced by a weight $\beta$, and (ii) two connectivities are identified if they only differ by deformations of the loop segments that do not change the connectivity of the $2N$ nodes. The diagrammatic action of $\mathcal{E}\mathsf{PTL}_N(\beta)$ on $\mathsf{X}_{k,\ell,x,y,z}(N)$ respects these two rules. In particular, the second rule is satisfied because $a \cdot v$ is uniquely fixed in terms of the connectivity of the nodes, and does not depend on the precise paths taken by the loop segments. This is obvious if either $k$ or $\ell$ is zero.

For $k, \ell > 0$, the complex rule for the connection of defects makes this less obvious. It is in fact the specific choice (3.14) of $\mu$ that ensures that $a \cdot v$ is uniquely fixed and that the action of $\mathcal{E}\mathsf{PTL}_N(\beta)$ on the link states as defined above really produces a representation. Indeed, in many cases, computing the weight of a defect can be done in two ways. This is apparent for instance in the example (3.15), wherein the two ways of computing this weight can be summarised as

$$\overset{\bullet \quad \bullet}{\smile} = y^{-1} \overset{\bullet \quad \bullet}{\triangledown} = y^{-1} \cdot \mu \qquad \text{and} \qquad \overset{\bullet \quad \bullet}{\smile} = zx^{-1} \overset{\bullet \quad \bullet}{\triangledown} = zx^{-1} \cdot 1. \quad (3.18)$$

These indeed give identical results for $\mu = \frac{yz}{x}$. In contrast, for some diagrams, there is a unique way to compute the weight. This is the case for instance for

$$e_1 e_3 e_2 \cdot \underset{\substack{4 \qquad 1}}{\overset{\substack{3 \qquad 2}}{\bigcirc}} = \mu y \underset{\substack{4 \qquad 1}}{\overset{\substack{3 \qquad 2}}{\bigcirc}}, \qquad (3.19)$$

corresponding to the diagram $\overset{\curvearrowright}{\phantom{a}}$. The generator $e_2$ cannot be commuted to the left in any way, so its action on the link state is always resolved first. In other words, the weight of this diagram can only be computed by unwinding the defect around point b first, producing the twist factor $y$. Similarly, each example in (3.16) can be resolved in either one or two ways, and the result is always unique.

**Properties of the representations** Furthermore, we note that on the modules $\mathsf{X}_{k,\ell,x,y,z}(N)$, we have the identity

$$\forall v \in \mathsf{X}_{k,\ell,x,y,z}(N), \qquad \Omega^N \cdot v = z^{2(k-\ell)} v, \qquad (3.20)$$

and for the special case $k = \ell = 0$,

$$\forall v \in \mathsf{X}_{0,0,x,y,z}(N), \qquad E\,\Omega^{\pm 1} E \cdot v = (z + z^{-1})\,E \cdot v, \qquad (3.21)$$

where $E = e_2 e_4 \cdots e_N$.

### 3.3 Gluing, induction and fusion

We consider two algebras $\mathcal{E}\mathsf{PTL}_{N_a}(\beta)$ and $\mathcal{E}\mathsf{PTL}_{N_b}(\beta)$ and two standard modules $\mathsf{W}_{k,x}(N_a)$ and $\mathsf{W}_{\ell,y}(N_b)$ over these algebras, with $N_a, N_b \in \mathbb{N}$. The gluing of $u_a \in \mathsf{W}_{k,x}(N_a)$ and $u_b \in \mathsf{W}_{\ell,y}(N_b)$, denoted $g_z(u_a, u_b)$, is a linear map from $\mathsf{W}_{k,x}(N_a) \otimes \mathsf{W}_{\ell,y}(N_b)$, seen as a module over $\mathsf{TL}_{N_a}(\beta) \otimes \mathsf{TL}_{N_b}(\beta)$, to $\mathsf{X}_{k,\ell,x,y,z}(N)$, with $N = N_a + N_b$. The output is a diagram where both $u_a$ and $u_b$ are drawn inside a disc with $N_a + N_b$ nodes on its perimeter. The states $u_a$ and $u_b$ are associated to the marked point $a$ and $b$ in $\mathsf{X}_{k,\ell,x,y,z}(N)$, respectively, and their nodes are attached to the nodes with labels $1, \ldots, N_a$, and $N_a + 1, \ldots, N$. We illustrate this with two examples with $(N_a, N_b) = (4, 4)$ and $(5, 6)$:

$$(3.22\text{a})$$

$$(3.22\text{b})$$

By definition, the image of the map $g_z$ has the structure of $\mathsf{X}_{k,\ell,x,y,z}(N)$. In other words, acting with elements of $\mathcal{E}\mathsf{PTL}_N(\beta)$ on $g_z(u_a, u_b)$ assigns weights $\alpha_{ab} = z + z^{-1}$ to non-contractible loops, or produces twist factors that depend on $z$ for defects that wind around the two marked points. Moreover, the gluing map preserves the action of $\mathsf{TL}_{N_a}(\beta) \otimes \mathsf{TL}_{N_b}(\beta)$ on $\mathsf{W}_{k,x}(N_a) \otimes \mathsf{W}_{\ell,y}(N_b)$. Indeed, for any $u_a \in \mathsf{W}_{k,x}(N_a)$ and $u_b \in \mathsf{W}_{\ell,y}(N_b)$, the generators $e_j$ satisfy the relations

$$e_j \cdot g_z(u_a, u_b) = g_z(e_j \cdot u_a, u_b) \qquad \text{for } 1 \leqslant j \leqslant N_a - 1, \tag{3.23a}$$

$$e_{N_a+j} \cdot g_z(u_a, u_b) = g_z(u_a, e_j \cdot u_b) \qquad \text{for } 1 \leqslant j \leqslant N_b - 1. \tag{3.23b}$$

In contrast, the generators $\Omega^{\pm 1}$, $e_{N_a}$ and $e_N$ do not commute with $g_z$, namely their action can in general not be simplified in terms of the standard action of $\mathsf{TL}_{N_a}(\beta) \otimes \mathsf{TL}_{N_b}(\beta)$ in $\mathsf{W}_{k,x}(N_a) \otimes \mathsf{W}_{\ell,y}(N_b)$.

Recalling from Section 2.5 the definition of the states $v_k(\ell)$, we note that the gluing $g_z\big(v_k(N_a/2), v_\ell(N_b/2)\big)$ produces a link state with a total number of $2k + 2\ell$ defects and with the maximal number $N/2 - k - \ell$ of through-arcs. Proposition 4.1 below will show that the module $\mathsf{X}_{k,\ell,x,y,z}(N)$ is generated from the repeated action of $\mathcal{E}\mathsf{PTL}_N(\beta)$ on such states, for $q, z$ generic. As a result, we can write

$$\mathsf{X}_{k,\ell,x,y,z}(N) = \mathcal{E}\mathsf{PTL}_N(\beta) \cdot g_z\big(\mathsf{W}_{k,x}(N_a), \mathsf{W}_{\ell,y}(N_b)\big). \tag{3.24}$$

Crucially, although this relation is similar to (3.1), it should not be mistaken for a definition of $\mathsf{X}_{k,\ell,x,y,z}(N)$, as the gluing map $g_z$ appearing on the right-hand side is by definition a map onto $\mathsf{X}_{k,\ell,x,y,z}(N)$, and therefore assumes that the definition of this module is known. The statement of (3.24) is instead that the image of $\mathsf{W}_{k,x}(N_a) \otimes \mathsf{W}_{\ell,y}(N_b)$ under the action of $g_z$ produces the entire module $\mathsf{X}_{k,\ell,x,y,z}(N_a + N_b)$ when acted on by $\mathcal{E}\mathsf{PTL}_N(\beta)$.

This construction is somewhat analogous to that for $\mathsf{TL}_N(\beta)$ described in Section 3.1, with the gluing map playing a similar role to the selective tensor product in (3.1). This justifies our claim that $\mathsf{X}_{k,\ell,x,y,z}(N)$ should be interpreted as the fusion of two standard modules:

$$\mathsf{W}_{k,x}(N_a) \times_z \mathsf{W}_{\ell,y}(N_b) = \mathsf{X}_{k,\ell,x,y,z}(N). \tag{3.25}$$

In this equation, we use the notation $\times_z$ to indicate that the parameter $z$ does not solely belong to $\mathsf{W}_{k,x}(N_{\mathsf{a}})$ or $\mathsf{W}_{\ell,y}(N_{\mathsf{b}})$, but instead describes the interaction between the two modules in their fusion, either through the weights given to loops encircling both marked points $\mathsf{a}$ and $\mathsf{b}$ or by its coupling to the winding of defects around these points.

We note however that an entirely algebraic definition of this fusion $\mathsf{W}_{k,x}(N_{\mathsf{a}}) \times_z \mathsf{W}_{\ell,y}(N_{\mathsf{b}})$, similar to (3.1), is thus currently lacking. Such a definition should include in particular extra quotient relations (3.20) and (3.21), as well as other relations accounting for the complex diagrammatic rule for the connection of defects described in Section 3.2.

# 4 The structure of the modules $\mathsf{X}_{k,\ell,x,y,z}(N)$

In this section, we investigate the decomposition of the modules $\mathsf{X}_{k,\ell,x,y,z}(N)$. The main result is the following theorem.

THEOREM 1 *For generic values of $q$ and $z$, we have the module decomposition*

$$\mathsf{X}_{k,\ell,x,y,z}(N) \simeq \mathsf{W}_{k-\ell,z}(N) \oplus \bigoplus_{m=k-\ell+1}^{N/2} \bigoplus_{n=0}^{2m-1} \mathsf{W}_{m,z^{(k-\ell)/m}\mathsf{e}^{\mathsf{i}\pi n/m}}(N), \qquad for \ k \geqslant \ell. \qquad (4.1)$$

*Moreover, we have $\mathsf{X}_{k,\ell,x,y,z}(N) \simeq \mathsf{X}_{\ell,k,y,x,z^{-1}}(N)$, which fixes the module decomposition of $\mathsf{X}_{k,\ell,x,y,z}(N)$ for $k < \ell$.*

This theorem holds for all $x, y \in \mathbb{C}^\times$. We also note that the explicit decompositions for $q, z$ generic in fact do not depend on the values of $x$ and $y$. We discuss this further in Section 6.

## 4.1 Quotients and dimensions

In this subsection, we describe the tower structure of the module $\mathsf{X}_{k,\ell,x,y,z}(N)$ and use it to compute its dimension.

**Depth and quotient modules.** For a given link state in $\mathsf{X}_{k,\ell,x,y,z}(N)$, we define its *depth* $p$ as the number of its through-arcs. The possible values of $p$ are $0, 1, \ldots, \frac{N}{2} - k - \ell$. In the example (3.6), $v_1$ and $v_2$ have depth $p = 2$ whereas $v_3$ has depth $p = 1$. Moreover, in (3.7), (3.9) and (3.13), the states are drawn with the depth constant on each line.

The action of the algebra $\mathcal{E}\mathsf{PTL}_N(\beta)$ on a basis state of $\mathsf{X}_{k,\ell,x,y,z}(N)$ of depth $p$ only produces states whose depth is smaller or equal to $p$. We denote by $\mathsf{X}^{(p)}_{k,\ell,x,y,z}(N)$ the submodule of $\mathsf{X}_{k,\ell,x,y,z}(N)$ spanned by link states of depth at most $p$. These submodules define a filtration of $\mathsf{X}_{k,\ell,x,y,z}(N)$:

$$\mathsf{X}^{(0)}_{k,\ell,x,y,z}(N) \subset \mathsf{X}^{(1)}_{k,\ell,x,y,z}(N) \subset \cdots \subset \mathsf{X}^{(N/2-k-\ell)}_{k,\ell,x,y,z}(N) = \mathsf{X}_{k,\ell,x,y,z}(N). \qquad (4.2)$$

For $p \geqslant 1$, we introduce an equivalence relation for two elements $v_1, v_2$ of $\mathsf{X}^{(p)}_{k,\ell,x,y,z}(N)$:

$$v_1 \equiv v_2 \ [[p-1]] \qquad \text{iff} \qquad v_1 - v_2 \in \mathsf{X}^{(p-1)}_{k,\ell,x,y,z}(N). \qquad (4.3)$$

The corresponding quotient modules, made of the equivalence classes under this relation, are denoted

$$\mathsf{M}^{(p)}_{k,\ell,x,y,z}(N) = \mathsf{X}^{(p)}_{k,\ell,x,y,z}(N) \Big/ \mathsf{X}^{(p-1)}_{k,\ell,x,y,z}(N), \qquad p = 0, 1, \ldots, \tfrac{N}{2} - k - \ell, \qquad (4.4)$$

where we use the conventions $\mathsf{X}^{(-1)}_{k,\ell,x,y,z}(N) = 0$ and $\mathsf{M}^{(0)}_{k,\ell,x,y,z}(N) = \mathsf{X}^{(0)}_{k,\ell,x,y,z}(N)$.

For $k, \ell > 0$, the submodule $\mathsf{X}^{(0)}_{k,\ell,x,y,z}(N)$ decomposes further. In this case, certain link states of $\mathsf{X}^{(0)}_{k,\ell,x,y,z}(N)$ have defects attached to both points $\mathsf{a}$ and $\mathsf{b}$. The action of $\mathcal{EPTL}_N(\beta)$ on these states cannot increase the total number of defects, but can decrease it. We define the submodule $\mathsf{X}^{(0,\,r)}_{k,\ell,x,y,z}(N)$ spanned by link states that have at most $2r$ total defects, with $r = |k - \ell|, |k - \ell| + 1, \ldots, k + \ell$. These define a filtration for $\mathsf{X}^{(0)}_{k,\ell,x,y,z}(N)$:

$$\mathsf{X}^{(0,\,|k-\ell|)}_{k,\ell,x,y,z}(N) \subset \mathsf{X}^{(0,\,|k-\ell|+1)}_{k,\ell,x,y,z}(N) \subset \cdots \subset \mathsf{X}^{(0,\,k+\ell)}_{k,\ell,x,y,z}(N) = \mathsf{X}^{(0)}_{k,\ell,x,y,z}(N). \tag{4.5}$$

The corresponding quotient modules are defined as

$$\mathsf{N}^{(r)}_{k,\ell,x,y,z}(N) = \mathsf{X}^{(0,\,r)}_{k,\ell,x,y,z}(N)\Big/\mathsf{X}^{(0,\,r-1)}_{k,\ell,x,y,z}(N), \qquad r = |k - \ell|, |k - \ell| + 1, \ldots, k + \ell, \tag{4.6}$$

with the conventions $\mathsf{X}^{(0,\,|k-\ell|-1)}_{k,\ell,x,y,z}(N) = 0$ and $\mathsf{N}^{(|k-\ell|)}_{k,\ell,x,y,z}(N) = \mathsf{X}^{(0,\,|k-\ell|)}_{k,\ell,x,y,z}(N)$.

**Dimension counting.** The dimension of $\mathsf{X}_{k,\ell,x,y,z}(N)$ can be computed by studying the dimensions of the quotient modules $\mathsf{M}^{(p)}_{k,\ell,x,y,z}(N)$ and $\mathsf{N}^{(r)}_{k,\ell,x,y,z}(N)$. We first discuss the case $\mathsf{M}^{(p)}_{k,\ell,x,y,z}(N)$ for $p > 0$. Given a link state in $v \in \mathsf{M}^{(p)}_{k,\ell,x,y,z}(N)$, namely a link state of depth $p$, we draw a thick bridge between the points $\mathsf{a}$ and $\mathsf{b}$. There are $2(k + \ell + p)$ nodes from the perimeter that are attached to this bridge. We map the resulting diagram to a link state in $\mathsf{W}_{k+\ell+p,z}(N)$, with the bridge playing the role of the unique marked point of the standard module and all loop segments attached to it becoming defects. For instance, for the three states in (3.6), this construction yields

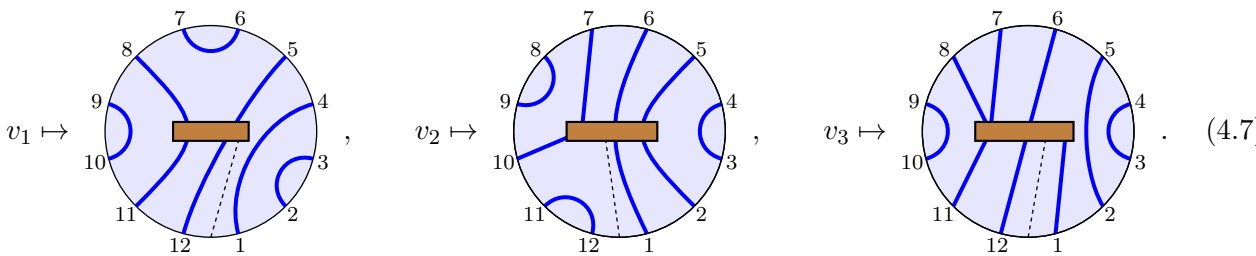

This map is not one-to-one. Instead, for each state in $\mathsf{W}_{k+\ell+p,z}(N)$, there are precisely $2(k + \ell + p)$ states in the pre-image. The nodes attached to the bridge partitions into four adjacent parts: $2k$ defects attached to the point $\mathsf{a}$, $p$ nodes attached to through-arcs, $2\ell$ defects attached to the point $\mathsf{b}$, and finally $p$ more nodes attached to through-arcs. The $2(k + \ell + p)$ states in the pre-image correspond to the possible rotations of this formation. As a result, we have

$$\dim \mathsf{M}^{(p)}_{k,\ell,x,y,z}(N) = 2(k + \ell + p) \dim \mathsf{W}_{k+\ell+p,z}(N), \qquad p = 1, \ldots, \tfrac{N}{2} - k - \ell. \tag{4.8}$$

For $p = 0$, we study separately each quotient module $\mathsf{N}^{(0,\,r)}_{k,\ell,x,y,z}(N)$. For a given state $v$ in this module, namely a link state of zero depth with a total of $2r$ defects, we draw the bridge between the points $\mathsf{a}$ and $\mathsf{b}$ as above, and observe that the $2r$ defects are attached to the bridge. We map the resulting diagram to the corresponding link state in $\mathsf{W}_{r,z}(N)$, with the bridge replaced by the marked point. This map is one-to-one for $r = |k - \ell|$, but not for $r > |k - \ell|$. In this last case, there are instead $2r$ states in the pre-image, corresponding to the $2r$ possible rotations of the defects. This yields

$$\dim \mathsf{N}^{(r)}_{k,\ell,x,y,z} = \begin{cases} \dim \mathsf{W}_{|k-\ell|,z}(N) & r = |k - \ell|, \\ 2r \dim \mathsf{W}_{r,z}(N) & r > |k - \ell|. \end{cases} \tag{4.9}$$

Altogether, we find

$$\dim \mathsf{X}_{k,\ell,x,y,z}(N) = \sum_{r=|k-\ell|}^{k+\ell} \dim \mathsf{N}_{k,\ell,x,y,z}^{(r)}(N) + \sum_{p=1}^{N/2-k-\ell} \dim \mathsf{M}_{k,\ell,x,y,z}^{(p)}(N)$$

$$= \dim \mathsf{W}_{|k-\ell|,z}(N) + \sum_{r=|k-\ell|+1}^{k+\ell} 2r \, \dim \mathsf{W}_{r,z}(N) + \sum_{p=1}^{N/2-k-\ell} 2(k+\ell+p) \, \dim \mathsf{W}_{k+\ell+p,z}(N)$$

$$= \dim \mathsf{W}_{|k-\ell|,z}(N) + \sum_{m=|k-\ell|+1}^{N/2} 2m \, \dim \mathsf{W}_{m,z}(N). \tag{4.10}$$

We recall that the dimension of $\mathsf{W}_{k,z}(N)$, given in (2.13), does not depend on $z$. Moreover, two modules $\mathsf{W}_{k,y}(N)$ and $\mathsf{W}_{k,z}(N)$ are in general non-isomorphic even if they have the same dimension. Thus the above formulas for the dimension do not fix the decomposition of $\dim \mathsf{X}_{k,\ell,x,y,z}(N)$. With this in mind, the formula (4.10) for the dimension of $\mathsf{X}_{k,\ell,x,y,z}(N)$ is nonetheless reminiscent of the result of Theorem 1. A simple inductive argument allows us to simplify the formula for the dimension to

$$\dim \mathsf{X}_{k,\ell,x,y,z}(N) = \left(\tfrac{N}{2} - |k-\ell| + 1\right) \binom{N}{\frac{N}{2} - |k-\ell|}. \tag{4.11}$$

We note that the dimension of $\mathsf{X}_{k,\ell,x,y,z}(N)$ depends on $k$ and $\ell$ only through the difference $|k-\ell|$.

## 4.2 Decompositions for $q, z$ generic

In this section, we present a proof of Theorem 1. We achieve this by constructing two families of non-zero homomorphisms

$$\text{(i)} \quad \mathsf{W}_{m,\omega}(N) \to \mathsf{M}_{k,\ell,x,y,z}^{(p)}(N) \quad \text{with } m = k + \ell + p, \qquad \text{(ii)} \quad \mathsf{W}_{r,\omega}(N) \to \mathsf{N}_{k,\ell,x,y,z}^{(r)}(N), \tag{4.12}$$

for certain special values of $\omega$ given below. First, we describe the map of type (i) for a system size $N = 2m$. Second, we give the construction of the map of type (i) for any value of $N$, using the insertion algorithm. Third, we discuss the homomorphisms of type (ii).

**Homomorphisms of type (i) for $N = 2m$.** Throughout this discussion, we set $m = k + \ell + p$. Let us denote as $v_{k,\ell}(m)$ the unique the link state in $\mathsf{X}_{k,\ell,x,y,z}(2m)$ that has depth $p$ and its nodes with labels $1, \ldots, 2k$ attached to the point $\mathsf{a}$ by defects. For instance for $\mathsf{X}_{\frac{3}{2},\frac{1}{2},x,y,z}(12)$, this state is

$$v_{\frac{3}{2},\frac{1}{2}}(6) = \qquad . \tag{4.13}$$

The vector space of $\mathsf{M}_{k,\ell,x,y,z}^{(p)}(2m)$ is $2m$-dimensional and is spanned by $v_{k,\ell}(m)$ and its rotations $\Omega \cdot v_{k,\ell}(m), \ldots, \Omega^{2m-1} \cdot v_{k,\ell}(m)$. The action of the generators $e_j$ on $v_{k,\ell}(m)$ is always proportional to a link state with depth strictly less than $p$. We write this as

$$e_j \cdot v_{k,\ell}(m) \equiv 0 \; [[p-1]], \qquad j = 1, \ldots, 2m. \tag{4.14}$$

We introduce the states

$$w_{k,\ell,x,y,z}(m,n) = \frac{1}{2m} \sum_{j=0}^{2m-1} \omega_n^{-j} \, \Omega^j \cdot v_{k,\ell}(m), \qquad \omega_n = z^{(k-\ell)/m} \, \mathrm{e}^{\mathrm{i}\pi n/m}, \qquad n = 0, \ldots, 2m-1. \quad (4.15)$$

These are in fact obtained by acting on $v_{k,\ell}(m)$ with the projectors $\Pi_n$ (defined in (2.17)). By construction, they satisfy the relations

$$\Omega \cdot w_{k,\ell,x,y,z}(m,n) = \omega_n \, w_{k,\ell,x,y,z}(m,n), \qquad e_j \cdot w_{k,\ell,x,y,z}(m,n) \equiv 0 \ [[p-1]], \qquad j = 1, \ldots, 2m. \quad (4.16)$$

This implies that, as an element of $\mathsf{M}^{(p)}_{k,\ell,x,y,z}(2m)$, the state $w_{k,\ell,x,y,z}(m,n)$ spans a one-dimensional submodule of $\mathcal{E}\mathsf{PTL}_N(\beta)$, isomorphic to $\mathsf{W}_{m,\omega_n}(2m)$. Hence, the linear application that maps the unique state $v_m(m)$ of $\mathsf{W}_{m,\omega_n}(2m)$ to $w_{k,\ell,x,y,z}(m,n)$ is a homomorphism

$$\mathsf{W}_{m,\omega_n}(2m) \to \mathsf{M}^{(p)}_{k,\ell,x,y,z}(2m), \qquad m = k+\ell+p, \qquad \begin{cases} p \geqslant 1, \\ n = 0, \ldots, 2m-1. \end{cases} \quad (4.17)$$

**Homomorphisms of type (i) for $N > 2m$.** In this case, we build a family of homomorphisms

$$\mathsf{W}_{m,\omega_n}(N) \to \mathsf{M}^{(p)}_{k,\ell,x,y,z}(N), \qquad m = k+\ell+p, \qquad \begin{cases} p \geqslant 1, \\ n = 0, \ldots, 2m-1, \end{cases} \quad (4.18)$$

with $\omega_n = z^{(k-\ell)/m} \, \mathrm{e}^{\mathrm{i}\pi n/m}$ as above. The construction uses an insertion map similar to the one used in Section 2.5. For $u \in \mathsf{W}_{m,\omega_n}(N)$, $\Phi(u)$ is obtained from $u$ by selecting its $2m$ nodes attached to defects, erasing those defects, and instead connecting these nodes to the state $w_{k,\ell,x,y,z}(m,n)$. To illustrate, here is an example with a link state in $\mathsf{W}_{2,\omega_n}(12)$ mapped into $\mathsf{M}^{(1)}_{0,1,x,y,z}(12)$:



where the pink disc indicates the insertion of the state $w_{0,1,x,y,z}(2,n)$. As a second example, the four link states of $\mathsf{W}_{1,\omega_n}(4)$ depicted in (2.11b) are mapped into $\mathsf{M}^{(1)}_{0,0,x,y,z}$ as

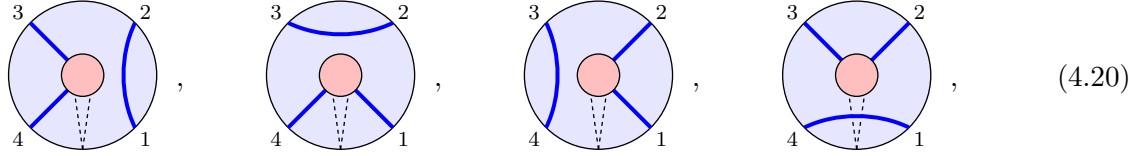

where the pink disc represents the state $w_{0,0,x,y,z}(1,n)$. The two above examples are of course very reminiscent of (2.29) and (2.30).

By construction, this homomorphism is non-zero. Indeed, the action of the generators $e_j$ yields a vanishing result in $\mathsf{M}^{(p)}_{k,\ell,x,y,z}(N)$ if it connects nodes attached to $w_{k,\ell,x,y,z}(m,n)$, as it then produces states with lesser depth. Otherwise, it permutes the nodes tied to $w_{k,\ell,x,y,z}(m,n)$, in precisely the same way as it does for the defects in $\mathsf{W}_{m,\omega_n}(N)$. Equivalently, the map $\Phi$ satisfies the homomorphism relation

$$\forall a \in \mathcal{E}\mathsf{PTL}_N(\beta), \qquad \forall v \in \mathsf{W}_{m,\omega_n}(N), \qquad a \cdot \Phi(v) = \Phi(a \cdot v). \quad (4.21)$$

From these properties, we deduce that the states in the image of $\Phi$ span a nonzero submodule of $\mathsf{M}^{(p)}_{k,\ell,x,y,z}(N)$. For $q$ and $z$ generic, the standard modules $\mathsf{W}_{m,\omega_n}(N)$ are all irreducible and are pairwise non-isomorphic. This implies that each map in (4.18) is injective. Therefore $\mathsf{M}^{(p)}_{k,\ell,x,y,z}(N)$ has a submodule isomorphic to $\bigoplus_n \mathsf{W}_{m,\omega_n}(N)$, wherein no two summands are isomorphic. Using the dimension counting (4.8), we observe that this direct summand exhausts the dimension of $\mathsf{M}^{(p)}_{k,\ell,x,y,z}(N)$, and conclude that

$$\mathsf{M}^{(p)}_{k,\ell,x,y,z}(N) \simeq \bigoplus_{n=0}^{2m-1} \mathsf{W}_{m,\omega_n}(N), \qquad \omega_n = z^{(k-\ell)/m}\mathsf{e}^{in\pi/m}, \qquad m = k + \ell + p, \qquad p \geqslant 1. \qquad (4.22)$$

**Homomorphisms of type (ii).** The link states of $\mathsf{M}^{(0)}_{k,\ell,x,y,z}(N) = \mathsf{X}^{(0)}_{k,\ell,x,y,z}(N)$ have no through-arcs. If $\ell = 0$, the vector space of $\mathsf{X}^{(0)}_{k,\ell,x,y,z}(N)$ is identical to that of $\mathsf{W}_{k,z}(N)$. Moreover, the action of $\mathcal{E}\mathrm{PTL}_N(\beta)$ acts identically in both modules: closing two defects attached to the point $\mathsf{a}$ gives a vanishing result, and any defect that winds around the cylinder always crosses both dashed segments and is then assigned the weights $z$ or $z^{-1}$. We conclude that

$$\mathsf{X}^{(0)}_{k,0,x,y,z}(N) \simeq \mathsf{W}_{k,z}(N). \qquad (4.23)$$

The same ideas apply to the case $k = 0$, resulting in

$$\mathsf{X}^{(0)}_{0,\ell,x,y,z}(N) \simeq \mathsf{W}_{\ell,z^{-1}}(N). \qquad (4.24)$$

For $k = \ell = 0$, both (4.23) and (4.24) are valid because $\mathsf{W}_{0,z}(N) \simeq \mathsf{W}_{0,z^{-1}}(N)$.

If $k$ and $\ell$ are both non-zero, the module $\mathsf{X}^{(0)}_{k,\ell,x,y,z}(N)$ decomposes as a direct sum, which we now investigate in terms of the quotient modules $\mathsf{N}^{(r)}_{k,\ell,x,y,z}(N)$. Repeating the argument that lead to (4.23) and (4.24), we find for the case $r = |k - \ell|$ the simple decomposition

$$\mathsf{N}^{(|k-\ell|)}_{k,\ell,x,y,z}(N) = \mathsf{X}^{(0,\,|k-\ell|)}_{k,\ell,x,y,z}(N) \simeq \mathsf{W}_{|k-\ell|,z^{\mathrm{sign}(k-\ell)}}(N). \qquad (4.25)$$

For $k = \ell$, $\mathrm{sign}(k - \ell)$ can be taken to be either $1$ or $-1$ because $\mathsf{W}_{0,z}(N) \simeq \mathsf{W}_{0,z^{-1}}(N)$. For $r > |k-\ell|$, we use the insertion algorithm to define maps

$$\mathsf{W}_{r,\omega_n}(N) \to \mathsf{N}^{(r)}_{k,\ell,x,y,z}(N), \qquad \omega_n = z^{(k-\ell)/r}\mathsf{e}^{in\pi/r}, \qquad n = 0, \dots, 2r - 1. \qquad (4.26)$$

For a given link state $u \in \mathsf{W}_{r,\omega_n}(N)$, we select its $2r$ nodes attached to defects, erase those defects, and connect these nodes to the state $w_{r_\mathsf{a},r_\mathsf{b},x,y,z}(r,n)$, with $r_\mathsf{a} = \frac{1}{2}(r + k - \ell)$ and $r_\mathsf{b} = \frac{1}{2}(r - k + \ell)$. The result $\Phi(u)$ is a linear combination of link states of $\mathsf{N}^{(r)}_{k,\ell,x,y,z}(N)$ that only differ in the way the defects of $u$ are distributed between the points $\mathsf{a}$ and $\mathsf{b}$. For instance, an example of the map $\mathsf{W}_{2,\omega_n}(12) \to \mathsf{N}^{(2)}_{3/2,1/2,x,y,z}(12)$ is obtained from (4.19) by modifying the state inside the disc on the right-hand side to $w_{3/2,1/2,x,y,z}(2,n)$. Likewise, the four states obtained from the map $\mathsf{W}_{1,\omega_n}(4) \to \mathsf{N}^{(1)}_{1/2,1/2,x,y,z}(4)$ are obtained from (4.20) by inserting the state $w_{1/2,1/2,x,y,z}(1,n)$ inside the pink disc.

We note that the two insertion maps described above for the homomorphisms of types (i) and (ii) are in fact defined in exactly the same way. The only difference is that the inserted state $w_{k,\ell,x,y,z}(m,n)$ is restricted to $k + \ell < m$ for the maps of type (i) and to $k + \ell = m$ for the maps of type (ii).

It is straightforward to show that the maps (4.26) are non-zero. For $q$ and $z$ generic, the standard modules $\mathsf{W}_{r,\omega_n}(N)$ are irreducible, and as a result the homomorphisms mapping them into $\mathsf{N}^{(r)}_{k,\ell,x,y,z}(N)$ are injective. The dimension of the direct sum of these inequivalent standard modules exhausts the dimension (4.9) of $\mathsf{N}^{(r)}_{k,\ell,x,y,z}(N)$, so we conclude that

$$\mathsf{N}^{(r)}_{k,\ell,x,y,z}(N) \simeq \bigoplus_{n=0}^{2r-1} \mathsf{W}_{r,\omega_n}(N), \qquad r = |k - \ell| + 1, \dots, |k + \ell|. \qquad (4.27)$$

**End of the proof of Theorem 1.** From the above construction of the homomorphisms, we have a complete list of the composition factors of $\mathsf{X}_{k,\ell,x,y,z}(N)$, namely

$$\bigcup_{p=1}^{\frac{N}{2}-k-\ell}\bigcup_{n=0}^{2(k+\ell+p)-1}\mathsf{W}_{k+\ell+p,\omega_n}(N), \qquad \mathsf{W}_{|k-\ell|,z^{\mathrm{sign}(k-\ell)}}(N), \qquad \bigcup_{r=|k-\ell|+1}^{k+\ell}\bigcup_{n=0}^{2r-1}\mathsf{W}_{r,\omega_n}(N), \qquad (4.28)$$

with $\omega_n$ defined in (4.18) for $\mathsf{W}_{k+\ell+p,\omega_n}(N)$ and in (4.26) for $\mathsf{W}_{r,\omega_n}(N)$.

For generic values of $q$ and $z$, these cannot form indecomposable yet reducible modules. Indeed, such modules can only arise between two standard modules if they have identical eigenvalues for the braid transfer matrices $\boldsymbol{F}$ and $\bar{\boldsymbol{F}}$. Using (2.18), we readily observe that no two modules in the list (4.28) share the same eigenvalue of $\boldsymbol{F}$. This therefore confirms that the items in (4.28) all appear as direct summands in the decomposition of $\mathsf{X}_{k,\ell,x,y,z}(N)$. This direct sum can be reorganised conveniently into (4.1), ending the proof of Theorem 1.

## 4.3 Further properties of $\mathsf{X}_{k,\ell,x,y,z}(N)$ for $q,z$ generic

In this section, we describe the explicit form of the homomorphisms into $\mathsf{X}_{k,\ell,x,y,z}(N)$, and show that these modules are generated from the action of the algebra on the link states with maximal depth.

**The homomorphisms $\mathsf{W}_{m,\omega_n}(N) \to \mathsf{X}_{k,\ell,x,y,z}(N)$.** In Section 4.2, we constructed two types of homomorphisms, from the standard modules $\mathsf{W}_{m,\omega_n}(N)$ onto the quotient modules $\mathsf{M}^{(p)}_{k,\ell,x,y,z}(N)$ or $\mathsf{N}^{(r)}_{k,\ell,x,y,z}(N)$. Having worked out the complete decomposition of $\mathsf{X}_{k,\ell,x,y,z}(N)$, we now describe the explicit homomorphisms into $\mathsf{X}_{k,\ell,x,y,z}(N)$ as follows. Let us focus on the case $k \geqslant \ell$. The decomposition of the nested submodules is

$$\mathsf{X}^{(p)}_{k,\ell,x,y,z}(N) \simeq \mathsf{W}_{k-\ell,z}(N) \oplus \bigoplus_{m=k-\ell+1}^{k+\ell+p}\bigoplus_{n=0}^{2m-1}\mathsf{W}_{m,z^{(k-\ell)/m}\mathsf{e}^{\mathrm{i}\pi n/m}}(N), \qquad 1 \leqslant p \leqslant \tfrac{N}{2}-k-\ell, \qquad (4.29\mathrm{a})$$

$$\mathsf{X}^{(0,r)}_{k,\ell,x,y,z}(N) \simeq \mathsf{W}_{k-\ell,z}(N) \oplus \bigoplus_{m=k-\ell+1}^{r}\bigoplus_{n=0}^{2m-1}\mathsf{W}_{m,z^{(k-\ell)/m}\mathsf{e}^{\mathrm{i}\pi n/m}}(N), \qquad k-\ell+1 \leqslant r \leqslant k+\ell, \quad (4.29\mathrm{b})$$

$$\mathsf{X}^{(0,k-\ell)}_{k,\ell,x,y,z}(N) \simeq \mathsf{W}_{k-\ell,z}(N). \qquad (4.29\mathrm{c})$$

We denote by

$$f_0(z) = q^{k-\ell}z + q^{\ell-k}z^{-1}, \qquad f_{m,n} = q^m\omega_n + q^{-m}\omega_n^{-1}, \qquad (4.30)$$

the eigenvalues of $\boldsymbol{F}$ for $\mathsf{W}_{k-\ell,z}(N)$ and $\mathsf{W}_{m,\omega_n}(N)$, respectively, with $\omega_n = z^{(k-\ell)/m}\mathsf{e}^{\mathrm{i}\pi n/m}$. From the above results, the spectrum of $\boldsymbol{F}$ on $\mathsf{X}^{(p)}_{k,\ell,x,y,z}(N)$ and $\mathsf{X}^{(0,r)}_{k,\ell,x,y,z}(N)$ is

$$\mathsf{X}^{(p)}_{k,\ell,x,y,z}(N): \quad \{f_0(z)\} \;\cup\; \{f_{m,n} \mid m = k-\ell+1,\ldots,k+\ell+p,\ n = 0,1,\ldots 2m-1\}, \qquad (4.31\mathrm{a})$$

$$\mathsf{X}^{(0,r)}_{k,\ell,x,y,z}(N): \quad \{f_0(z)\} \;\cup\; \{f_{m,n} \mid m = k-\ell+1,\ldots,r,\ n = 0,1,\ldots 2m-1\}. \qquad (4.31\mathrm{b})$$

The homomorphisms into $\mathsf{X}_{k,\ell,x,y,z}(N)$ can be presented in a unified way as $\mathsf{W}_{m,\omega_n}(N) \to \mathsf{X}_{k,\ell,x,y,z}(N)$, with $m = k+\ell+p$ for homomorphisms of type (i) and $m = r$ for homomorphisms of type (ii). The minimal value of $m$ is $k-\ell$ and corresponds to a homomorphism of type (ii). In this case, the homomorphism is trivial: it simply replaces the single marked point of the states in $\mathsf{W}_{k-\ell,z}(N)$ by two adjacent marked points. The maximal value of $m$ is $N/2$ and corresponds to a homomorphism of

type (i). In this case, the image of the homomorphism $\mathsf{W}_{m,\omega_n}(N) \to \mathsf{X}_{k,\ell,x,y,z}(N)$ is spanned by the single eigenvector of $\boldsymbol{F}$ in $\mathsf{X}_{k,\ell,x,y,z}^{(p)}(2m)$ of eigenvalue $f_{m,n}$. This eigenvector, denoted $\widehat{w}_{k,\ell,x,y,z}(m,n)$, is written as

$$\widehat{w}_{k,\ell,x,y,z}(m,n) = Q_{m,n} \cdot v_{k,\ell}(m), \qquad Q_{m,n} = \frac{\boldsymbol{F} - f_0}{f_{m,n} - f_0} \prod_{\substack{m'=k-\ell+1}}^{m} \prod_{\substack{n'=0 \\ (m',n')\neq(m,n)}}^{2m'-1} \frac{\boldsymbol{F} - f_{m',n'}}{f_{m,n} - f_{m',n'}}, \qquad (4.32)$$

where we recall that $v_{k,\ell}(m)$ is defined in Section 4.2. The operator $Q_{m,n}$ is a projector on the eigenspace of $\boldsymbol{F}$ of eigenvalue $f_{m,n}$. Moreover, $\widehat{w}_{k,\ell,x,y,z}(m,n)$ is clearly non-zero in its corresponding quotient module, so it is also non-zero in $\mathsf{X}_{k,\ell,x,y,z}^{(p)}(2m)$.

For $k - \ell < m < \frac{N}{2}$, the homomorphisms $\mathsf{W}_{m,\omega_n}(N) \to \mathsf{X}_{k,\ell,x,y,z}(N)$ are constructed using the insertion algorithm. For each link state of $\mathsf{W}_{m,\omega_n}(N)$, its $2m$ defects are erased and replaced with the state $\widehat{w}_{k,\ell,x,y,z}(m,n)$. The projectors $Q_{m,n}$ thus allow us to give an explicit form for the homomorphisms into $\mathsf{X}_{k,\ell,x,y,z}(N)$.

**Cyclicity of the modules $\mathsf{X}_{k,\ell,x,y,z}(N)$.** A module is cyclic if it can be generated by the action of the algebra on a single state. While it would be useful to address the question of the cyclicity of $\mathsf{X}_{k,\ell,x,y,z}(N)$ for all $q, z \in \mathbb{C}^\times$, the proposition below instead focuses on the generic values, showing that $\mathsf{X}_{k,\ell,x,y,z}(N)$ can be generated by the action of $\mathcal{E}\mathsf{PTL}_N(\beta)$ on any state of maximal depth $p = \frac{N}{2} - k - \ell$.

PROPOSITION 4.1 *For $k + \ell > 0$ and generic $q, z$, the repeated action of the algebra $\mathcal{E}\mathsf{PTL}_N(\beta)$ on the state $v_{k,\ell}(N/2)$ generates the full module $\mathsf{X}_{k,\ell,x,y,z}(N)$. For $k = \ell = 0$, the same result holds if either $\alpha_{\mathsf{a}} \neq 0$ or $\alpha_{\mathsf{b}} \neq 0$.*

PROOF. Let us recall that $v_{k,\ell}(m)$ is defined at the beginning of Section 4.2. First, we note that with the action of $\mathcal{E}\mathsf{PTL}_N(\beta)$ on $v_{k,\ell}(\frac{N}{2})$, we can produce link states with arbitrary depths, by acting iteratively with operators of the form

$$E_{ij} = e_{j-1}e_{j-2}\dots e_i, \qquad j \geqslant i+1. \qquad (4.33)$$

For instance, for $k + \ell > 0$, we define the sequence of states

$$u_0 = v_{k,\ell}(\tfrac{N}{2}), \qquad u_1 = E_{2k,N} \cdot u_0, \qquad u_2 = E_{2k-1,N-1} \cdot u_1, \qquad \dots \qquad (4.34)$$

From the graphical rules defining $\mathsf{X}_{k,\ell,x,y,z}(N)$, we see that $u_0, u_1, u_2$ are link states of depth $p_0, p_0 - 1, p_0 - 2$, with $p_0 = \frac{N}{2} - k - \ell$. The process can be iterated, and through the action of the operators $E_{ij}$, one generates link states of any depth. Similarly, in the zero-depth sector, well-chosen iterations of the $E_{ij}$ reduce the number of defects by steps of two, producing a sequence of link states with total number of defects $2r$, with $r = k + \ell, k + \ell - 1, \dots, |k - \ell|$.

For $k = \ell = 0$, the link state $v_{0,0}(\frac{N}{2})$ has depth $p_0 = \frac{N}{2}$. The only way to produce a state of depth $\frac{N}{2} - 1$ by acting on $v_{0,0}(\frac{N}{2})$ is to form a closed loop around the point $\mathsf{a}$ or $\mathsf{b}$. If $\alpha_{\mathsf{b}} \neq 0$, then

$$\begin{aligned}
u_0 &= v_{0,0}(\tfrac{N}{2}), \\
u_1 &= \alpha_{\mathsf{b}}^{-1}\, E_{\frac{N}{2},N} \cdot u_0, \\
u_2 &= \alpha_{\mathsf{b}}^{-1}\, E_{\frac{N}{2}-1,N-1} \cdot u_1, \\
&\vdots \\
u_{\frac{N}{2}} &= \alpha_{\mathsf{b}}^{-1}\, E_{1,\frac{N}{2}+1} \cdot u_{\frac{N}{2}-1},
\end{aligned} \qquad (4.35)$$

is a sequence of link states of depths $\frac{N}{2}, \frac{N}{2} - 1, \ldots, 0$. If $\alpha_{\mathsf{a}} \neq 0$, a similar sequence can be constructed starting from $u_0 = \Omega^{N/2} \cdot v_{0,0}(\frac{N}{2})$. In contrast, if $\alpha_{\mathsf{a}} = \alpha_{\mathsf{b}} = 0$, it is impossible to create a link state with depth $\frac{N}{2} - 1$, and the resulting module is not cyclic.

As explained above, the vectors $Q_{N/2,n} \cdot v_{k,\ell}(\frac{N}{2})$, with $n = 0, \ldots, N - 1$, are the generators of the one-dimensional submodules of $\mathsf{X}_{k,\ell,x,y,z}(N)$ isomorphic to $\mathsf{W}_{N/2,\omega_n}(N)$. For $m < N/2$, we may also produce states in the submodule isomorphic to $\mathsf{W}_{m,\omega_n}(N)$ using the action of $Q_{m,n}$. This is possible thanks to the push-through property (2.8). Indeed, the action of $\boldsymbol{F}$ on a link state of $\mathsf{X}_{k,\ell,x,y,z}(N)$ pushes outwards all the arcs that are not through-arcs. To illustrate, here is an example for $N = 12$:

$$\boldsymbol{F} \cdot \begin{array}{c}\text{(diagram)}\end{array} = \begin{array}{c}\text{(diagram)}\end{array} = \begin{array}{c}\text{(diagram)}\end{array} = \begin{array}{c}\text{(diagram)}\end{array} . \tag{4.36}$$

Thus the action of $\boldsymbol{F} \in \mathcal{E}\mathsf{PTL}_{12}(\beta)$ on this link state yields a diagram wherein the braid transfer matrix $\boldsymbol{F} \in \mathcal{E}\mathsf{PTL}_6(\beta)$ is *inserted* on a restricted set of nodes, namely those attached to the defects and through-arcs of the original link state. The same applies to any polynomial in $\boldsymbol{F}$, and in particular to $Q_{m,n}$: all arcs push outwards, and the result sees $Q_{m,n}$ inserted and acting on the $2m$ nodes attached to defects and through-arcs.

Thus, acting with $Q_{k+\ell+p,n}$ on a link state $v$ of depth $p$, itself produced by the action of $\mathcal{E}\mathsf{PTL}_N(\beta)$ on $v_{k,\ell}(\frac{N}{2})$, we obtain a state in the image of the map $\mathsf{W}_{m,\omega_n}(N) \to \mathsf{X}_{k,\ell,x,y,z}(N)$. For generic $q, z$, the module $\mathsf{W}_{m,\omega_n}(N)$ is irreducible, so the action of $\mathcal{E}\mathsf{PTL}_N(\beta)$ on $Q_{k+\ell+p,n} \cdot v$ generates the full submodule isomorphic to $\mathsf{W}_{m,\omega_n}(N)$. The same idea applies to states $v$ of depth $p = 0$ and with $2r$ defects, to show that the submodule $\mathsf{W}_{r,\omega_n}(N)$ can be generated from the action of $\mathcal{E}\mathsf{PTL}_N(\beta)$ on $Q_{r,\omega_n} \cdot v$. Because $v$ is obtained from the action of the algebra on $v_{k,\ell}(\frac{N}{2})$, all factors in the decomposition of $\mathsf{X}_{k,\ell,x,y,z}(N)$ are thus generated from $v_{k,\ell}(N/2)$, ending the proof. $\blacksquare$

## 4.4 Examples of indecomposable modules for $z$ non-generic

In this subsection, we investigate examples of module decompositions in the case where $z$ is non-generic. We set $N$ to an even integer and focus our attention on the module $\mathsf{X}_{0,0,x,y,z}(N)$ where there are no defects at all. We also set $z = \varepsilon q^k$ with $k$ a non-zero integer and $\varepsilon \in \{-1, +1\}$. The discussion below uses two features that pertain to generic values of $z$: (i) the invariant Gram product, and (ii) a conjecture for certain components of eigenvectors of $\boldsymbol{F}$ in the representation $\mathsf{X}_{0,0,x,y,z}(N)$. After presenting these two elements, we discuss the module structure of $\mathsf{X}_{0,0,x,y,z}(N)$ for $z = \varepsilon q^k$, first for $k = \frac{N}{2}$, and second for the other values of $k$.

**The Gram product.** We define the Gram product $\langle v, w \rangle$ for $v, w$ two link states $v \in \mathsf{X}_{0,0,x',y',z}(N)$ and $w \in \mathsf{X}_{0,0,x,y,z}(N)$ as follows. We draw $w$ on the top cap of the cylinder and reflect $v$ vertically, embedding it on the bottom cap of the cylinder. Joining the two caps produces a diagram of non-intersecting loop segments drawn on the sphere with four marked points. The two marked points on the top cap are denoted $\mathsf{a}$ and $\mathsf{b}$ whereas those on the bottom cap are denoted $\bar{\mathsf{a}}$ and $\bar{\mathsf{b}}$. A closed loop on this sphere can wind around the four marked points in eight possible ways. We assign it a weight $\beta$ if it encircles none (or all) of the marked points, and $\alpha_{\mathsf{a}}, \alpha_{\mathsf{b}}, \alpha_{\bar{\mathsf{a}}}, \alpha_{\bar{\mathsf{b}}}, \alpha_{\mathsf{a},\mathsf{b}}, \alpha_{\mathsf{a},\bar{\mathsf{a}}}, \alpha_{\mathsf{a},\bar{\mathsf{b}}}$ if it encircles a subset of the marked points. For instance, a loop encircling the points $\mathsf{a}, \bar{\mathsf{a}}$ and $\bar{\mathsf{b}}$ is equivalent to a loop encircling only the point $\mathsf{b}$ and is assigned the weight $\alpha_{\mathsf{b}}$. For the same reason, the two parameters $z$

are chosen identically for the modules $\mathsf{X}_{0,0,x,y,z}(N)$ and $\mathsf{X}_{0,0,x',y',z}(N)$, so that $\alpha_{\mathsf{ab}} = \alpha_{\bar{\mathsf{a}}\bar{\mathsf{b}}} = z + z^{-1}$. The Gram product $\langle v, w \rangle$ is then equal to the product of the weights of its loops. This product is extended sesquilinearly to all $v \in \mathsf{X}_{0,0,x',y',z}(N)$ and $w \in \mathsf{X}_{0,0,x,y,z}(N)$. Here are two examples to illustrate:

$$\left\langle \vcenter{\hbox{\includegraphics{fig1}}} , \vcenter{\hbox{\includegraphics{fig2}}} \right\rangle = \alpha_{\mathsf{ab}} , \qquad \left\langle \vcenter{\hbox{\includegraphics{fig3}}} , \vcenter{\hbox{\includegraphics{fig4}}} \right\rangle = \beta \alpha_{\bar{\mathsf{b}}} . \tag{4.37}$$

The Gram product is invariant under the action of $\mathcal{E}\mathsf{PTL}_N(\beta)$, namely $\langle v, a \cdot w \rangle = \langle a^\dagger \cdot v, w \rangle$, where $a^\dagger$ is obtained from $a$ by a vertical reflection of the cylinder. In particular, we have $e_j^\dagger = e_j$, $\Omega^\dagger = \Omega^{-1}$ and $(e_i e_j)^\dagger = e_j e_i$.

Furthermore, we note that, restricted to states $v, w$ of zero depth, the Gram product is identical to the same product defined over $\mathsf{W}_{0,z}(N)$, with $\alpha_{\mathsf{a},\mathsf{b}} \to \alpha$. We recall that in this case the radical of the Gram product, namely the set of states $w \in \mathsf{W}_{0,z}(N)$ satisfying $\langle v, w \rangle = 0$ for all $v$ in $\mathsf{W}_{0,z}(N)$, is the maximal non-trivial submodule of $\mathsf{W}_{0,z}(N)$. The quotient of $\mathsf{W}_{0,z}(N)$ by this submodule is the irreducible module $\mathsf{I}_{0,z}(N)$, see Section 2.4.

**Conjectural form for an eigenvector component.** Let us define the two link states

$$v_0 = \vcenter{\hbox{\includegraphics{fig5}}} , \qquad v_1 = \vcenter{\hbox{\includegraphics{fig6}}} , \tag{4.38}$$

whose depths are $\frac{N}{2}$ and $0$, respectively. For generic values of $q$ and $z$, on the module $\mathsf{X}_{0,0,x,y,z}(N)$ the operator $\boldsymbol{F}$ satisfies the identity

$$\left(\boldsymbol{F} - f_0(z)\mathbf{1}\right) \prod_{m=1}^{N/2} \prod_{n=0}^{2m-1} \left(\boldsymbol{F} - f_{m,n}\mathbf{1}\right) = 0, \tag{4.39}$$

where

$$f_0(z) = z + z^{-1}, \qquad f_{m,n} = q^m \mathrm{e}^{\mathrm{i}n\pi/m} + q^{-m}\mathrm{e}^{-\mathrm{i}n\pi/m}, \tag{4.40}$$

are the eigenvalues of $\boldsymbol{F}$. We construct the unique state $\psi_\varepsilon$ in the one-dimensional submodule of $\mathsf{X}_{0,0,x,y,z}(N)$ isomorphic to $\mathsf{W}_{N/2,\varepsilon}(N)$ as

$$\psi_\varepsilon = \frac{\boldsymbol{F} - f_0(z)\mathbf{1}}{f_{N/2,j} - f_0(z)} \prod_{\substack{m=1 \\ (m,n)\neq(N/2,j)}}^{N/2} \prod_{n=0}^{2m-1} \frac{\boldsymbol{F} - f_{m,n}\mathbf{1}}{f_{N/2,j} - f_{m,n}} \cdot v_0, \qquad j = \begin{cases} 0 & \varepsilon = +1, \\ \frac{N}{2} & \varepsilon = -1. \end{cases} \tag{4.41}$$

Its eigenvalue of $\boldsymbol{F}$ is $\varepsilon(q^{N/2} + q^{-N/2})$. Clearly, $\psi_\varepsilon$ is a non-trivial linear combination of link states. We denote by $\kappa_\varepsilon$ its component along the state $v_1$. We formulate the following conjecture.

CONJECTURE 1 *For $\varepsilon = +1$ and $\varepsilon = -1$, the component $\kappa_\varepsilon$ is*

$$\kappa_\varepsilon = \frac{\prod_{i=-(N-2)/4}^{(N-2)/4}(q^{2i}xy + \varepsilon)(q^{2i}x^{-1} + \varepsilon y^{-1})}{2(q^{N/2}z - \varepsilon)(z^{-1} - \varepsilon q^{-N/2})\prod_{i=1}^{(N-2)/2}(q^i - q^{-i})^2}. \tag{4.42}$$

We checked this conjecture for $N = 2, 4, \ldots, 12$ using our computer implementation of the module $\mathsf{X}_{0,0,x,y,z}(N)$.

**Module decomposition of $\mathsf{X}_{0,0,x,y,z}(N)$ for $z = \varepsilon q^{N/2}$.** The above conjecture has important implications for the Jordan cell structure of $\boldsymbol{F}$ and the module decomposition of $\mathsf{X}_{0,0,x,y,z}(N)$ for $z = \varepsilon q^k$. We start by discussing the case $k = \frac{N}{2}$. In this case, we write $z_c = \varepsilon q^{N/2}$ and $j_c = 0, N/2$ for $\varepsilon = +1, -1$ respectively. We focus on generic values of $x$ and $y$, namely those for which the numerator in (4.42) is non-zero. Under these circumstances, the equation (4.39) still holds, and two of the factors on the left-hand side are identical because $f_0(z_c) = f_{N/2,j_c}$. This implies that $\boldsymbol{F}$ may have Jordan cells of rank two tying the subrepresentations $\mathsf{W}_{N/2,\varepsilon}(N)$ and $\mathsf{W}_{0,\varepsilon q^{N/2}}(N)$ that appear as direct summands for generic $z$. Because $\mathsf{W}_{N/2,\varepsilon}(N)$ is one-dimensional, there can be at most one such Jordan cell. To see that this rank-two cell indeed arises, we consider the Laurent series of $\psi_\varepsilon$ around $z = z_c$:

$$\psi_\varepsilon = \frac{\psi_\varepsilon^{(-1)}}{z - z_c} + \psi_\varepsilon^{(0)} + \mathcal{O}(z - z_c). \tag{4.43}$$

We know from (4.42) that $\psi_\varepsilon^{(-1)}$ is non-zero. Moreover, because $z$ arises in $\mathsf{X}_{0,0,x,y,z}(N)$ only in its submodule $\mathsf{X}_{0,0,x,y,z}^{(0)}(N) \simeq \mathsf{W}_{0,z}(N)$, the only components of $\psi_\varepsilon$ that depend on $z$ have depth $p = 0$. We therefore conclude that $\psi_\varepsilon^{(-1)} \in \mathsf{X}_{0,0,x,y,z}^{(0)}(N)$. Let us also write the Taylor series

$$\boldsymbol{F} = \boldsymbol{F}^{(0)} + (z - z_c)\boldsymbol{F}^{(1)} + \mathcal{O}((z - z_c)^2). \tag{4.44}$$

The identity $(\boldsymbol{F} - f_{N/2,j_c}\mathbf{1}) \cdot \psi_\varepsilon = 0$ is satisfied at all orders in $z - z_c$. Equating the first two orders to zero separately, we find

$$(\boldsymbol{F}^{(0)} - f_{N/2,j_c}\mathbf{1}) \cdot \psi_\varepsilon^{(-1)} = 0, \qquad (\boldsymbol{F}^{(0)} - f_{N/2,j_c}\mathbf{1}) \cdot \psi_\varepsilon^{(0)} = -\boldsymbol{F}^{(1)} \cdot \psi_\varepsilon^{(-1)} = (z_c^{-2} - 1)\psi_\varepsilon^{(-1)}. \tag{4.45}$$

The last equality follows from the fact that $\frac{d\boldsymbol{F}}{dz} = (1 - z^{-2})\delta_{p=0}\mathbf{1}$. We conclude that the pair $(\psi_\varepsilon^{(-1)}, \psi_\varepsilon^{(0)})$ forms a Jordan cell for $\boldsymbol{F}^{(0)}$ with the eigenvalue $f_{N/2,j_c}$.

We now want to determine the structure of the resulting module $\mathsf{X}_{0,0,x,y,z_c}(N)$. From the results of Graham and Lehrer (see Section 2.4), we know that the standard modules $\mathsf{W}_{0,\varepsilon q^{N/2}}(N)$ and $\mathsf{W}_{N/2,\varepsilon}(N)$ have the Loewy diagrams

$$\mathsf{W}_{0,\varepsilon q^{N/2}}(N) \simeq \quad \begin{matrix} \mathsf{I}_{0,\varepsilon q^{N/2}}(N) \\ \searrow \\ \mathsf{I}_{N/2,\varepsilon}(N) \end{matrix} \quad , \qquad \mathsf{W}_{N/2,\varepsilon} \simeq \mathsf{I}_{N/2,\varepsilon}. \tag{4.46}$$

(All the other standard modules $\mathsf{W}_{m,e^{i\pi n/m}}(N)$ that appear in the decomposition (4.1) for $q, z$ generic are irreducible.) The element $\boldsymbol{F}$ has a Jordan cell tying these two factors, so these two standard modules cannot appear as the direct sum $\mathsf{W}_{0,\varepsilon q^{N/2}}(N) \oplus \mathsf{W}_{N/2,\varepsilon}(N)$ in the decomposition of $\mathsf{X}_{0,0,x,y,z_c}(N)$. They instead join to form an indecomposable module. The two possible structures are

$$\begin{matrix} \mathsf{I}_{N/2,\varepsilon}(N) \\ \swarrow \\ \mathsf{I}_{0,\varepsilon q^{N/2}}(N) \\ \searrow \\ \mathsf{I}_{N/2,\varepsilon}(N) \end{matrix} \qquad \text{and} \qquad \begin{matrix} \mathsf{I}_{0,\varepsilon q^{N/2}}(N) \quad \mathsf{I}_{N/2,\varepsilon}(N) \\ \searrow \qquad \swarrow \\ \mathsf{I}_{N/2,\varepsilon}(N) \end{matrix}. \tag{4.47}$$

To determine which structure is the correct one, we define the state

$$\tilde{\psi}_\varepsilon^{(0)} = Q_\varepsilon(\boldsymbol{F}) \cdot v_0 \Big|_{z=z_c}, \qquad Q_\varepsilon(\boldsymbol{F}) = \prod_{\substack{m=1 \\ (m,n)\neq(N/2,j_c)}}^{N/2} \prod_{n=0}^{2m-1} \frac{\boldsymbol{F} - f_{m,n}\mathbf{1}}{f_{N/2,j_c} - f_{m,n}}. \tag{4.48}$$

This is an alternative construction of a Jordan partner to the state $\psi_\varepsilon^{(-1)}$. Indeed, the existence of a rank-two Jordan cell implies that

$$(\boldsymbol{F} - f_{N/2,j_c}\mathbf{1})Q_\varepsilon(\boldsymbol{F})\Big|_{z=z_c} \neq 0. \tag{4.49}$$

This in turn implies that $\tilde{\psi}_\varepsilon^{(0)}$ is non-zero and satisfies

$$(\boldsymbol{F} - f_{N/2,j_c}\mathbf{1}) \cdot \tilde{\psi}_j^{(0)} \neq 0, \qquad (\boldsymbol{F} - f_{N/2j_c}\mathbf{1})^2 \cdot \tilde{\psi}_j^{(0)} = 0. \tag{4.50}$$

Therefore $\tilde{\psi}_\varepsilon^{(0)} - \psi_\varepsilon^{(0)}$ is a scalar multiple of $\psi_\varepsilon^{(-1)}$. Because $\tilde{\psi}_\varepsilon^{(0)}$ has components with depth $\frac{N}{2}$, it has a non-zero component in the factor $\mathsf{I}_{N/2,\varepsilon}(N)$ that lies in the head of the module.

Let us now consider the state $e_N \cdot \tilde{\psi}_\varepsilon^{(0)}$. This state has no non-zero components with depth $p = \frac{N}{2}$. It therefore does not enter the factor $\mathsf{I}_{N/2,\varepsilon}(N)$ of the head. It is instead in the submodule $\mathsf{X}_{0,0,x,y,z_c}^{(0)}(N) \simeq \mathsf{W}_{0,\varepsilon q^{N/2}}(N)$ of depth zero. If the rightmost structure in (4.47) is the correct one, then this implies that the state $e_N \cdot \tilde{\psi}_\varepsilon^{(0)}$ is in the radical of the Gram product. A simple calculation shows that this is not the case:

$$\begin{aligned} \langle v_1, e_N \cdot \tilde{\psi}_\varepsilon^{(0)} \rangle &= \langle e_N \cdot v_1, \tilde{\psi}_\varepsilon^{(0)} \rangle = \beta \langle v_1, Q_\varepsilon(\boldsymbol{F}) \cdot v_0 \rangle = \beta \langle Q_\varepsilon(\boldsymbol{F}) \cdot v_1, v_0 \rangle \\ &= \beta \, Q_\varepsilon\big(f_0(z_c)\big)\langle v_1, v_0 \rangle = \alpha_{\mathsf{b}}^{N/2}\beta \, Q_\varepsilon\big(f_0(z_c)\big). \end{aligned} \tag{4.51}$$

This is non-zero for generic values of the parameters $\alpha_{\mathsf{b}}$ and $\beta$. This calculation uses the invariance of the Gram product, as well as the property $\boldsymbol{F}^\dagger = \boldsymbol{F}$. We conclude that, for generic values of $x$ and $y$, the decomposition of $\mathsf{X}_{0,0,x,y,z_c}(N)$ is

$$\mathsf{X}_{0,0,x,y,z_c}(N) \simeq \mathsf{I}_{0,\varepsilon q^{N/2}} \begin{array}{c} \nearrow \mathsf{I}_{N/2,\varepsilon} \\ \\ \searrow \mathsf{I}_{N/2,\varepsilon} \end{array} \oplus \bigoplus_{\substack{m=1 \\ (m,n)\neq(N/2,j_c)}}^{N/2} \bigoplus_{n=0}^{2m-1} \mathsf{W}_{m,\mathrm{e}^{\mathrm{i}\pi n/m}}. \tag{4.52}$$

We comment briefly on values of $z$ of the form $z_j = q^{N/2}\mathrm{e}^{2\mathrm{i}\pi j/N}$ with $j \in \{1,\ldots,\frac{N}{2}-1\} \cup \{\frac{N}{2}+1,\ldots,N-1\}$. In this case, we also have the equality $f_0(z_j) = f_{N/2,j}$ of the eigenvalues of $\boldsymbol{F}$ over the factors $\mathsf{W}_{0,z_j}(N)$ and $\mathsf{W}_{N/2,\mathrm{e}^{2\mathrm{i}\pi j/N}}(N)$. One may thus think that an indecomposable module tying these two factors can form for $z = z_j$. However, this cannot happen because the eigenvalues of $\bar{\boldsymbol{F}}$ on these factors do not coincide for $z = z_j$. Thus in this case, $\mathsf{X}_{0,0,x,y,z_j}(N)$ simply decomposes as the direct sum (4.1).

**Module decomposition of $\mathsf{X}_{0,0,x,y,z}(N)$ for $z = \varepsilon q^k$.** For $k = 1,\ldots,N-1$, the insertion algorithm allows us to obtain the module decomposition for $z = z_c = \varepsilon q^k$, with $j_c = 0, k$ for $\varepsilon = +1, -1$ respectively. For a given link state $v \in \mathsf{W}_{k,\varepsilon}(N)$, we select the $2k$ nodes of $v$ attached to through arcs,

erase those through-arcs, and obtain $\Phi(v) \in \mathsf{X}_{0,0,x,y,z_c}(N)$ by attaching to these nodes the state $\psi_\varepsilon^{(0)}$ on $2k$ nodes that we constructed above. Acting with $\boldsymbol{F}$ on this state, all arcs not attached to the inserted state push outwards, as in the example (4.36).

Therefore $\boldsymbol{F}$ acts non-trivially only on the nodes attached to the inserted state $\psi_\varepsilon^{(0)}$. Repeating the above analysis, we find that $\Phi(v)$ is the Jordan partner in a rank-two Jordan cell of $\boldsymbol{F}$. The determination of the module structure follows the same arguments as above and yields the decomposition

$$\mathsf{X}_{0,0,x,y,\varepsilon q^k}(N) \simeq \mathsf{I}_{0,\varepsilon q^k}(N) \qquad \oplus \bigoplus_{\substack{m=1 \\ (m,n) \neq (k,j_c)}}^{N/2} \bigoplus_{n=0}^{2m-1} \mathsf{W}_{m,\mathrm{e}^{\mathrm{i}\pi n/m}}(N), \qquad k = 1, \ldots, \tfrac{N}{2}. \qquad (4.53)$$

with arrows from $\mathsf{I}_{k,\varepsilon}(N)$ (above) and $\mathsf{I}_{k,\varepsilon}(N)$ (below) pointing to $\mathsf{I}_{0,\varepsilon q^k}(N)$.

# 5  Connectivity operators and correlation functions

In this section, we describe the relation between the modules $\mathsf{X}_{k,\ell,x,y,z}(N)$ and connectivity operators in the loop model, and argue that the module decomposition of $\mathsf{X}_{k,\ell,x,y,z}(N)$ gives useful information about the physical correlation functions in the scaling limit.

We note that this section is intended as an illustration of the relevance of the representation-theoretic results presented in the previous sections, especially for the readers interested in the CFT description of critical random curves – one of the main motivations mentionned in the Introduction. However, the reader should be warned that, for the sake of conciseness, we choose to present below some of the concepts and examples more loosely than in the rest of the article.

## 5.1  Connectivity operators

Let us restrict the system size $N$ to an even integer, and the defect numbers $k$ and $\ell$ to integers. We define the connectivity operators $\mathcal{O}_{k,x}(j)$ for $j = 1, \ldots, N$ as the diagrams

$$\mathcal{O}_{0,x}(j) = \quad , \qquad \mathcal{O}_{k,x}(j) = \qquad \text{for } k > 0, \qquad (5.1)$$

where the indices $j + 1, \ldots, j + k - 1$ are understood modulo $N$. For $k \geqslant 2$, the connectivity operator for $j > N - k + 1$ is defined in such a way that the dashed segment connects the marked point with midpoint between $1$ and $N$ on the perimiter without crossing any defect. These operators are not elements of $\mathcal{E}\mathsf{PTL}_N(\beta)$. They can instead be seen as maps $\mathsf{W}_{\ell,y}(N) \to \mathsf{X}_{k,\ell,x,y,z}(N)$. The action of $\mathcal{O}_{k,x}(j)$ on a link state $v$ of $\mathsf{W}_{\ell,y}(N)$ is defined as usual, by drawing $v$ inside $\mathcal{O}_{k,x}(j)$. The output is a state of $\mathsf{X}_{k,\ell,x,y,z}(N)$ wherein the marked points of $\mathcal{O}_{k,x}(j)$ and of $v$ are identified as the points $\mathsf{a}$ and $\mathsf{b}$, respectively. The result is then simplified with the diagrammatic rules described in Section 3.2. This action depends on the parameter $z$, which we do not include as a label on $\mathcal{O}_{k,x}(j)$.

With these definitions, it is straightforward to check that the following relations hold for $k > 0$:

$$e_i \, \mathcal{O}_{k,x}(j) = \mathcal{O}_{k,x}(j) \, e_i \,, \qquad \{i, i+1\} \cap \{j, j+1, \dots j+k-1\} = \emptyset \,, \tag{5.2a}$$

$$e_i \, \mathcal{O}_{k,x}(j) = 0 \,, \qquad i = j, j+1, \dots, j+k-2 \,, \tag{5.2b}$$

$$\Omega^{-1} \, \mathcal{O}_{k,x}(j) \, \Omega = \mathcal{O}_{k,x}(j+1) \,, \qquad j = 1, \dots, N-k \,. \tag{5.2c}$$

For $k = 0$, we instead have

$$e_i \, \mathcal{O}_{0,x}(j) = \mathcal{O}_{0,x}(j) \, e_i \,, \qquad i \neq j \,, \tag{5.3a}$$

$$e_j \, \mathcal{O}_{0,x}(j) e_j = (x + x^{-1}) \, e_j \, \mathcal{O}_{0,x}(j) \,, \tag{5.3b}$$

$$\Omega^{-1} \, \mathcal{O}_{0,x}(j) \, \Omega = \mathcal{O}_{0,x}(j+1) \,, \qquad j = 1, \dots, N-1 \,. \tag{5.3c}$$

It is easy to see that we have the identity

$$z^k \Omega^{-k} \mathcal{O}_{k,x}(1) \cdot v_\ell(\tfrac{N}{2}) = v_{k,\ell}(\tfrac{N}{2}) \qquad \text{for} \quad k \leqslant \tfrac{N}{2} - \ell, \tag{5.4}$$

from which we conclude that the image of $\mathsf{W}_{\ell,y}(N)$ under $\mathcal{O}_{k,x}(1)$ includes a link state of maximal depth. Applying the proper rotations, we find that the same result holds with $\mathcal{O}_{k,x}(j)$ for the other values of $j$. From Proposition 4.1, for generic $q$ and $z$, the repeated action of the algebra $\mathcal{E}\mathsf{PTL}_N(\beta)$ on $\mathcal{O}_{k,x}(j) \cdot \mathsf{W}_{\ell,y}(N)$ thus generates the whole module $\mathsf{X}_{k,\ell,x,y,z}(N)$:

$$\mathsf{X}_{k,\ell,x,y,z}(N) = \widehat{\mathcal{O}}_{k,x} \cdot \mathsf{W}_{\ell,y}(N) \,, \tag{5.5}$$

where we have introduced the set of *dressed* connectivity operators

$$\widehat{\mathcal{O}}_{k,x} = \left\{ a \, \mathcal{O}_{k,x}(1) \, b \,, \quad a, b \in \mathcal{E}\mathsf{PTL}_N(\beta) \right\}. \tag{5.6}$$

Diving in one layer deeper, the connectivity operators also allow us to describe the standard modules in terms of the vacuum module $\mathsf{V}(N)$. The action of connectivity operators on $\mathsf{V}(N)$ takes as input link states with no marked point and outputs link states with a single marked point. These operators are thus seen as maps $\mathsf{V}(N) \to \mathsf{W}_{k,x}(N)$. Moreover, the action of $\mathcal{O}_{k,x}(j)$ is clearly nonzero. As a result, for $q$ and $x$ generic, the action of $\mathcal{E}\mathsf{PTL}_N(\beta)$ on $\mathcal{O}_{k,x}(j) \cdot \mathsf{V}(N)$ generates the full irreducible standard module $\mathsf{W}_{k,x}(N)$. In this sense, $\mathcal{O}_{k,x}(j)$ is a connectivity operator associated to the standard module $\mathsf{W}_{k,x}(N)$. We summarise these facts in terms of dressed connectivity operators as

$$\mathsf{W}_{k,x}(N) = \widehat{\mathcal{O}}_{k,x} \cdot \mathsf{V}(N) \,. \tag{5.7}$$

Combining the above results, we find that the operators $\mathcal{O}_{k,x}(i)$ and $\mathcal{O}_{\ell,y}(j)$, together with the repeated action of $\mathcal{E}\mathsf{PTL}_N(\beta)$, generate the module $\mathsf{X}_{k,\ell,x,y,z}(N)$ from the vacuum module:

$$\mathsf{X}_{k,\ell,x,y,z}(N) = \widehat{\mathcal{O}}_{k,x} \cdot \widehat{\mathcal{O}}_{\ell,y} \cdot \mathsf{V}(N) \,. \tag{5.8}$$

More generally, the action of $n$ connectivity operators $\mathcal{O}_{k_i,x_i}(j_i)$ on $\mathsf{V}(N)$ produces link states with $n$ marked nodes. This action involves a number of parameters $z_i$ that describe the interaction between the connectivity operators. In the case where there are no defects, namely $k_i = 0$ for each $i$, there are precisely $2^n - n - 1$ such parameters: $2^n$ is the number of ways the closed loops can encircle the $n$ marked points in the disc, $-n$ accounts for the parameters $x_i$ that already parameterise the weight of the loops surrounding the individual marked points, and the extra $-1$ accounts for the fact that loops encircling none of the marked points have the fugacity $\beta = -q - q^{-1}$.

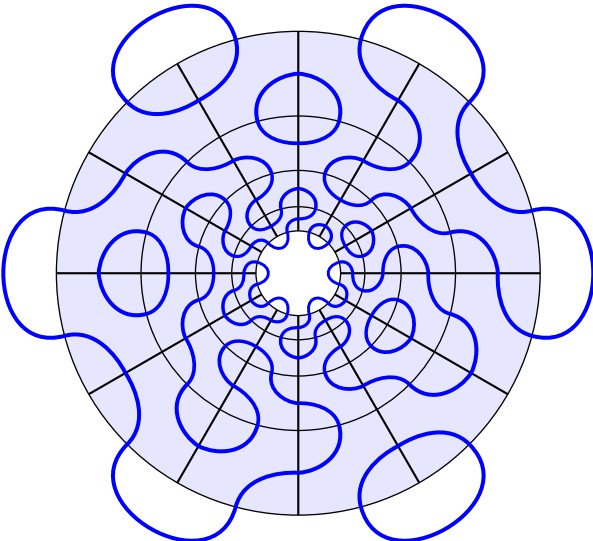

Figure 1: A loop configuration on the cylinder of perimeter $N = 12$ and height $M = 4$.

## 5.2 Correlation functions in the loop model

**The loop model on a cylinder.** We consider the dense loop model on a cylinder of perimeter $N$ and height $M$, and with simple reflecting boundary conditions at the two ends of the cylinder. A configuration of this model is a choice of the two possible loop tiles for each of the $MN$ faces of the lattice. An example of a configuration is given in Figure 1. The Boltzmann weight of a loop configuration $c$ is defined as

$$w(c) = \beta^{\#(c)} , \tag{5.9}$$

where $\#(c)$ is the total number of closed loops in $c$. The corresponding Gibbs measure is

$$\langle \mathcal{F} \rangle = \frac{1}{Z} \sum_c w(c)\, \mathcal{F}(c) , \qquad Z = \sum_c w(c) , \tag{5.10}$$

where $\mathcal{F}$ denotes any function of the loop configuration $c$. Here we shall focus on physical observables where $\mathcal{F}$ is a product of connectivity operators. The transfer matrix $\boldsymbol{T}$ for this system is

$$\boldsymbol{T} = \quad \text{where} \quad \square = \diagup\!\!\!\diagdown + \diagdown\!\!\!\diagup . \tag{5.11}$$

The partition function on the cylinder is given by

$$Z = \left\langle b, \boldsymbol{T}^M \cdot b \right\rangle , \tag{5.12}$$

where $\langle v, w \rangle$ is the Gram product of two states $v, w$ in $\mathsf{V}(N)$, and $b$ is the link state in $\mathsf{V}(N)$ that defines the boundary conditions:

$$b = \qquad . \tag{5.13}$$

**Correlation functions.** To define the correlation functions of the connectivity operators, we fix $M$ to be an even integer and define coordinates $(s,t)$ on the cylinder, where $s \in [1, N]$ and $t \in [-\frac{M}{2}, \frac{M}{2}]$ are the positions along the perimeter and height of the cylinder, respectively. The insertion of an operator $\mathcal{O}_{k,x}$ at the position $(s,t)$ corresponds to replacing $\boldsymbol{T}^M$ in (5.12) by $\boldsymbol{T}^{M/2-t}\mathcal{O}_{k,x}(s)\boldsymbol{T}^{M/2+t}$. We thus define the connectivity operators in the "Heisenberg picture" as

$$\mathcal{O}_{k,x}(s,t) = \boldsymbol{T}^{-t}\mathcal{O}_{k,x}(s)\boldsymbol{T}^t. \tag{5.14}$$

The $n$-point correlation function on the $M \times N$ cylinder is then given by

$$\langle \mathcal{O}_{k_1,x_1}(s_1,t_1)\dots\mathcal{O}_{k_n,x_n}(s_n,t_n)\rangle_{[z]}^{(M)} = \frac{1}{Z}\left\langle \boldsymbol{T}^{M/2}\cdot b, \mathcal{O}_{k_1,x_1}(s_1,t_1)\dots\mathcal{O}_{k_n,x_n}(s_n,t_n)\boldsymbol{T}^{M/2}\cdot b\right\rangle_{[z]}, \tag{5.15}$$

where $t_1 \geqslant t_2 \geqslant \dots \geqslant t_n$.

On the right-hand side, $\langle v, w\rangle_{[z]}$ denotes a generalised Gram product wherein $v$ is an element of $\mathsf{V}(N)$, whereas $w$ belongs to a module of $\mathcal{E}\mathsf{PTL}_N(\beta)$ similar to $\mathsf{X}_{k,\ell,x,y,z}(N)$, but generalised to involve $n$ marked points. The Gram product of these two states is defined similarly to the one described in Section 4.4. Here we give its definition only for the restricted case where all twist parameters are set to 1, namely for the situation that is relevant for the discussion of Section 5.3. Let $v$ and $w$ be elements of modules similar to $\mathsf{X}_{k,\ell,x,y,z}(N)$, but whose link states have $n_1$ and $n_2$ marked points, respectively. To compute $\langle v, w\rangle_{[z]}$, we draw $v$ and $w$ on the top and bottom hemispheres of a sphere, respectively. Writing $n = n_1 + n_2$ and labelling the $n$ marked points with the integers $i = 1, \dots, n$, we assign to a closed loop surrounding the points $i_1, \dots, i_m$ the weight $\alpha_{i_1,\dots,i_m} = z_{i_1,\dots,i_m} + z_{i_1,\dots,i_m}^{-1}$. Because these loops live on a sphere, we impose that for any subset $I$ of $\{1, \dots, n\}$ and its complement $\bar{I} = \{1, \dots, n\}\backslash I$, the parameters obey $z_I = z_{\bar{I}}$. The Gram product $\langle v, w\rangle_{[z]}$ is then equal to the product of the weights of its loops. It is thus defined in terms of the set $[z]$ of variables $z_{i_1,\dots,i_m}$. With this definition, we have the self-adjoint property $\langle \mathcal{O}_{k,x}(s,t)\cdot v, w\rangle_{[z]} = \langle v, \mathcal{O}_{k,x}(s,t)\cdot w\rangle_{[z]}$.

In the limit $M \to \infty$, the cylinder becomes infinitely long, and we have $\boldsymbol{T}^{M/2}\cdot b \sim \Lambda^{M/2}v_0$, where $v_0$ is the Perron-Frobenius eigenvector of $\boldsymbol{T}$ in $\mathsf{V}(N)$ and $\Lambda$ is its eigenvalue. We normalise it so that $\langle v_0, v_0\rangle = 1$. We obtain the correlation function on the infinite cylinder of circumference $N$, by sending $M$ to infinity while keeping $t_1, \dots, t_n$ fixed:

$$\langle \mathcal{O}_{k_1,x_1}(\mathbf{r}_1)\dots\mathcal{O}_{k_n,x_n}(\mathbf{r}_n)\rangle_{[z]} = \lim_{M\to\infty}\langle \mathcal{O}_{k_1,x_1}(\mathbf{r}_1)\dots\mathcal{O}_{k_n,x_n}(\mathbf{r}_n)\rangle_{[z]}^{(M)}$$
$$= \langle v_0, \mathcal{O}_{k_1,x_1}(\mathbf{r}_1)\dots\mathcal{O}_{k_n,x_n}(\mathbf{r}_n)\cdot v_0\rangle_{[z]}. \tag{5.16}$$

Here we have also introduced the more compact notation $\mathbf{r}_j = (s_j, t_j)$ for the position of the operators. Using the self-adjoint property of $\mathcal{O}_{k,x}(s,t)$, we can express the two-, three- and four-point correlation functions as

$$\langle \mathcal{O}_{k,x}(\mathbf{r}_1)\mathcal{O}_{k,x}(\mathbf{r}_2)\rangle = \langle v_0, \mathcal{O}_{k,x}(\mathbf{r}_1)\mathcal{O}_{k,x}(\mathbf{r}_2)\cdot v_0\rangle, \tag{5.17a}$$

$$\langle \mathcal{O}_{k_1,x_1}(\mathbf{r}_1)\mathcal{O}_{k_2,x_2}(\mathbf{r}_2)\mathcal{O}_{k_3,x_3}(\mathbf{r}_3)\rangle = \langle \mathcal{O}_{k_1,x_1}(\mathbf{r}_1)\cdot v_0, \mathcal{O}_{k_2,x_2}(\mathbf{r}_2)\mathcal{O}_{k_3,x_3}(\mathbf{r}_3)\cdot v_0\rangle, \tag{5.17b}$$

$$\langle \mathcal{O}_{k_1,x_1}(\mathbf{r}_1)\mathcal{O}_{k_2,x_2}(\mathbf{r}_2)\mathcal{O}_{k_3,x_3}(\mathbf{r}_3)\mathcal{O}_{k_4,x_4}(\mathbf{r}_4)\rangle_{[z]} = \langle \mathcal{O}_{k_2,x_2}(\mathbf{r}_2)\mathcal{O}_{k_1,x_1}(\mathbf{r}_1)\cdot v_0, \mathcal{O}_{k_3,x_3}(\mathbf{r}_3)\mathcal{O}_{k_4,x_4}(\mathbf{r}_4)\cdot v_0\rangle_{[z]}. \tag{5.17c}$$

Here, the states in the second entry of the Gram products are elements of certains modules $\mathsf{X}_{k,\ell,x,y,z}(N)$. In the first two lines, the constraint that the Gram product be well-defined fixes the extra parameters $z_i$ entirely (and we omit the indices $[z]$ in these cases to lighten the notation). For instance, for the

two-point function with $k = 0$, a loop encircling both marked points is equivalent on the sphere to a loop encircling none of the two points. It is thus assigned a weight $\alpha_{\mathsf{ab}} = \beta$, corresponding to $z = -q$ or $z = -1/q$. Loops that separate $\mathbf{r}_1$ and $\mathbf{r}_2$ instead have a weight $x + x^{-1}$. Hence the Gram product takes place over $\mathsf{V}(N) \otimes \mathsf{X}_{0,0,x,x,-q}(N)$. In the three point function (5.17b) with $k_1 = k_2 = k_3 = 0$, a similar argument shows that the parameter $z$ must be set to $z = x_1$ or $z = 1/x_1$. Also, it is clear that the three-point function vanishes if one of the defect numbers is larger than the sum of the two others (for instance if $k_3 > k_1 + k_2$). This is consistent with the fact that, in (5.17b), the vector $\mathcal{O}_{k_1,x_1}(\mathbf{r}_1) \cdot v_0$ lives in $\mathsf{W}_{k_1,x_1}(N)$, whereas $\mathcal{O}_{k_2,x_2}(\mathbf{r}_2)\mathcal{O}_{k_3,x_3}(\mathbf{r}_3) \cdot v_0$ lives in $\mathsf{X}_{k_2,k_3,x_2,x_3,z}(N)$, which decomposes on standard modules $\mathsf{W}_{m,\omega}(N)$ with $m \geqslant |k_2 - k_3|$. Hence, if $k_3 > k_1 + k_2$, then $k_1 < |k_2 - k_3|$ and the Gram product in (5.17b) is zero.

Lastly, in the four-point function with $k_1 = \cdots = k_4 = 0$, the subscript $[z]$ accounts for four free variables that parameterise the loops with non-trivial windings, and which can be chosen as needed according to the correlation that we wish to study. If some of the defect numbers are non-zero, these extra parameters $z_i$ arise in the twist factors as defects wind around the marked points.

The two- and three-point correlation functions (5.17a) and (5.17b) can be related directly to "physical" correlation functions, defined in terms of partition functions with certain constraints or modified Boltzmann weights. For example,

$$\mathbb{P}^c(\mathbf{r}_1, \mathbf{r}_2) = \langle \mathcal{O}_{0,\mathsf{i}}(\mathbf{r}_1)\mathcal{O}_{0,\mathsf{i}}(\mathbf{r}_2)\rangle \qquad \text{and} \qquad \mathbb{P}^\ell(\mathbf{r}_1, \mathbf{r}_2) = \langle \mathcal{O}_{1,1}(\mathbf{r}_1)\mathcal{O}_{1,1}(\mathbf{r}_2)\rangle \qquad (5.18)$$

are the probabilities that $\mathbf{r}_1$ and $\mathbf{r}_2$ sit on the same cluster and on the same closed loop, respectively. The former is the natural physical observable in the Fortuin-Kasteleyn interpretation of the loop model, with the weight of clusters set to $Q = \beta^2$. Furthermore, the three-point function

$$G(\mathbf{r}_1, \mathbf{r}_2, \mathbf{r}_3) = \langle \mathcal{O}_{0,x_1}(\mathbf{r}_1)\mathcal{O}_{0,x_2}(\mathbf{r}_2)\mathcal{O}_{0,x_3}(\mathbf{r}_3)\rangle \qquad (5.19)$$

is built from the partition function with modified loop weights introduced in [15].

**Four-point functions.** We consider a four-point correlation function of the form

$$G(\mathbf{r}_1, \mathbf{r}_2, \mathbf{r}_3, \mathbf{r}_4) = \langle \mathcal{O}_{\ell,y}(\mathbf{r}_1)\mathcal{O}_{k,x}(\mathbf{r}_2)\mathcal{O}_{k,x}(\mathbf{r}_3)\mathcal{O}_{\ell,y}(\mathbf{r}_4)\rangle_z . \qquad (5.20)$$

Instead of considering the general case, from here onwards we focus on a special situation where all the twist factors for windings of defects are set to 1. Moreover, for $k = \ell$, we choose to assign the weight $z + z^{-1}$ to the loops encircling one, two or three of the points $\mathbf{r}_1, \ldots, \mathbf{r}_4$, and the weight $\beta = -q - q^{-1}$ to all the other loops. The resulting observable then depends on a single parameter $z$, which we write as a subscript without brackets in (5.20). In this case, the four-point function (5.17c) is written in terms of a Gram product over two copies of $\mathsf{X}_{k,k,x,y,z}(N)$ for $k = \ell$, and two copies of $\mathsf{X}_{k,\ell,x,y,1}(N)$ for $k \neq \ell$. Using the decomposition of $\mathsf{X}_{k,\ell,x,y,z}(N)$, we can write

$$G(\mathbf{r}_1, \mathbf{r}_2, \mathbf{r}_3, \mathbf{r}_4) = \begin{cases} G_{0,z}(\mathbf{r}_1, \mathbf{r}_2, \mathbf{r}_3, \mathbf{r}_4) + \displaystyle\sum_{m=1}^{N/2} \sum_{n=0}^{2m-1} G_{m,\exp(\mathsf{i}\pi n/m)}(\mathbf{r}_1, \mathbf{r}_2, \mathbf{r}_3, \mathbf{r}_4), & k = \ell, \\[2ex] G_{|k-\ell|,1}(\mathbf{r}_1, \mathbf{r}_2, \mathbf{r}_3, \mathbf{r}_4) + \displaystyle\sum_{m=|k-\ell|+1}^{N/2} \sum_{n=0}^{2m-1} G_{m,\exp(\mathsf{i}\pi n/m)}(\mathbf{r}_1, \mathbf{r}_2, \mathbf{r}_3, \mathbf{r}_4), & k \neq \ell, \end{cases}$$

$$(5.21)$$

where each $G_{m,\omega}$ is a quadratic sum of Gram products in the submodule of $\mathsf{X}_{k,k,x,y,z}(N)$ or $\mathsf{X}_{k,\ell,x,y,1}(N)$ isomorphic to $\mathsf{W}_{m,\omega}(N)$. Indeed, let us denote by $\{\mu_{m,\omega,j}\}$ an orthonormal basis for the Gram product in this submodule, for $q,\omega$ generic. Then $G_{m,\omega}$ is defined as

$$G_{m,\omega}(\mathbf{r}_1,\mathbf{r}_2,\mathbf{r}_3,\mathbf{r}_4) = \sum_{j=1}^{\dim \mathsf{W}_{m,\omega}(N)} \langle \mathcal{O}_{k,x}(\mathbf{r}_2)\mathcal{O}_{\ell,y}(\mathbf{r}_1)\cdot v_0, \mu_{m,\omega,j}\rangle \times \langle \mu_{m,\omega,j}, \mathcal{O}_{k,x}(\mathbf{r}_3)\mathcal{O}_{\ell,y}(\mathbf{r}_4)\cdot v_0\rangle . \quad (5.22)$$

This gives the contribution of the internal sector $\mathsf{W}_{m,\omega}(N)$ to the correlation function, in the fusion channel

$$(5.23)$$

Here and below, we sometimes drop the dependence on $N$ of the modules $\mathsf{W}_{k,z}(N)$ and $\mathsf{X}_{k,\ell,x,y,z}(N)$. An example of a physical four-point correlation function for the percolation or Fortuin-Kasteleyn cluster configurations is the probability that all four points lie in the same cluster. This observable corresponds to $G(\mathbf{r}_1,\mathbf{r}_2,\mathbf{r}_3,\mathbf{r}_4)$ with $k=\ell=0$, and with the weights of all loops that encircle a non-trivial subset of the four points set to zero.

## 5.3 Scaling limit

**Closure of the fusion algebra.** Before discussing the scaling limit, let us remark that the definition (5.20) of four-point functions leads to a special situation where the fusion procedure closes on a finite set of modules. This situation is realised in a loop model where all the loops that wind non-trivially around a non-trivial subset of the marked points are given the weight $\alpha = z + z^{-1}$, with $z$ generic, and any winding of the defects around the marked points is allocated a unit weight. The corresponding fusion of two standard modules is defined as

$$\mathsf{W}_{k,x} \times \mathsf{W}_{\ell,y} := \begin{cases} \mathsf{X}_{k,\ell,x,y,z} & \text{if } k = \ell, \\ \mathsf{X}_{k,\ell,x,y,1} & \text{if } k \neq \ell. \end{cases} \quad (5.24)$$

The decompositions of the modules $\mathsf{X}_{k,k,x,y,z}$ and $\mathsf{X}_{k,\ell,x,y,1}$ then produce the fusion rules

$$\mathsf{W}_{0,z} \times \mathsf{W}_{0,z} \to \mathsf{W}_{0,z} \oplus \bigoplus_{k=1}^{N/2}\bigoplus_{j=0}^{2k-1} \mathsf{W}_{k,\exp(\mathrm{i}\pi j/k)}, \quad (5.25a)$$

$$\mathsf{W}_{0,z} \times \mathsf{W}_{m,\exp(\mathrm{i}\pi n/m)} \to \mathsf{W}_{m,1} \oplus \bigoplus_{k=m+1}^{N/2}\bigoplus_{j=0}^{2k-1} \mathsf{W}_{k,\exp(\mathrm{i}\pi j/k)}, \quad (5.25b)$$

$$\mathsf{W}_{m,\exp(\mathrm{i}\pi n/m)} \times \mathsf{W}_{m',\exp(\mathrm{i}\pi n'/m')} \to \mathsf{W}_{|m-m'|,1} \oplus \bigoplus_{k=|m-m'|+1}^{N/2}\bigoplus_{j=0}^{2k-1} \mathsf{W}_{k,\exp(\mathrm{i}\pi j/k)}, \quad m \neq m', \quad (5.25c)$$

$$\mathsf{W}_{m,\exp(\mathrm{i}\pi n/m)} \times \mathsf{W}_{m,\exp(\mathrm{i}\pi n'/m)} \to \mathsf{W}_{0,z} \oplus \bigoplus_{k=1}^{N/2}\bigoplus_{j=0}^{2k-1} \mathsf{W}_{k,\exp(\mathrm{i}\pi j/k)}, \quad (5.25d)$$

where $m,m'$ take values in $1,\ldots,N/2$. Hence, this fusion closes on the set of modules:

$$\{\mathsf{W}_{0,z}\} \cup \{\mathsf{W}_{m,\exp(\mathrm{i}\pi n/m)} \mid m = 1,\ldots,\tfrac{N}{2}, \ n = 0,\ldots,2m-1\}. \quad (5.26)$$

Crucially, we note that the standard modules in (5.26) are precisely those that are required to express the Markov trace on the torus (with weight $\alpha$ for non-contractible loops) as a sum of traces [4, 53].

An important subtlety arises if one want to compute a correlation function such as $\langle \mathcal{O}_{0,z}(\mathbf{r}_1)\mathcal{O}_{0,z}(\mathbf{r}_2)\rangle$. In this case, the loops surrounding both marked points are assigned a weight $\alpha_{\mathsf{ab}} = \beta$, so in the fusion $\mathsf{W}_{0,z} \times \mathsf{W}_{0,z}$, one must instead select the fusion channel with $z \to -q$. This amounts to changing the first factor on the right side of (5.25a) to $\mathsf{W}_{0,-q}$. This module $\mathsf{W}_{0,-q}$ is reducible and has a non-zero quotient module isomorphic to the vacuum module $\mathsf{V}$, and as a result the correlator is non-zero. A similar process must be applied to compute a correlation function with more than two points: the fusion rule for $\mathsf{W}_{0,z} \times \mathsf{W}_{0,z}$ is used repeatedly in the $z$ channel, until only two fields are left and then one must use the $-q$ channel. In this process, the module $\mathsf{W}_{0,-q}$ only appears at the very last step, and thus it is not necessary to understand how it fuses with the other modules.

**Conformal field theory description of the loop model.** In the scaling limit, the loop model with $-2 < \beta \leqslant 2$ is described by a conformal field theory with central charge

$$c = 1 - 6(b^{-1} - b)^2 \,, \qquad \beta = -2\cos(\pi b^2) \,, \qquad 0 < b \leqslant 1 \,. \tag{5.27}$$

We recall the Kac notation for the conformal dimensions

$$h_{r,s} = \frac{(rb^{-1} - sb)^2 - (b^{-1} - b)^2}{4} \,. \tag{5.28}$$

We denote by $\mathcal{M}(h,\bar{h})$ the module of heighest weight $(h,\bar{h})$, over the pair of Virasoro algebras $\mathsf{Vir} \otimes \overline{\mathsf{Vir}}$. For generic values of $h$ and $\bar{h}$, $\mathcal{M}(h,\bar{h})$ is an irreducible module. By Coulomb gas arguments [4], one finds that a generic standard module scales to

$$\mathsf{W}_{m,\exp(\mathrm{i}\pi\mu)} \to \bigoplus_{p=-\infty}^{+\infty} \mathcal{M}(h_{\mu+p,m}, h_{\mu+p,-m}) \,. \tag{5.29}$$

The primary operators in the right-hand side of (5.29) are denoted $\Phi_{\mu+p,m}$, with conformal dimensions $(h_{\mu+p,m}, h_{\mu+p,-m})$. For $|\mu| \leqslant 1/2$, the leading primary operator is $\Phi_{\mu,m}$. More generally, if $r - 1/2 \leqslant \mu \leqslant r + 1/2$ with $r \in \mathbb{Z}$, the leading primary operator is $\Phi_{\mu-r,m}$. We note that, at finite $N$, the module $\mathsf{W}_{m,\exp(\mathrm{i}\pi\mu)}$ is invariant under $\mu \to \mu+2$, whereas in the scaling limit, the right side of (5.29) is periodic with $\mu \to \mu + 1$. This is because $\mathsf{W}_{m,\exp(\mathrm{i}\pi\mu)}$ and $\mathsf{W}_{m,-\exp(\mathrm{i}\pi\mu)}$ have the same scaling limit, although they are not isomorphic. Indeed, the leading eigenvalues, which survive in the scaling limit, are equal up to an overall minus sign. They therefore have the same conformal data.

For the set of standard modules discussed above, we get

$$\mathsf{W}_{0,z} \to \bigoplus_{p=-\infty}^{+\infty} \mathcal{M}(h_{\mu+p,0}, h_{\mu+p,0}) \,, \qquad \mathsf{W}_{m,\exp(\mathrm{i}\pi n/m)} \to \bigoplus_{p=-\infty}^{+\infty} \mathcal{M}(h_{n/m+p,m}, h_{n/m+p,-m}) \,, \tag{5.30}$$

where $z = \exp(\mathrm{i}\pi\mu)$ is set to a generic value. In the case $\alpha = \beta$, by setting $\mu = 1 - b^2$, one can identify the primary conformal dimensions in the zero-defect sector as

$$h_{\mu+p,0}\big|_{\mu=1-b^2} = h_{1+p,1} \,, \qquad p \in \mathbb{Z} \,, \tag{5.31}$$

which are degenerate under the Virasoro algebra for $p \geqslant 0$. This is the set of "energy-like" operators of the loop model, as already pointed out in [4]. In this situation with a non-generic parameter $z = -q$, we can expect from our analysis of the representations $\mathsf{X}_{k,\ell,x,y,z}$ the appearance of indecomposable

Virasoro modules in the scaling limit, with submodules which may decouple from the Hilbert space. Hence, some of the Virasoro fusion rules will be different from the Temperley-Lieb ones, as some of the submodules are suppressed by this quotient. However, the analysis of this non-generic case is beyond the scope of the present work.

Back to generic values of $z$, let us consider the scaling limit of the four-point function (5.20). Each connectivity operator scales to the leading primary operator in (5.29):

$$\mathcal{O}_{k,x} \to \Phi_{\lambda,k}, \qquad \mathcal{O}_{\ell,y} \to \Phi_{\nu,\ell}, \tag{5.32}$$

where

$$x = \exp(i\pi\lambda), \qquad y = \exp(i\pi\nu), \qquad -1/2 \leqslant \lambda, \nu \leqslant 1/2. \tag{5.33}$$

It is sufficient to consider this range for $\lambda$ and $\nu$, because the operators $\mathcal{O}_{k,x}$ and $\mathcal{O}_{k,-x}$ have the same scaling limit, up to alternating lattice factors. Hence the four-point function (5.20) scales to

$$\langle \Phi_{\nu,\ell}(w_1, \bar{w}_1) \Phi_{\lambda,k}(w_2, \bar{w}_2) \Phi_{\lambda,k}(w_3, \bar{w}_3) \Phi_{\nu,\ell}(w_4, \bar{w}_4) \rangle_{\text{cyl}}, \tag{5.34}$$

where we now use the complex coordinates $w = t + is, \bar{w} = t - is$ on the cylinder. Since $\Phi_{e,m}$ has conformal dimensions $(h_{e,m}, h_{e,-m})$, we also write $\Phi_{e,m}(w, \bar{w}) = \phi_{e,m}(w) \otimes \bar{\phi}_{e,-m}(\bar{w})$ for any $e, m$.

In the partial sums $G_{m,\omega}(\mathbf{r}_1, \mathbf{r}_2, \mathbf{r}_3, \mathbf{r}_4)$ (5.22), the intermediary states $\mu_{m,w,j}$ scale to an orthonormal basis of the Virasoro modules (5.29). The sum over these states can be organised as a double sum:

$$G_{m,\omega}(\mathbf{r}_1, \mathbf{r}_2, \mathbf{r}_3, \mathbf{r}_4) \to \sum_{p=-\infty}^{+\infty} \sum_{[r]} \langle 0 | \Phi_{\nu,\ell}(\mathbf{r}_1) \Phi_{\lambda,k}(\mathbf{r}_2) | \Phi^{[r]}_{\mu+p,m} \rangle \times \langle \Phi^{[r]}_{\mu+p,m} | \Phi_{\lambda,k}(\mathbf{r}_3) \Phi_{\nu,\ell}(\mathbf{r}_1) | 0 \rangle, \tag{5.35}$$

where $\omega = \exp(i\pi\mu)$ and for a given $|\Phi_{e,m}\rangle$, the set $\{|\Phi^{[r]}_{e,m}\rangle\}$ denotes an orthonormal basis in the module $\mathcal{M}(h_{e,m}, h_{e,-m})$. As a result, we get, up to simple overall prefactors, the decompositions over conformal blocks

$$G_{0,z}(\mathbf{r}_1, \mathbf{r}_2, \mathbf{r}_3, \mathbf{r}_4) \to \sum_{p=-\infty}^{+\infty} C^2_{\mu+p,0} F_{\mu+p,0}(\eta) \bar{F}_{\mu+p,0}(\bar{\eta}), \tag{5.36a}$$

$$G_{m,\exp(i\pi n/m)}(\mathbf{r}_1, \mathbf{r}_2, \mathbf{r}_3, \mathbf{r}_4) \to \sum_{p=-\infty}^{+\infty} C^2_{n/m+p,m} F_{n/m+p,m}(\eta) \bar{F}_{n/m+p,-m}(\bar{\eta}), \tag{5.36b}$$

where $F_{e,m}, \bar{F}_{e,-m}$ are the conformal blocks with internal conformal dimensions $h_{e,m}, h_{e,-m}$ respectively:

$$\tag{5.37}$$

The cross-ratio is given by

$$\eta = \frac{\sinh \frac{\pi w_{34}}{L} \sinh \frac{\pi w_{21}}{L}}{\sinh \frac{\pi w_{31}}{L} \sinh \frac{\pi w_{24}}{L}}, \qquad w_{ij} = w_i - w_j, \tag{5.38}$$

where $L$ is the physical circumference of the cylinder. The constants $C_{e,m}$ in (5.36a) and (5.36b) are the structure constants associated to the fusion $\Phi_{\lambda,k} \times \Phi_{\nu,\ell} \to \Phi_{e,m}$. A bootstrap argument [14], using the degenerate "energy-like" operator of dimensions $(h_{21}, h_{21})$ which lives in the sector corresponding to $\mathsf{V}$, yields the ratio of coefficients $C_{e+1,m}/C_{e-1,m}$. Up to our knowledge, a full determination of the coefficients $C_{e,m}$ for the loop model is still lacking.

Here, to get the decompositions in terms of conformal blocks in the planar geometry, we have used the conformal map

$$w \mapsto \frac{\sinh \frac{\pi(w-w_4)}{L} \sinh \frac{\pi w_{21}}{L}}{\sinh \frac{\pi(w-w_1)}{L} \sinh \frac{\pi w_{24}}{L}} \,. \tag{5.39}$$

This analysis of the four-point functions strongly suggests that, for generic $\lambda, \nu$ and $z = \exp(\mathrm{i}\pi\mu)$, the leading primary operators obey the fusion rules:

$$\Phi_{\lambda,k} \times \Phi_{\nu,k} \to \sum_{p=-\infty}^{\infty} \Phi_{\mu+p,0} + \sum_{m=1}^{\infty} \sum_{p=-\infty}^{\infty} \Phi_{p/m,m} \,, \tag{5.40a}$$

$$\Phi_{\lambda,k} \times \Phi_{\nu,\ell} \to \sum_{p=-\infty}^{\infty} \Phi_{p,|k-\ell|} + \sum_{m=|k-\ell|+1}^{\infty} \sum_{p=-\infty}^{\infty} \Phi_{p/m,m} \,, \qquad k \neq \ell \,. \tag{5.40b}$$

Crucially, this is precisely the expected form of fusion for non-chiral conformal fields, namely it involves the fields that appear in the decomposition of the torus partition function whose fractional Kac indices have arbitrarily large denominators [4]. We note that some of the operators in the right-hand side may be suppressed if the corresponding contribution to $G_{0,z}$ or $G_{m,\exp(\mathrm{i}\pi n/m)}$ turns out to vanish. Since the subleading primary operators $\Phi_{u+p,k}$ are obtained by fusing $\Phi_{u,k}$ with the degenerate operator of conformal dimensions $(h_{21}, h_{21})$, by the associativity of the fusion algebra, the above fusion rules extend to all primary operators in the loop model, as long as their parameters $\lambda, \nu$ are generic. Some subtle issues will occur if we set $z \to -q$, due to the appearance of Jordan blocks in the Virasoro representations. The full study of this problem is left for future work.

# 6 Conclusion

In this paper, we formulated a new prescription for the fusion of standard modules of the enlarged periodic Temperley-Lieb algebra $\mathcal{E}\mathsf{PTL}_N(-q - q^{-1})$. The corresponding representations $\mathsf{X}_{k,\ell,x,y,z}(N)$ have link states drawn on discs with two marked points. As such, these states can be seen as living on a surface of genus two. We obtained the decomposition of these representations over the irreducible standard modules for generic values of $q$ and $z$, and gave examples of non-trivial reducible yet indecomposable modules for non-generic values of $z$. We also showed that the representations $\mathsf{X}_{k,\ell,x,y,z}(N)$ are elegantly described in terms of the connectivity operators $\mathcal{O}_{k,x}(j)$ of the dense loop model, whose correlation functions are the physically interesting observables of the model.

With our prescription, the fusion of standard modules $\mathsf{W}_{k,x}(N_\mathsf{a}) \times_z \mathsf{W}_{\ell,y}(N_\mathsf{b})$ is well-defined for all values of $x$ and $y$, and is closed for generic values of $q$ and $z$. Moreover, it is stable as a function of $N_\mathsf{a}$ and $N_\mathsf{b}$, namely as these numbers increases, the dependence on $k$ and $\ell$ remains unchanged and the decomposition of the fusion product depends only on $N_\mathsf{a}$ and $N_\mathsf{b}$ through their sum, in the bounds of the direct sum of irreducible factors. It also satisfies the relation

$$\mathsf{W}_{k,x}(N_\mathsf{a}) \times_z \mathsf{W}_{\ell,y}(N_\mathsf{b}) = \mathsf{W}_{\ell,y}(N_\mathsf{b}) \times_{1/z} \mathsf{W}_{k,x}(N_\mathsf{a}) \,. \tag{6.1}$$

It is therefore not quite commutative, as exchanging the two modules requires selecting a different fusion channel. We note however that, for certain specific physical situations like the one considered in Section 5.3, the fusion product is in fact commutative. Furthermore, we note that an explicit algebraic definition of this fusion similar to the definition (3.1) for the ordinary Temperley-Lieb algebra is still missing.

Perhaps surprisingly, the module decompositions of $\mathsf{X}_{k,\ell,x,y,z}(N)$ for $q$ and $z$ generic given in Theorem 1 do not depend on the value of $x$ and $y$. It would however be incorrect to say that all dependence over $x$ and $y$ is lost. These numbers appear directly in the matrix representatives of the algebra's generators, but they only enter in the couplings between the different depth sectors. For $q, z$ generic, these couplings do not result in the appearance of indecomposable yet reducible modules over $\mathcal{E}\mathsf{PTL}_N(\beta)$. The situation is however different for non-generic values of the parameters. This is exemplified in the calculation of Section 4.4, where the module decomposition (4.52) holds only for generic values of $x$ and $y$, namely those where the numerator in (4.42) is non-vanishing. This resulting module structure is thus different for generic versus non-generic values of $x$ and $y$. A similar situation occurs if one studies the decomposition of the standard module $\mathsf{W}_{k,z}(N)$ as a module over $\mathsf{TL}_N(\beta)$. For $q, z$ generic, $\mathsf{W}_{k,z}(N)$ is isomorphic to a simple direct sum of irreducible standard modules $\mathsf{V}_\ell(N)$, whereas for non-generic values, one gets a sum of indecomposables whose structures depend on the relation between $q$ and $z$. Returning to the modules $\mathsf{X}_{k,\ell,x,y,z}(N)$, although $x$ and $y$ do not enter the module decomposition for $q, z$ generic, we nevertheless expect these numbers to appear in the conformal blocks in (5.36), and therefore to be physically relevant parameters.

Another natural question regards the associativity of the fusion product that we have defined, namely

$$\left(\mathsf{W}_{k_\mathsf{a},x_\mathsf{a}}(N_\mathsf{a}) \times \mathsf{W}_{k_\mathsf{b},x_\mathsf{b}}(N_\mathsf{b})\right) \times \mathsf{W}_{k_\mathsf{c},x_\mathsf{c}}(N_\mathsf{c}) \overset{?}{=} \mathsf{W}_{k_\mathsf{a},x_\mathsf{a}}(N_\mathsf{a}) \times \left(\mathsf{W}_{k_\mathsf{b},x_\mathsf{b}}(N_\mathsf{b}) \times \mathsf{W}_{k_\mathsf{c},x_\mathsf{c}}(N_\mathsf{c})\right). \tag{6.2}$$

Answering this equation requires understanding how to fuse $\mathsf{W}_{k_\mathsf{a},x_\mathsf{a}}(N_\mathsf{b})$ with $\mathsf{X}_{k_\mathsf{b},k_\mathsf{c},x_\mathsf{b},x_\mathsf{c},z}(N)$. In its current state, our prescription of fusion is not fully general, as it only applies to pairs of standard modules. The question of associativity of the fusion product thus remains unanswered. It will moreover be desirable in future work to obtain a completely general algebraic definition of this fusion, that holds for all modules over $\mathcal{E}\mathsf{PTL}_N(\beta)$, including the standard modules $\mathsf{W}_{k,z}(N)$, the irreducible submodules and quotients of $\mathsf{W}_{k,z}(N)$ for $q$ and/or $z$ non-generic, the modules $\mathsf{X}_{k,\ell,x,y,z}(N)$, and the modules arising in the periodic XXZ spin chain.

Our construction is not equivalent to the two previous proposals by other authors [37–39], as is made clear by the different module decompositions. A closer comparison is worthwhile. The fusion of [37,38] is constructed using the presence of two commuting subalgebras $\mathcal{E}\mathsf{PTL}_{N_\mathsf{a}}(\beta)$ and $\mathcal{E}\mathsf{PTL}_{N_\mathsf{b}}(\beta)$ in $\mathcal{E}\mathsf{PTL}_{N_\mathsf{a}+N_\mathsf{b}}(\beta)$. These are realised using the so-called *braid translation*, whereby the generators $\Omega$, $\Omega^{-1}$ and $e_0$ of the smaller algebras are embedded in the larger one using braid tiles. With this definition, the fusion of $\mathsf{W}_{k,x}(N_\mathsf{a})$ and $\mathsf{W}_{\ell,y}(N_\mathsf{b})$ is non-commutative and yields a vanishing result except if the ratio $x/y$ is fixed to certain integer powers of $q^{1/2}$. In constrast, our construction gives non-vanishing results for all values of $x$ and $y$, both generic and non-generic. It also only uses the existence of the subalgebra $\mathsf{TL}_{N_\mathsf{a}}(\beta) \otimes \mathsf{TL}_{N_\mathsf{b}}(\beta)$ of $\mathcal{E}\mathsf{PTL}_{N_\mathsf{a}+N_\mathsf{b}}(\beta)$. In terms of this last property, our construction is somewhat closer to the construction of [39]. This prescription for fusion constructs representations of $\mathcal{E}\mathsf{PTL}_{N_\mathsf{a}+N_\mathsf{b}}(\beta)$ from two standard modules $\mathsf{V}_k(N_\mathsf{a}), \mathsf{V}_\ell(N_\mathsf{b})$ of the regular Temperley-Lieb algebra. There are thus no twist parameters in this prescription for the fusion of standard modules. Moreover, in contrast to (3.25), no extra quotient relations are imposed, so the resulting modules are infinite-dimensional. The resulting modules decompose as direct sums of projective indecomposable modules of $\mathcal{E}\mathsf{PTL}_{N_\mathsf{a}+N_\mathsf{b}}(\beta)$, which are themselves infinite-dimensional.

Our work leaves open a number of questions. In particular, we believe it will be worthwhile to investigate in greater detail the module decomposition of $\mathsf{X}_{k,\ell,x,y,z}(N)$ for non-generic values of $q$ and $z$. The example worked out in Section 4.4 reveals that for non-generic $z$, the fusion does not close on the standard modules. Furthermore, the structure of the fused modules are of course expected to be more intricate if $q$ is set to a root of unity. In the scaling limit, the algebra $\mathsf{Vir} \otimes \overline{\mathsf{Vir}}$ is known to admit a large zoo of indecomposable modules. Repeatedly fusing the standard modules together using the fusion prescription defined in this paper should produce a subset of these indecomposable representations, which should be identified as the physically relevant ones for the computation of the connectivity correlation functions. It will also be interesting to work out the fusion rules for the irreducible modules that appear as submodules and quotients in the decomposition of the standard modules. Lastly, the next step in the program would be to apply the conformal bootstrap with the fusion rules obtained in this paper, to obtain expressions for the structure constants in the operator product expansion.

### Acknowledgments

The authors thank Jean-Bernard Zuber and Yvan Saint-Aubin for their comments on the manuscript. The authors also thank the referees for insightful reports.

## A     Properties of the Jones-Wenzl projectors

We list here certain known properties of the Jones-Wenzl projectors $P_n$. These can be represented diagrammatically in the plane in terms of tiles as

The projector $P_n$ thus involves $n(n-1)/2$ diamond boxes. As an element of $\mathsf{TL}_n(\beta)$, it is a sum of $2^{n(n-1)/2}$ diagrams, obtained by expanding each diamond box in terms of the two possible diagrams, and weighted by the proper product of factors $[k]_q/[k+1]_q$. These projectors satisfy the identities

$$P_m P_n = P_n P_m = P_n \qquad \text{for } n \geqslant m\,, \tag{A.2a}$$

$$P_n e_j = e_j P_n = 0 \qquad \text{for } j = 1, \ldots, n-1\,, \tag{A.2b}$$

$$e_n P_n e_n = -\frac{[n+1]_q}{[n]_q} P_{n-1} e_n. \tag{A.2c}$$

From (A.1) and (A.2b), one can show that the projectors also satisfy the recursive relations

$$P_n = (\mathbf{1}_1 \otimes P_{n-1})\left(\mathbf{1} + \sum_{j=1}^{n-1} \frac{[n-j]_q}{[n]_q} e_1 e_2 \cdots e_j\right) \tag{A.3a}$$

$$= (P_{n-1} \otimes \mathbf{1}_1)\left(\mathbf{1} + \sum_{j=1}^{n-1} \frac{[j]_q}{[n]_q} e_{n-1} e_{n-2} \cdots e_j\right). \tag{A.3b}$$

Here, we denote as $a_1 \otimes a_2$ the element of $\mathsf{TL}_N(\beta)$ where $a_1 \in \mathsf{TL}_n(\beta)$ and $a_2 \in \mathsf{TL}_{N-n}(\beta)$ are drawn side-by-side, for any $n < N$.

Finally, it is clear from its recursive definition that $P_n$ can be written as

$$P_n = \mathbf{1} + \sum_{j=1}^{n-1} a_j e_j \tag{A.4}$$

for some elements $a_j \in \mathsf{TL}_n(\beta) \subset \mathsf{TL}_N(\beta)$.

# B    Proofs of the homomorphism relations

In this section, we give proofs of the three homomorphism relations (2.28). First, we show that $w_{k,z}(\ell)$ is non-zero. Using (A.4) for $n = 2\ell$, we find

$$w_{k,z}(\ell) = v_k(\ell) + \sum_{j=1}^{2\ell-1} a_j e_j \cdot v_k(\ell), \qquad \text{for some } a_j \in \mathsf{TL}_{2\ell}(\beta). \tag{B.1}$$

Each term in the sum is a linear combination of link states with strictly less than $\ell - k$ arches crossing the dashed line. In the link state basis, the coefficient of $w_{k,z}(\ell)$ along $v_k(\ell)$ is thus equal to one, confirming that $w_{k,z}(\ell)$ is a non-zero element of $\mathsf{W}_{k,z}(2\ell)$.

Second, the action of $\Omega$ on $w_{k,z}(\ell)$ for $z \in \{q^\ell, -q^\ell, q^{-\ell}, -q^{-\ell}\}$ is derived from the following proposition.

PROPOSITION B.1 *For any values of $q$ and $z$, we have*

$$\Omega \cdot w_{k,z}(\ell) = \lambda_{k,\ell}(z)\, w_{k,z}(\ell) + \mu_{k,\ell}(z)\, (P_{2\ell-1} \otimes \mathbf{1}_1)\Omega \cdot v_k(\ell), \tag{B.2a}$$

*where*

$$\lambda_{k,\ell}(z) = \frac{[\ell+k]_q\, z + [\ell-k]_q\, z^{-1}}{[2\ell]_q}, \tag{B.2b}$$

$$\mu_{k,\ell}(z) = \frac{[\ell-k]_q[\ell+k]_q}{([2\ell]_q)^2}\left\{(q^\ell + q^{-\ell})^2 - (z + z^{-1})^2\right\}, \tag{B.2c}$$

*and $\mathbf{1}_1$ is the identity element of $\mathsf{TL}_1(\beta)$. In particular, for $z = \varepsilon q^{\pm\ell}$ with $\varepsilon \in \{-1, +1\}$, we have $\lambda_{k,\ell}(\varepsilon q^{\pm\ell}) = \varepsilon q^{\pm k}$, and $\mu_{k,\ell}(\varepsilon q^{\pm\ell}) = 0$.*

PROOF. Let us first consider the case $k > 0$. Using the property (A.3b) with $n = 2\ell$, we find

$$w_{k,z}(\ell) = (P_{2\ell-1} \otimes \mathbf{1}_1)\left(1 + \frac{[\ell-k]_q}{[2\ell]_q}e_{2\ell-1}e_{2\ell-2}\cdots e_{\ell-k} + \frac{[\ell+k]_q}{[2\ell]_q}e_{2\ell-1}e_{2\ell-2}\cdots e_{\ell+k}\right) \cdot v_k(\ell). \quad \text{(B.3)}$$

Indeed, all other terms of the sum vanish due to the relations

$$e_j \cdot v_k(\ell) = \begin{cases} e_{2\ell-j} \cdot v_k(\ell) & j \in \{1,\ldots,\ell-k-1\}\cup\{\ell+k+1,\ldots,2\ell-1\}, \\ 0 & j \in \{\ell-k+1,\ldots,\ell+k-1\}. \end{cases} \quad \text{(B.4)}$$

In the first case, the contribution is zero because the resulting generator $e_{2\ell-j}$ annihilates the projector $(P_{2\ell-1} \otimes \mathbf{1}_1)$, see (A.2b). The second and third terms in (B.3) are simplified using

$$e_{2\ell-1}e_{2\ell-2}\cdots e_{\ell-k} \cdot v_k(\ell) = z\,\Omega \cdot v_k(\ell), \qquad e_{2\ell-1}e_{2\ell-2}\cdots e_{\ell+k} \cdot v_k(\ell) = z^{-1}\Omega \cdot v_k(\ell), \quad \text{(B.5)}$$

which yields

$$w_{k,z}(\ell) = (P_{2\ell-1} \otimes \mathbf{1}_1)\left(1 + \frac{[\ell-k]_q z + [\ell+k]_q z^{-1}}{[2\ell]_q}\,\Omega\right) \cdot v_k(\ell) \qquad k > 0. \quad \text{(B.6)}$$

Similarly, for $k = 0$, only two terms contribute to $P_{2\ell} \cdot v_0(\ell)$, namely the identity term and the term $j = \ell$ in the sum (A.3b), and we get

$$w_{0,z}(\ell) = (P_{2\ell-1} \otimes \mathbf{1}_1)\left(1 + \frac{[\ell]_q}{[2\ell]_q}\alpha\,\Omega\right) \cdot v_0(\ell), \quad \text{(B.7)}$$

which shows that (B.6) conveniently extends to $k = 0$. Repeating the above argument with (A.3a) instead of (A.3b), we arrive at the two formulas

$$w_{k,z}(\ell) = (\mathbf{1}_1 \otimes P_{2\ell-1})\left(1 + \frac{[\ell+k]_q z + [\ell-k]_q z^{-1}}{[2\ell]_q}\,\Omega^{-1}\right) \cdot v_k(\ell), \quad \text{(B.8a)}$$

$$w_{k,z}(\ell) = (P_{2\ell-1} \otimes \mathbf{1}_1)\left(1 + \frac{[\ell-k]_q z + [\ell+k]_q z^{-1}}{[2\ell]_q}\,\Omega\right) \cdot v_k(\ell). \quad \text{(B.8b)}$$

From (B.8a), one has

$$\Omega \cdot w_{k,z}(\ell) = (P_{2\ell-1} \otimes \mathbf{1}_1)\left(\Omega + \frac{[\ell+k]_q z + [\ell-k]_q z^{-1}}{[2\ell]_q}\,1\right) \cdot v_k(\ell), \quad \text{(B.9)}$$

which, when compared to (B.8b), yields Proposition B.1 after some simple algebra. ∎

Third, from the property (A.2b) of the Jones-Wenzl projectors, it readily follows that

$$e_j \cdot w_{k,z}(\ell) = 0, \qquad j = 1, \ldots, 2\ell - 1. \quad \text{(B.10)}$$

It thus only remains to show that $e_{2\ell} \cdot w_{k,z}(\ell) = 0$ for $z \in \{q^\ell, -q^\ell, q^{-\ell}, -q^{-\ell}\}$. This stems from the following proposition.

PROPOSITION B.2 *For any values of $q$ and $z$, we have*

$$e_{2\ell} \cdot w_{k,z}(\ell) = h_{k,\ell}(z)\,(\mathbf{1}_1 \otimes P_{2\ell-2} \otimes \mathbf{1}_1) \cdot v_k(\ell), \quad \text{(B.11)}$$

*where*

$$h_{k,\ell}(z) = \frac{[\ell+k]_q[\ell-k]_q}{[2\ell]_q[2\ell-1]_q}\left\{(z+z^{-1})^2 - (q^\ell + q^{-\ell})^2\right\}, \quad \text{(B.12)}$$

*and $\mathbf{1}_1$ is the identity element of $\mathsf{TL}_1(\beta)$.*

PROOF. We start by applying $e_{2\ell}$ to (B.8b), namely

$$e_{2\ell} \cdot w_{k,z}(\ell) = e_{2\ell} \left(P_{2\ell-1} \otimes \mathbf{1}_1\right) \left(\mathbf{1} + \frac{[\ell-k]_q\, z + [\ell+k]_q\, z^{-1}}{[2\ell]_q} \Omega\right) \cdot v_k(\ell) \ . \tag{B.13}$$

We can then simplify each term in (B.13) separately. The first simplifies to

$$e_{2\ell}(P_{2\ell-1} \otimes \mathbf{1}_1) \cdot v_k(\ell) = \frac{1}{\beta}\, e_{2\ell}(P_{2\ell-1} \otimes \mathbf{1}_1)e_{2\ell} \cdot v_k(\ell) = -\frac{1}{\beta}\frac{[2\ell]_q}{[2\ell-1]_q}(\mathbf{1}_1 \otimes P_{2\ell-2} \otimes \mathbf{1}_1)e_{2\ell} \cdot v_k(\ell)$$

$$= -\frac{[2\ell]_q}{[2\ell-1]_q}(\mathbf{1}_1 \otimes P_{2\ell-2} \otimes \mathbf{1}_1) \cdot v_k(\ell). \tag{B.14}$$

At the second equality, we used the reflected version of (A.2c). Expanding the projector $P_{2\ell-1}$ of the second term of (B.13) using (A.3a), for similar reasons as above we find that only two terms survive:

$$e_{2\ell}(P_{2\ell-1} \otimes \mathbf{1}_1)\Omega \cdot v_k(\ell) = e_{2\ell}(\mathbf{1}_1 \otimes P_{2\ell-2} \otimes \mathbf{1}_1)\left(\frac{[\ell+k]_q}{[2\ell-1]_q}e_1 e_2 \cdots e_{\ell-k-1}\right.$$

$$\left. + \frac{[\ell-k]_q}{[2\ell-1]_q}e_1 e_2 \cdots e_{\ell+k-1}\right)\Omega \cdot v_k(\ell). \tag{B.15}$$

We commute $e_{2\ell}$ across $\mathbf{1}_1 \otimes P_{2\ell-2} \otimes \mathbf{1}_1$, use the properties

$$e_{2\ell}e_1 e_2 \cdots e_{\ell-k-1}\Omega \cdot v_k(\ell) = z\, v_k(\ell), \qquad e_{2\ell}e_1 e_2 \cdots e_{\ell+k-1}\Omega \cdot v_k(\ell) = z^{-1}v_k(\ell), \tag{B.16}$$

and find that the second term in (B.13) is also proportional to $(\mathbf{1}_1 \otimes P_{2\ell-2} \otimes \mathbf{1}_1)v_k(\ell)$. The final result precisely has the form (B.11) with $h_{k,\ell}(z)$ given by

$$h_{k,\ell}(z) = -\frac{[2\ell]_q}{[2\ell-1]_q} + \left(z\frac{[\ell-k]_q}{[2\ell]_q} + z^{-1}\frac{[\ell+k]_q}{[2\ell]_q}\right)\left(z\frac{[\ell+k]_q}{[2\ell-1]_q} + z^{-1}\frac{[\ell-k]_q}{[2\ell-1]_q}\right). \tag{B.17}$$

This expression simplifies to (B.12) after simple algebra. ∎

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
