# Peer review of "Fusion in the periodic Temperley-Lieb algebra and connectivity operators of loop models"

_SciPost Physics_

## Round 1 · Referee Report · Anonymous (Referee 2) · 2021-7-6

Strengths

  1. New construction for the fusion of Temperley-Lieb fused modules
  2. Impressive main result (Theorem 1)
  3. Good explanations and illustrations of the construction

Weaknesses

  1. Very sketchy section 5

Report

The main result of the paper is really interesting: the new fused representation of the enlarged periodic Temperley-Lieb algebra. The main theorem (Theorem 1) on the decomposition of modules is very impressive and seems to be helpful for the computation of the correlation functions. I recommend this paper for publication after minor revision suggested below.

Requested changes

  1. The main result (Theorem 1) needs some comments: while the representation depends on the parameters $x$ and $y$ they disappear completely in the decomposition in right hand side. I think that authors should comment about the role of these parameters in the module decomposition.
  2. The part on the correlation functions (subsection 5.2) is extremely sketchy. The Gram product with defects is not really defined and there is no comments about the parameters of the Gram product pertinent for the computation of the correlation functions.
  3. The authors should correct the first example in (3.10c)
  4. The authors should (probably) better explain the identification (3.11) it is not completely clear how it appears.

---

## Round 1 · Referee Report · Jules Lamers (Referee 1) · 2021-7-6

Strengths

1- The authors introduce a new class of representations with two marked points of the extended periodic (= affine) Temperley-Lieb algebra, and study their structure 2- These representations are interpreted as the result of fusion, and used to study the operator product expansion of observables for the dense $O(n)$ loop model 3- Sections 2-4 are self contained and detailed 4- Beautiful diagrams

Weaknesses

1- In Section 3.2 it is not clear whether the module $X_{k,l,x,y,z}(N)$ is well defined for $k,l>0$ 2- The discussion in Section 3.3 seems to need some clarification 3- Some mathematical terminology is used imprecisely, which may lead to confusion

Report

The extended periodic (= affine) Temperley-Lieb algebra $EPTL_N$ is closely related to exactly-solvable models from statistical mechanics, such as the dense $O(n)$ model, and conformal field theory. This algebra has standard modules $W_{k,x}(N)$ with $2k$ defects (= lines connected to a marked point) and $x$ parametrising either the weight of loops enclosing the marked point ($k=0$) or the unwinding of defects ($k>0$).
Most of this paper is devoted to the definition and study of a new family of (reducible) $EPTL_N$-modules $X_{k,l,x,y,z}(N)$ with $two$ marked points. The authors interpret these modules as the fusion of two standard modules $W_{k,x}(N_a)$ and $W_{l,y}(N_b)$ with $N_a+N_b = N$, further involving a twist parameter $z$. This construction differs from the proposals for fusion of Gainutdinov et al [37,38] and of Belletête and Saint-Aubin [39].
For generic values of the parameters the authors obtain the decomposition of $X_{k,l,x,y,z}(N)$ in terms of irreducible standard modules. The authors present a conjecture for a simple factorised component of a particular vector in $X_{0,0,x,y,z}(N)$. For particular non-generic values of $z$ they show that $X_{0,0,x,y,z}(N)$ features a piece that is reducible but indecomposible.
In the final section $X_{k,l,x,y,z}(N)$ is related to so-called connectivity operators, and the authors exploit the consequences of the decomposition of $X_{k,l,x,y,z}(N)$ for generic parameters to argue that their construction fits with the fusion rules for the computation of correlation functions in the scaling limit.

The material of Sections 2-4 is essentially self-contained and quite detailed. I enjoyed reading these sections and believe it has the potential to open a new pathway, with clear potential for multipronged follow-up work. Section 5, on which I am admittedly not an expert, appears to be less self-contained and seems less clear to me. For this reason I mostly have comments on the first sections.

One concern is that there appears to be an inconsistency in Section 3.2 in the definition of $X_{k,l,x,y,z}(N)$ for $k,l>0$. In this case the action of $EPTL_N$ can connect defects between the two marked points, and the rules for reading off the corresponding weights are not quite clear, and I am not convinced that the parameters $x,y,z$ are indeed independent in this case.
Specifically, the results seem to differ depending on whether one goes from $a$ to $b$ or the other way around. For example, the second diagram in (3.13) gives $x^{-1}$ as indicated when one resolves the winding by viewing this as a defect starting at $a$ (green), but $(yz)^{-1}$ when one starts at $b$ (purple) instead. Consistency appears to require $x=yz$. The same holds for the third to fifth example (reading column by column). Moreover, the middle diagram in the second column naively appears to have four different ways of attaching a weight; the two ways where the weights are read off by going from $a$ to $b$ (or reversely) seem to give weight $y(yz)^{-1} = x^{-1}(xz^{-1})=z^{-1}$? The situation is even less clear for the next example.

Secondly, the characterisation of $X_{k,l,x,y,z}(N)$ in Section 3.3 is not quite clear to me.
In part this is because $EPTL_{N_a} \otimes EPTL_{N_b}$ is not obviously a subalgebra of $EPTL_{N}$. By viewing $W_{k,x}(N_a)$ as a module for $TL_{N_a} \subset EPTL_{N_a}$, as written in the text, one loses the dependence on the parameter $x$, yielding $V_k(N_a) = \mathrm{Res}^{EPTL_{N_a}}_{TL_{N_a} } \, W_{k,x}(N_a)$. It is somewhat misleading to retain the notation $W_{k,x}(N_a)$. The way out might be to instead restrict from $EPTL_{N}$ to $PTL_{N_a} \otimes PTL_{N_b}$. If I'm not mistaken this is a subalgebra (with, in particular, $e_{N_a} \otimes 1 \mapsto e_{N_a}$ and $1 \otimes e_{N_b} \mapsto e_N$), and one can keep the dependence on the parameters $W_{k,x}(N_a) \otimes W_{l,y}(N_b)$ and induce up to the full algebra.
In addition, it is not clear to me how the parameter $z$ is exactly included from this point of view. Can the sentence following (3.19) be made precise by, e.g., taking some quotient?

Once this fusion procedure is defined one naturally wonders if it is associative (perhaps up to isomorphism). This question is not addressed in Sections 2-4, and only implicitly in Section 5.

Requested changes

My main requests are 1- Address the apparent inconsistency in Section 3.2 as per above 2- Clarify Section 3.3 as per above 3- Explain the associativity of the proposed fusion (preferably at the level of Sections 2-4 rather than 5) 4- Theorem 1: the right-hand side is independent of $x,y$, which seems surprising (cf the standard modules $W_{k,x}$ and $W_{k,y}$ are not isomorphic). Can you comment on this?

Additional minor requests/comments - Sect 1 (and perhaps 2.1): mention the related work of Al Qasimi et al, arXiv:1710.04058 and 1903.08677 - Sect 2.1: the definition of $[k]_q$ is somewhat unusual; although I understand the connection to the fairly common short-hand $[x] = \theta(x), \sinh(x), \sin(x),\dots$ I would be in favour of letting $[k]_q = (q^k - q^{-k})/(q-q^{-1})$ be the $q$-analogue of $k \in \mathbb{N}$, which would make $\beta = -[2]_q$ look simpler, whereas (2.5) and App A,B are unaffected as the values occur in ratios - Sect 2.1: for completeness, mention that $PTL_N$ is the subalgebra generated by $e_1,\dots,e_N$ - Jones-Wenzl: perhaps point out that these are not orthogonal - Sect 2.2: might be useful to include the relevant references (probably mentioned at the start of 2.1) for the special elements in Sect 2.2 - $F$, $\bar{F}$ and $\Omega^N$ all lie in the center of $EPTL_N$; do they generate it? - Sect 2.3: mention $z\in\mathbb{C}^\times$ from the start - define 'link state' (= basis element in terms of diagrams without closed loops and up to homotopy?) - p7: it would be useful to give or summarise the rules defining the standard action diagrammatically for easy reference - "Loops cannot encircle the marked point in this case": perhaps clarify that the action of $F,\bar{F}$ has to be expanded via (2.7) - emphasise that $z$ enters in separate ways depending on whether $k=0$ (via $\alpha=z+z^{-1}$) or $k>0$; are these occurences completely independent? - (2.14) would be more clear with $\alpha$ rather than $z+z^{-1}$ (easier comparison for the case of two defects later on) - point out whether the projectors (2.17) are orthogonal - the line between (2.21) and (2.22) further uses that the standard modules are also independent under inverting $q$ - after (2.22) perhaps point out that these homomorphisms will be constructed explicitly in Sect 2.5 - still p8: "... have identical eigenvalues..." clarify: the same eigenvalue (as opposed to the same spectrum, including multiplicities) - quotient $W_{k,\varepsilon q^{\pm l}}/...$ should be by the image of $W_{l,\varepsilon q^{\pm k}}$ or by the irrep $I$. - I'm confused by (2.24) and the following text: either the $I$s are irreducibles (quotient for the head), or they are summands (Grothendieck sum) that are not direct sums (as modules) so that the arrow and sentence "... can produce non-zero states" seem to make sense; please clarify - Sect 2.5: the discussion seems to fix $k,l$ and vary $N$, while a priori it seems more natural to fix $k,N$ and vary $l$; although equivalent it would help to either change this or explain the viewpoint - p10 (twice) and in the remainder of the text (several times): the authors write that states generate a (sub)module, where I believe it's meant that they span that space -- this is potentially confusing, as there is a (different) notion of (cyclic, or finitely generated) modules that are generated by a few vectors; please correct here and elsewhere - below (2.29): "the action of ... on these four states is invariant" $\to$ leaves the states invariant? - just below, "... vanishing result if two nodes attached to the projector ..." and "... permutes the nodes tied to the projector ...": nodes = links? - preceding (3.1): what is meant precisely by 'obtained from'? Is $V_k(N_a) \times V_l(N_b)$ equal to (3.1), obtained from it in some way - maybe good to mention that the construction in (3.1) is called a tensor product over a subalgebra - a little before (3.5), "... the representations depend only on the sum $N_a + N_b$": as in their dependence on $N_a,N_b$ is on the sum only, or it does not depend on anything else (I think it also depends on $k,l$) - just before (3.5), "the fusion ... is read from the decomposition": what does this mean precisely? - (3.5) or text preceding: mention that $N = N_a + N_b$ - just below (3.5): here fusion = decomposition of tensor product? (in the context of Yangians, quantum loop algebras, etc it is used for a construction involving a projection onto one irreducible component in such a decomposition) - (3.8) and preceding text: in addition to these examples it would again be useful to give or summarise the rules defining the standard action diagrammatically for easy reference - cases $k>0,l=0$ and $k=0,l>0$: need to change $k\leftrightarrow l$ - before (3.9), "the defects do not cross * the two dashed segments ": is meant * either or * both ? (likewise in the case $k,l>0$) - p14, "In this construction, the point $c$ ...": this was already relevant in the case $k=l=0$ - just below: "also connects the defects of $b$ together": I believe that this should be $a$? -footnote 2: formulation unclear; "cyclic" means generated by a single vector, but here it's written that it's produced by the action on the states (plural) - (3.13): clarify what "diagram : weight" means; can we replace the diagram by weight (or is it weight times the diagram with two marked points but no lines)? - same equation: $\mu$ appears out of nowhere, whereas up to now the ordering of the two marked points didn't seem to matter; please clarify this before presenting those examples in (3.13) - (3.15b), first equality: this actually seems to require something slightly stronger than the rules given so far, namely to move a defect through a dashed line (unwind it) in the case that both marked points have a defect, but without producing crossing blue lines; are we allowed to just exchange the positions of the marked points, or is this where we introduce a weight $\mu$? - likewise for (3.16), second equality - (3.16), first equality: this weight seems to differ from that in the middle diagram in the right-most column of (3.13) - Sect 3.3: mention again that $N=N_a+N_b$ - (4.18) include the name $\Phi$ here already - (4.25) what's $\mathrm{sign}(0)$? - preceding (4.32): the vector is at most unique up to nonzero scalar multiples - just above Prop 4.1, "unique state": again, it's certainly not unique without further qualification; replace e.g. by "single state" - right before Prop 4.1, "on the states of maximal depth": is meant any single state of maximal depth? - Prop 4.1 or just below: recall that $v_{k,l}(m)$ was defined in (4.13) - Sect 4.4, "reflect $v$ vertically": just to clarify, is really meant that $v$ should be reflected so that the dashed lines now meet at the top? - just before (4.34): is the scalar product to be extended bilinearly or sesquilinearly? - just before "Conjectural form for ...": is the radical the or a* maximal submodule? - (5.2b): does this also hold for $i=j+k-1$ or only up to $i=j+k-2$? - (5.2c): does this also hold for $j=1$? (seems to require some unwinding) - (5.4): does this also hold when $k>l$, so that there will be at least on link from $a$ to $b$? - (5.11): there does not seem to be any need to indicate the little arcs in this context, without spectral parameter? - (A.1): for completeness consider explaining how the diagonal tile works, since the main text only uses square/rectangular tiles - p39, top: $w_{k,z}(l)$ is not quite invariant (but the subspace it spans is) - Prop B.1, proof: it would make more sense here to prove (B.8a) and leave (B.8b) as an exercise to be done analogously, since the proposition is actually proven from (B.8a)

Finally, a few typos - As the second author will know, compounding authors (Temperley-Lieb, Graham-Lehrer, Jones-Wenzl, Fortuin-Kasteleyn) should really be done using an em-dash (-- in latex, unfortunately unavailable here) rather than a hyphen - p7: "However, it will turn out to be more natural" - p8: "We say that the parameter $z$ of" - Sect 3.1: $TL_{N_a} \otimes TL_{N_b}$ missing $\beta$ - p13 top: no neither nor ? - below (3.12): "is a link state in" - p29, last line: redundant "are"

---

## Editorial Decision

resubmitted